# Predicting What You Already Know Helps: Provable Self-Supervised Learning

Jason D. Lee[1],   Qi Lei[1],   Nikunj Saunshi[1],   Jiacheng Zhuo[2]

[1] Princeton University    [2] University of Texas at Austin
{jasonlee@,qilei@,nsaunshi@cs}.princeton.edu,  jzhuo@utexas.edu

## Abstract

Self-supervised representation learning solves auxiliary prediction tasks (known as pretext tasks) without requiring labeled data to learn useful semantic representations. These pretext tasks are created solely using the input features, such as predicting a missing image patch, recovering the color channels of an image from context, or predicting missing words in text; yet predicting this *known* information helps in learning representations effective for downstream prediction tasks.

We posit a mechanism exploiting the statistical connections between certain *reconstruction-based* pretext tasks that guarantee to learn a good representation. Formally, we quantify how the approximate independence between the components of the pretext task (conditional on the label and latent variables) allows us to learn representations that can solve the downstream task by just training a linear layer on top of the learned representation. We prove the linear layer yields small approximation error even for complex ground truth function class and will drastically reduce labeled sample complexity.

## 1 Introduction

Self-supervised learning revitalizes machine learning models in computer vision, NLP, and control problems (see reference therein [36, 38, 15, 63, 35]). Training a model with auxiliary tasks based only on input features reduces the extensive costs of data collection and semantic annotations for downstream tasks. It is also known to improve the adversarial robustness of models [29, 11, 12]. Self-supervised learning creates pseudo labels solely based on input features, and solves auxiliary prediction tasks (or pretext tasks) in a supervised manner. However, the underlying principles of self-supervised learning are mysterious since it is a-priori unclear why predicting what we already know should help. We thus raise the following question:

*What conceptual connection between pretext and downstream tasks ensures good representations?*
*What is a good way to quantify this?*

As a thought experiment, consider a simple downstream task of classifying desert, forest, and sea images. A meaningful pretext task is to predict the background color of images (known as image colorization [66]). Denote $X_1, X_2, Y$ to be the input image, color channel, and the downstream label respectively. Given knowledge of the label $Y$, one can possibly predict the background $X_2$ without knowing much about $X_1$. In other words, $X_2$ is approximately independent of $X_1$ conditional on the label $Y$. Consider another task of inpainting [48] the front of a building ($X_2$) from the rest ($X_1$). While knowing the label "building" ($Y$) is not sufficient for successful inpainting, adding additional latent variables $Z$ such as architectural style, location, window positions, etc. will ensure that variation in $X_2$ given $Y, Z$ is small. We can mathematically interpret this as $X_1$ being approximate conditionally independent of $X_2$ given $Y, Z$.

The main insight that we exploit in this work is that with approximate conditional independence (as in the above examples), a method that predicts $X_2$ from $X_1$ will inadvertently implicitly encode and

35th Conference on Neural Information Processing Systems (NeurIPS 2021).

learn to predict $Y$ (and $Z$) from $X_1$ as an intermediate step, and then predict $X_2$ from $Y$[1]. Building upon this insight, we make the following contributions.

**Contributions.** The goal of this paper, as in statistical learning theory, is to investigate the *statistical connections* between the random variables of input features (in this paper $(X_1, X_2)$) and downstream labels $Y$, and show how specific connections can guarantee a successful learning procedure. For self-supervised learning (SSL), success is measured using the following 2 notions, 1) expressivity, i.e. does the learned representation from SSL have the ability to express the ground truth prediction function for labels $Y$, and 2) sample complexity, i.e. can it do so with way fewer labeled samples than what would be required without SSL.

In this work, we show such guarantees for a class of *reconstruction-based* SSL methods under a statistical assumption of *approximate conditional independence (ACI)*. In particular we show that under such an assumption, the learned representation from SSL will end up having the following properties, 1) it can express the ground truth label as a *linear function*, thus guaranteeing expressivity, and 2) will also end up being low-rank (or low-dimensional), thus guaranteeing smaller labeled sample complexity. Note that such an expressive and sample efficient (summarized as *good*) representation is often not a-priori available. For instance, the original input features themselves may not be able to express the ground truth function linearly, while kernel methods with a fixed kernel, while expressive, may not be sample efficient for many problems of interest. The strategy in modern machine learning is to find such a *good* representation as the output of a complicated neural network. The benefit of SSL, as we formally show here, is that the complicated but *good* representation function can be learned using just *unlabeled data*, so that labeled data is just needed to learn a linear function.

The *reconstruction-based* SSL method (differentiated from other SSL methods in Section 1.1) we consider is strongly motivated by empirical works [66, 48, 15, 25], but is a simplification that captures the essence of the problem and is amenable to a precise theoretical analysis. We consider a two-staged pipeline, where we first learn a representation function $\psi$ (e.g. output of a neural network) from input $X_1$ and pretext target $X_2$ using *unlabeled data* by minimizing $\mathbb{E}_{(X_1, X_2)}[\|X_2 - \psi(X_1)\|^2]$. In the second stage of downstream task, we learn a *linear layer* on top of representation $\psi$ using labeled samples $(X_1, Y)$, thus restricting to learning from a significantly smaller hypothesis class of $\mathcal{H}_\psi = \{f : X_1 \rightarrow Y | f \text{ is linear in } \psi\}$. The key non-trivial question of expressivity is now whether the ground truth predictor $f^* \equiv \mathbb{E}[Y|X_1]$ can be approximated well by this class $\mathcal{H}_\psi$, and the question of sample complexity reduces to the understanding the sample complexity of learning $\mathcal{H}_\psi$. Under appropriate statistical connections[2] between input data $X_1, X_2$ and target $Y$, we prove both the desired properties, expressivity and low sample complexity, for the aforementioned SSL method.

Our statistical assumption based on approximate conditional independence (ACI) helps us demonstrate how solving pretext tasks created from *known information* can learn useful representations. Specifically, we show that once the complicated representation function $\psi$ is learned using an abundance of *unlabeled data* in the SSL stage, not only is $\psi$ expressive enough, but it will also require only $\tilde{\mathcal{O}}(k)$ *labeled samples* to solve a $k$-way supervised learning task under exact conditional independence (CI). In contrast, solving the downstream task without any pretraining will require a lot of labeled data to learn the representation function from scratch. Since the strong exact conditional independence assumption will likely not be satisfied in practice, our main contribution is to derive similar risk bounds when only approximate CI (ACI) is satisfied. We quantify the notion of ACI using the norm of a certain partial covariance matrix (Definition 4.1) and our risk bound scales linearly with it. We verify this and other aspects of our main Theorem 4.2 using simulations and also find that pretext task helps when CI is approximately enforced in text domain. We further demonstrate on a real-world image dataset that a pretext task-based linear model performs at least as well as many baselines.

## 1.1 Related work

**Self-supervised learning (SSL) methods in practice:** There has been a flurry of self-supervised methods lately. One class of methods reconstruct images from corrupted or incomplete versions of it, like denoising auto-encoders [61], image inpainting [48], and split-brain autoencoder [67]. Pretext

---

[1]This is formally demonstrated in the proof sketch of Lemma 3.1.
[2]We note that since the representation $\psi$ is the result of the SSL method and not something we have access to a-priory, we cannot make any direct assumptions on it.

tasks are also created using visual common sense, including predicting rotation angle [22], relative patch position [16], recovering color channels [66], solving jigsaw puzzle games [45], and discriminating images created from distortion [17]. We refer to the above procedures as reconstruction-based SSL. Another popular paradigm is contrastive learning [13, 14]. The idea is to learn representations that bring similar data points closer while pushing randomly selected points further away [63, 39, 5] or to maximize a contrastive-based mutual information lower bound between different views [30, 46, 54]. A popular approach for text domain is based on language modeling where models like BERT and GPT create auxiliary tasks for next word predictions [15, 49]. The natural ordering or topology of data is also exploited in video-based [64, 43, 19], graph-based [65, 33] or map-based [68] SSL. For instance, the pretext task is to determine the correct temporal order for video frames as in [43].

**Theory for SSL:**   While we theoretically study reconstruction-based SSL, prior work has different flavors of theoretical results for different kinds of SSL methods. Most relevant are the guarantees for representation learning using SSL methods on downstream tasks that just learn a linear classifier on top of the learned representations. [5] shows guarantees for representations from a contrastive learning objective: $L_1^{cont}(\psi) = \mathbb{E}_{(X_1, X_2), X_2'}[\log(1 + e^{-\psi(X_1)^\top \psi(X_2) + \psi(X_1)^\top \psi(X_2')})]$. Under a class conditional independence assumption, i.e. $X_1 \perp X_2 \mid Y$, they show that representation $\psi$ that does well on contrastive objective, i.e. $L_1^{cont}(\psi) \leq \epsilon$, will have $\mathcal{O}(\epsilon)$ linear classification loss on the average binary task involving pairs of classes $(y_1, y_2)$. However, their analysis cannot handle the general case of approximate conditional independence. Recently, Tosh *et al.* [56] show that contrastive learning representations can *linearly* recover continuous functions of the underlying topic posterior under a topic modeling assumption for text. While their assumption bears similarity to ours, the assumption of independent sampling of words is strong and does not generalizable to other domains like images. Most relevant is a concurrent work [57] that shows guarantees for a contrastive learning objective that looks like $L_2^{cont}(\psi, \eta) = \mathbb{E}_{(X_1, X_2), X_2'} \left[ \log(1 + e^{-\psi(X_1)^\top \eta(X_2)}) + \log(1 + e^{\psi(X_1)^\top \eta(X_2')}) \right]$, with a multi-view redundancy assumptions that is very similar to our ACI assumption. We take a closer look at their assumption in Section F.2. All the above objectives are different from the simple reconstruction-based objective we consider: $L(\psi) = \mathbb{E}_{(X_1, X_2)} \left[ \|X_2 - \psi(X_1)\|^2 \right]$. Saunshi *et al.* [51] show guarantees for representations learned using language modeling on sentence classification tasks. Some more recent work [58, 44, 55, 62] provide theoretical understanding on SSL respectively based on causality, mutual information, gradient-descent dynamics, and alignment/uniformity of representations, without explicit risk bounds for downstream tasks. There is a mutual information maximization view of contrastive learning, but [59] points out issues with it. Previous attempts to explain negative sampling [42] based methods use the theory of noise contrastive estimation [27, 40] to show asymptotic guarantees, without explicit connections to downstream tasks. CI is also used in sufficient dimension reduction [21, 20], while CI and redundancy assumptions on multiple views [37, 2] are used to analyze a canonical-correlation based dimension reduction algorithm and also for self-supervised learning algorithms like co-training [10]. Finally, [1, 60] provide a theoretical analysis for denoising auto-encoder.

## 1.2   Overview of results:

Section 2 introduces notation, setup, and the self-supervised learning procedure considered in this work. In Section 3, we analyze downstream sample complexity under exact CI and unlimited labeled data to highlight the key ideas. Section 4 presents our main result with relaxed conditions: under ACI with latent variables, and assuming finite samples in both pretext and downstream tasks, for various function classes, and both regression and classification tasks. Experiments verifying our theoretical findings are in Section 6. Proofs of most results are in the Appendix.

## 2   Preliminary

### 2.1   Notation

We use lower case symbols $(x)$ to denote scalar quantities, bold lower case symbols $(\boldsymbol{x})$ for vector values, capital letters $(X)$ for random variables, and capital and bold letters $\boldsymbol{X}$ for matrices. $P_X$ denotes the probability law of random variable $X$, and the space of square-integrable functions with probability $P$ is denoted by $L^2(P)$. We use standard $\mathcal{O}$ notation to hide universal factors and $\tilde{\mathcal{O}}$ to hide log factors. $\|\cdot\|$ stands for $\ell_2$-norm for vectors or Frobenius norm for matrices.

**Linear conditional expectation.** $\mathbb{E}^L[Y|X]$ denotes the prediction of $Y$ with linear regression:

$$\mathbb{E}^L[Y|X = \boldsymbol{x}] := \boldsymbol{W}^*\boldsymbol{x} + \boldsymbol{b}^*, \quad \text{where } \boldsymbol{W}^*, \boldsymbol{b}^* := \arg\min_{\boldsymbol{W},\boldsymbol{b}} \mathbb{E}[\|Y - \boldsymbol{W}X - \boldsymbol{b}\|^2].$$

In other words, $\mathbb{E}^L[Y|X]$ denotes the best linear predictor of $Y$ given $X$. We also note that $\mathbb{E}[Y|X] \equiv \arg\min_f \mathbb{E}[\|Y - f(X)\|^2]$ is the best predictor of $Y$ given $X$.

**(Partial) covariance matrix.** For random variables $X, Y$, we denote $\boldsymbol{\Sigma}_{XY}$ to be covariance matrix of $X$ and $Y$. For simplicity in most cases, we assume $\mathbb{E}[X] = 0$ and $\mathbb{E}[Y] = 0$; thus we do not distinguish $\mathbb{E}[XY]$ and $\boldsymbol{\Sigma}_{XY}$. The partial covariance matrix between $X$ and $Y$ given $Z$ is:

$$\boldsymbol{\Sigma}_{XY|Z} := \mathrm{cov}\{X - \mathbb{E}^L[X|Z], Y - \mathbb{E}^L[Y|Z]\} \equiv \boldsymbol{\Sigma}_{XY} - \boldsymbol{\Sigma}_{XZ}\boldsymbol{\Sigma}_{ZZ}^{-1}\boldsymbol{\Sigma}_{ZY}, \tag{1}$$

which captures the correlation between $X$ and $Y$ setting aside the effect of $Z$.

**Sub-gaussian random vectors.** $X \in \mathbb{R}^d$ is $\rho^2$-sub-gaussian if for every fixed unit vector $\boldsymbol{v} \in \mathbb{R}^d$, the variable $\boldsymbol{v}^\top X$ is $\rho^2$-sub-gaussian, i.e., $\mathbb{E}[e^{s\cdot\boldsymbol{v}^\top(X-\mathbb{E}[X])}] \leq e^{s^2\rho^2/2}$ ($\forall s \in \mathbb{R}$).

## 2.2 Setup and methodology

We denote by $X_1$ the input variable, $X_2$ the target random variable for the pretext task, and $Y$ the label for the downstream task, with $X_1 \in \mathcal{X}_1 \subset \mathbb{R}^{d_1}$, $X_2 \in \mathcal{X}_2 \subset \mathbb{R}^{d_2}$ and $Y \in \mathcal{Y} \subset \mathbb{R}^k$. If $\mathcal{Y}$ is finite with $|\mathcal{Y}| = k$, we assume $\mathcal{Y} \subset \mathbb{R}^k$ is the one-hot encoding of the labels. $P_{X_1 X_2 Y}$ denotes the joint distribution over $\mathcal{X}_1 \times \mathcal{X}_2 \times \mathcal{Y}$. $P_{X_1 Y}, P_{X_1}$ denote the corresponding marginal distributions. Our proposed self-supervised learning aims to fulfill the following two steps:

*Step 1 (pretext task):* Learn a representation $\psi(\boldsymbol{x}_1)$ close to $\psi^* := \arg\min_{g\in\mathcal{H}} \mathbb{E}\|X_2 - g(X_1)\|^2$, where $\mathcal{H}$ can vary for different settings that we will specify and discuss later.

*Step 2 (downstream task):* Perform linear regression on $Y$ with $\psi(X_1)$, i.e. $f(\boldsymbol{x}_1) := (\boldsymbol{W}^*)^\top\psi(\boldsymbol{x}_1)$, where $\boldsymbol{W}^* \leftarrow \arg\min_{\boldsymbol{W}} \mathbb{E}_{X_1,Y}[\|Y - \boldsymbol{W}^\top\psi(X_1)\|^2]$. Namely we learn $f(\cdot) = \mathbb{E}^L[Y|\psi(\cdot)]$.

We study this simplified version in the main text, where in practice, the SSL procedure may utilize an encoder-decoder structure, while the downstream task uses both $X_1$ and $X_2$ to predict $Y$. We incorporate these extensions in Appendix C.3 and G.

With finite samples, performance of a learned representation $\psi$ on the downstream task depends on the following quantities that capture expressivity and sample complexity respectively:

**Approximation error** indicates whether $Y$ is *linearly separable* by the learned representation $\psi$, thus measuring expressivity. We measure this by comparing $\boldsymbol{W}\psi(X_1)$ to the optimal predictor $f^* := \mathbb{E}[Y|X_1 = \boldsymbol{x}_1]$. Denote $e_{\mathrm{apx}}(\psi) = \min_{\boldsymbol{W}} \mathbb{E}[\|f^*(X_1) - \boldsymbol{W}\psi(X_1)\|^2]$. This gives a measure of how well $\psi$ can linearly predict $Y$ when given infinite samples for the task.

**Estimation error** measure sample complexity of $\psi$ on the downstream task and assume access to $n_2$ i.i.d. samples $(\boldsymbol{x}_1^{(1)}, \boldsymbol{y}^{(1)}), \cdots, (\boldsymbol{x}_1^{(n_2)}, \boldsymbol{y}^{(n_2)})$ drawn from $P_{X_1 Y}$. We express the $n_2$ samples collectively as $\boldsymbol{X}_1^{\mathrm{down}} \in \mathbb{R}^{n_2\times d_1}$, $\boldsymbol{Y} \in \mathbb{R}^{n_2\times k}$ and overload notation to say $\psi(\boldsymbol{X}_1^{\mathrm{down}}) = \left[\psi(\boldsymbol{x}_1^{(1)})|\psi(\boldsymbol{x}_1^{(2)})\cdots|\psi(\boldsymbol{x}_1^{(n_2)})\right]^\top \in \mathbb{R}^{n_2\times d_2}$. We perform linear regression on the learned representation $\psi$ and measure excess risk, that incorporates both approximation and estimation errors.

$$\hat{\boldsymbol{W}} \leftarrow \arg\min_{\boldsymbol{W}} \frac{1}{2n_2}\|\boldsymbol{Y} - \psi(\boldsymbol{X}_1)\boldsymbol{W}\|_F^2; \quad \mathrm{ER}_\psi(\hat{\boldsymbol{W}}) := \frac{1}{2}\mathbb{E}\|f^*(X_1) - \hat{\boldsymbol{W}}^\top\psi(X_1)\|_2^2.$$

# 3 Guaranteed recovery with conditional independence

In this section, we focus on the case where the input $X_1$ and pretext target $X_2$ are conditionally independent (CI) given the downstream label $Y$. While this is a strong assumption that is rarely satisfied in practice, it helps us understand the role of CI with clean results and builds up to our main results with ACI with latent variables in Section 4. As a warm-up, we show how CI helps when $(X_1, X_2, Y)$ are jointly Gaussian to give us a flavor for the results to follow in Appendix B. We then analyze it for general random variables under two settings: (a) when the function class used for $\psi$ is universal, (b) when $\psi$ is restricted to be a linear function of given features. For now we assume access to a large amount of unlabeled data so as to learn the optimal $\psi^*$ perfectly and this will be relaxed later in Section 4. The general recipe for the results is as follows:

1. Find a closed-form expression for the optimal solution $\psi^*$ for the pretext task.
2. Use conditional independence to show that optimal $f^*$ is linear in $\psi^*$, i.e., $e_{\text{apx}}(\psi^*)$ is small.
3. Exploit the low rank structure of $\psi^*$ to show small estimation error on downstream tasks.

**Data assumption.** Suppose $Y = f^*(X_1) + N$, where $f^* = \mathbb{E}[Y|X_1]$ and $\mathbb{E}[N] = 0$. We assume $N$ is $\sigma^2$-subgaussian. For simplicity, we assume non-degeneracy: $\mathbf{\Sigma}_{X_i X_i}$, $\mathbf{\Sigma}_{YY}$ are full rank.

**Assumption 3.1.** *Let $X_1 \in \mathbb{R}^{d_1}$, $X_2 \in \mathbb{R}^{d_2}$ be random variables from some unknown distribution. Let label $Y \in \mathcal{Y}$ be a discrete random variable with $k = |\mathcal{Y}| < d_2$. We assume conditional independence: $X_1 \perp X_2 | Y$.*

Here $Y$ can be interpreted as the multi-class labels where $k$ is the number of classes. For regression problems, one can think about $Y$ as the discretized values of continuous labels. We do not specify the dimension for $Y$ since $Y$ could be arbitrarily encoded but the results only depend on $k$ and the variance of $Y$ (conditional on the input $X_1$).

### 3.1 Universal function class.

Suppose we learn the optimal $\psi^*$ among all measurable functions The optimal function $\psi^*$ in this case is naturally given by conditional expectation: $\psi^*(\boldsymbol{x}_1) = \mathbb{E}[X_2|X_1 = \boldsymbol{x}_1]$. We show that CI implies that $\psi^*$ is good for downstream tasks, which is not apriori clear.

**Lemma 3.1** (Approximation error). *If random variables $X_1, X_2, Y$ satisfy Assumption 3.1, and $\boldsymbol{A} \in \mathbb{R}^{\mathcal{Y} \times d_2}$ with $\boldsymbol{A}_{y,:} := \mathbb{E}[X_2|Y = y]$ has rank $k = |\mathcal{Y}|$. Then $f^* \equiv \boldsymbol{W}^* \psi^*$, i.e., $e_{apx}(\psi^*) = 0$.*

This tells us that although $f^*$ could be nonlinear in $\boldsymbol{x}_1$, it is guaranteed to be linear in $\psi^*(\boldsymbol{x}_1)$.

*Proof Sketch of Lemma 3.1.* Lemma is proved by law of total expectation:

$$
\psi^*(\cdot) := \mathbb{E}[X_2|X_1] = \mathbb{E}[\mathbb{E}[X_2|X_1, Y]|X_1] = \mathbb{E}[\mathbb{E}[X_2|Y]|X_1] \quad \text{(uses CI)}
$$
$$
= \sum_y P(Y = y|X_1)\mathbb{E}[X_2|Y = y] =: f(X_1)^\top \boldsymbol{A},
$$

where $f(x_1)_y = P(Y = y|X_1 = x_1)$, and $\boldsymbol{A} \in \mathbb{R}^{\mathcal{Y} \times d_2}$ satisfies $\boldsymbol{A}_{y,:} = \mathbb{E}[X_2|Y = y]$. One could see that through predicting $X_2$, due to the CI assumption, $\psi^*$ has implicitly encoded the information of $Y|X_1$. Finally due to the fact that matrix $\boldsymbol{A}$ is full rank, we get that $f^*$ is linear in $\psi^*$ as well. $\square$

We see that besides CI, another important property is $\mathbb{E}[X_2|Y]$ being rank $k$. This means $X_2$ is correlated with every instance of $Y$, and thus captures information of every prediction class. This is naturally a necessary assumption for $X_2$ to be a reasonable pretext task for predicting $Y$. Note that this assumption does not trivialize the problem and that even though $\psi$ is designed to predict $X_2$, it can still be a better representation than $X_2$ for downstream tasks. Note that $Y$ does not have to be linear in $X_2$ but is proven to be linear in $\psi$, since $\psi$ learns to ignore some information in $X_2$ that is irrelevant to $Y$. We provide this simple example for better understanding:

**Example 3.1.** *Let $Y \in \{-1, 1\}$ be binary labels, and $X_1, X_2$ be $2-$mixture Gaussian random variables with $X_1 \sim \mathcal{N}(Y\boldsymbol{\mu}_1, \mathbf{I})$, $X_2 \sim \mathcal{N}(Y\boldsymbol{\mu}_2, \mathbf{I})$. In this example, $X_1 \perp X_2 | Y$. Although $\mathbb{E}[Y|X_2]$ and $\mathbb{E}[Y|X_1]$ are not linear, $\mathbb{E}[Y|\psi]$ is linear: $\psi(\boldsymbol{x}_1) = P(Y = 1|X_1 = \boldsymbol{x}_1)\boldsymbol{\mu}_2 - P(Y = -1|X_1 = \boldsymbol{x}_1)\boldsymbol{\mu}_2$ and $f^*(\boldsymbol{x}_1) = P(Y = 1|X_1 = \boldsymbol{x}_1) - P(Y = -1|X_1 = \boldsymbol{x}_1) \equiv \boldsymbol{\mu}_2^T \psi(\boldsymbol{x}_1)/\|\boldsymbol{\mu}_2\|^2$.*

Given that $\psi^*$ is good for downstream, we now care about the sample complexity. We will need to assume that the representation has some nice concentration properties. We make an assumption about the whitened data $\psi^*(X_1)$ to ignore scaling factors.

**Assumption 3.2.** *We assume the whitened feature variable $U := \mathbf{\Sigma}_\psi^{-1/2} \psi(X_1)$ is a $\rho^2$-subgaussian random variable, where $\mathbf{\Sigma}_\psi = \mathbb{E}[\psi(X_1)\psi(X_1)^\top]$.*

We note that all bounded random variables satisfy sub-gaussian property.

**Theorem 3.2** (General conditional independence). *Fix a failure probability $\delta \in (0, 1)$, under the same assumption as Lemma 3.1 and Assumption 3.2 for $\psi^*$, if additionally $n \gg \rho^4(k + \log(1/\delta))$, then the excess risk of the learned predictor $\boldsymbol{x}_1 \to \hat{\boldsymbol{W}}^\top \psi^*(\boldsymbol{x}_1)$ on the downstream task satsifies*

$$\mathrm{ER}_{\psi^*}[\hat{\boldsymbol{W}}] \le \tilde{\mathcal{O}}\left(\tfrac{k}{n_2}\sigma^2\right)[3]$$

**Remark 3.1.** *This analysis assumes we could perfectly learn $\psi^* = \mathbb{E}[X_2|X_1]$ disregarding the number of samples in the SSL phase (unlabeled data is cheap to obtain). Here by sample complexity we refer to the labeled data $(X_1, Y)$. We defer the effect of imprecise representation $\psi$ in Section 4.*

### 3.2 Function class induced by feature maps.

Given feature map $\phi_1 : \mathcal{X}_1 \to \mathbb{R}^{D_1}$, we consider the function class $\mathcal{H}_1 = \{\psi : \mathcal{X}_1 \to \mathbb{R}^{d_2} | \exists \boldsymbol{B} \in \mathbb{R}^{d_2 \times D_1}, \psi(\boldsymbol{x}_1) = \boldsymbol{B}\phi_1(\boldsymbol{x}_1)\}$.

**Claim 3.3** (Closed form solution)**.** *The optimal function in $\mathcal{H}$ is $\psi^*(\boldsymbol{x}_1) = \boldsymbol{\Sigma}_{X_2\phi_1}\boldsymbol{\Sigma}_{\phi_1\phi_1}^{-1}\phi_1(\boldsymbol{x}_1)$, where $\boldsymbol{\Sigma}_{X_2\phi_1} := \boldsymbol{\Sigma}_{X_2\phi_1(X_1)}$ and $\boldsymbol{\Sigma}_{\phi_1\phi_1} := \boldsymbol{\Sigma}_{\phi_1(X_1)\phi_1(X_1)}$.*

We again show the benefit of CI, but only comparing the performance of $\psi^*$ to the original features $\phi_1$. Since $\psi^*$ is linear in $\phi_1$, it cannot have smaller approximation error than $\phi_1$. However CI will ensure that $\psi^*$ has the same approximation error as $\phi_1$ and enjoys better sample complexity.

**Lemma 3.4** (Approximation error)**.** *If Assumption 3.1 is satisfied, and if the matrix $\boldsymbol{A} \in \mathbb{R}^{\mathcal{Y} \times d_2}$ with $\boldsymbol{A}_{y,:} := \mathbb{E}[X_2|Y = \boldsymbol{y}]$ is of rank $k = |\mathcal{Y}|$. Then $e_{apx}(\psi^*) = e_{apx}(\phi_1)$.*

We additionally need an assumption on the residual $a(\boldsymbol{x}_1) := \mathbb{E}[Y|X_1 = \boldsymbol{x}_1] - \mathbb{E}^L[Y|\phi_1(\boldsymbol{x}_1)]$.

**Assumption 3.3.** *(Bounded approx. error; Condition 3 in [32])) We have almost surely*

$$\|\boldsymbol{\Sigma}_{\phi_1\phi_1}^{-1/2}\phi_1(X_1)a(X_1)^\top\|_F \le b_0\sqrt{k}$$

**Theorem 3.5.** *(CI with approximation error) Fix a failure probability $\delta \in (0,1)$, under the same assumption as Lemma 3.4, Assumption 3.2 for $\psi^*$ and Assumption 3.3, if $n_2 \gg \rho^4(k + \log(1/\delta))$, then the excess risk of the learned predictor $\boldsymbol{x}_1 \to \hat{\boldsymbol{W}}^\top\psi^*(\boldsymbol{x}_1)$ on the downstream task satisfies:*

$$\mathrm{ER}_{\psi^*}[\hat{\boldsymbol{W}}] \le e_{apx}(\phi_1) + \tilde{\mathcal{O}}\left(\tfrac{k}{n_2}\sigma^2\right).$$

Thus with SSL, the requirement of labels is reduced from complexity for $D_1$ to $\mathcal{O}(k)$.

## 4 Beyond conditional independence

In the previous section, we focused on the case where we have exact CI. A weaker but more realistic assumption is that $Y$ captures some portion of the dependence between $X_1$ and $X_2$ but not all. We quantify this notion of approximate ACI through a quantity $\epsilon_{\mathrm{CI}}^2$ (Definition 4.1), and show excess risk bounds for the representation learned from SSL[4]. In particular, the excess risk will have the form $\tilde{\mathcal{O}}\left(\tfrac{d_2}{n_2} + \epsilon_{\mathrm{CI}}^2 + \epsilon_{\mathrm{pre}}^2\right)$, which suggests that only $n_2 = \mathcal{O}(d_2)$ labeled samples will be required to get small error on downstream task, as long as approximate CI is satisfied ($\epsilon_{\mathrm{CI}}^2$ is small) and the pretext task is solved well enough ($\epsilon_{\mathrm{pre}}^2$ is small). This is in contrast to not doing SSL, where many more labeled samples will be required to learn a solve the downstream task that learns a complicated representation function from scratch. We now describe the SSL method on finite samples, followed by the definition of ACI which we use to discuss the main excess risk bound and its consequences.

**SSL with finite samples and general function space:** Let $\boldsymbol{X}_1^{\mathrm{pre}} = [\boldsymbol{x}_1^{(1,\mathrm{pre})}, \cdots, \boldsymbol{x}_1^{(n_1,\mathrm{pre})}]^\top \in \mathbb{R}^{n_1 \times d_1}$ and $\boldsymbol{X}_2 = [\boldsymbol{x}_2^{(1)}, \cdots, \boldsymbol{x}_2^{(n_1)}]^\top \in \mathbb{R}^{n_1 \times d_2}$ be $n_1$ training samples for pretext task, where $(\boldsymbol{x}_1^{(i,\mathrm{pre})}, \boldsymbol{x}_2^{(i)})$ is sampled from $P_{X_1 X_2}$. The $n_2$ labeled samples for the downstream task are defined as $\boldsymbol{X}_1^{\mathrm{down}} \in \mathbb{R}^{n_2 \times d_1}$, $\boldsymbol{Y} \in \mathbb{R}^{n_2 \times d_3}$[5]. Given a representation function space $\mathcal{H} : \mathcal{X}_1 \to \mathbb{R}^{d_2}$, we learn $\tilde{\psi}$ from $\mathcal{H}$ using the $n_1$ unlabeled samples and then use the $n_2$ labeled samples to learn a linear classifier on the learned representation $\tilde{\psi}(\boldsymbol{X}_1^{\mathrm{down}})$ to fit $\boldsymbol{Y}$. This process is summarized below.

$$1)\ \tilde{\psi} := \underset{f \in \mathcal{H}}{\arg\min}\ \frac{1}{n_1}\|\boldsymbol{X}_2 - f(\boldsymbol{X}_1^{\mathrm{pre}})\|_F^2,\ 2)\ \hat{\boldsymbol{W}} \leftarrow \underset{\boldsymbol{W}}{\arg\min}\ \frac{1}{2n_2}\|\boldsymbol{Y} - \tilde{\psi}(\boldsymbol{X}_1^{\mathrm{down}})\boldsymbol{W}\|_F^2. \quad (2)$$

---

[3] We will use $\tilde{O}$ to hide log factor $\log(k/\delta)$ or $\log(d_2/\delta)$.

[4] Results for jointly-Gaussian variables is in Appendix D.1; ACI is quantified by the partial covariance matrix.

[5] $d_3 = k$ and $Y \equiv \phi_y(Y)$ (one-hot encoding) refers multi-class classification task, $d_3 = 1$ refers to regression.

In our main results, we consider two types of function spaces: $\mathcal{H} \in \{\mathcal{H}_1, \mathcal{H}_u\}$. Recall that $\mathcal{H}_1 = \{\psi(\cdot) = \boldsymbol{B}\phi_1(\cdot); \boldsymbol{B} \in \mathbb{R}^{d_2 \times D_1}\}$ is a class of *linear representations* induced by feature map $\phi_1 : \mathcal{X}_1 \to \mathbb{R}^{D_1}$. We use $\mathcal{H}_u$ to denote a function space with universal approximation power (e.g. deep networks) that ensures $\psi^* = \mathbb{E}[X_2|X_1] \in \mathcal{H}_u$. We define the optimal predictor in each case as $f_{\mathcal{H}}^*(X_1) = \mathbb{E}^L[Y|\phi_1(X_1)]$ when $\mathcal{H} = \mathcal{H}_1$, $f_{\mathcal{H}}^* = f^*$ for $\mathcal{H} = \mathcal{H}_u$, we define excess risk as

$$\mathrm{ER}_{\tilde{\psi}}(\hat{\boldsymbol{W}}) := \mathbb{E}_{X_1}\left[\|f_{\mathcal{H}}^*(X_1) - \hat{\boldsymbol{W}}^\top \tilde{\psi}(X_1)\|_2^2\right].$$

**Approximate conditional independence:** Our new assumption will generalize Assumption 3.1 in two ways, 1) we allow for additional latent variables $Z$ that together with $Y$ could potentially make $X_1$ and $X_2$ independent, and 2) we allow this conditional independence to be approximate. Note that allowing for extra latent variable can trivially make $X_1$ and $X_2$ to be conditionally independent by picking a large enough $Z$ (e.g. $Z = (X_1, X_2)$). However the following assumption, that needs the pretext target $X_2$ to correlate with all instances of variable $\bar{Y} = [Y, Z]$ (analogous to Lemma 3.1), will impose this restriction on how large $Z$ can be.

**Assumption 4.1** (Correlation between $X_2$ and $Y, Z$). *Suppose there exists latent variable $Z \in \mathcal{Z}, |\mathcal{Z}| = m$ that ensures $\boldsymbol{\Sigma}_{\phi_{\bar{y}} X_2}$ is full column rank and $\|\boldsymbol{\Sigma}_{Y\phi_{\bar{y}}} \boldsymbol{\Sigma}_{X_2 \phi_{\bar{y}}}^\dagger\|_2 = 1/\beta$, where $A^\dagger$ is pseudo-inverse, and $\phi_{\bar{y}}$ is the one-hot embedding for $\bar{Y} = [Y, Z]$.*

Just as in Section 3, this assumption will not assume away the problem (Example 3.1 can be suitably extended). The additional term $1/\beta$ here captures both the "scale" of $X_2$ and also the strength of correlation between $X_2$ and $[Y, Z]$ that was discussed after Lemma 3.1. For $\boldsymbol{\Sigma}_{\phi_{\bar{y}} X_2}$ to be full column rank, it is essential that $d_2 \geq km$, and this already gives an upper bound on the size of $Z$. Given this restriction on $Z$ (and thus $\bar{Y}$), we define the notion of approximate conditional independence.

**Definition 4.1** (Approximate conditional independence with function space $\mathcal{H}$). For $\bar{Y} = [Y, Z]$,
1. For $\mathcal{H} = \mathcal{H}_1$, define $\epsilon_{CI} := \|\boldsymbol{\Sigma}_{\phi_1 \phi_1}^{-1/2} \boldsymbol{\Sigma}_{\phi_1 X_2 | \phi_{\bar{y}}}\|_F$.
2. For $\mathcal{H} = \mathcal{H}_u$, define $\epsilon_{CI}^2 := \mathbb{E}_{X_1}[\|\mathbb{E}[X_2|X_1] - \mathbb{E}_{\bar{Y}}[\mathbb{E}[X_2|\bar{Y}]|X_1]\|^2]$.

Firstly we note that this is indeed an extension of exact CI, since exact CI in both cases will imply that $\epsilon_{CI} = 0$. We present a unified analysis in the appendix that shows the $\epsilon_{CI}$ for the second case is same as the first case, with covariance operators instead of matrices (A direct derivation is in Claim D.7). We also present more relaxed and general form of the above assumptions in Appendix F.1. With this assumption, we are ready to present our main bound.

**Bound on excess risk:** Recall that we assume that the residual term $N := Y - \mathbb{E}[Y|X_1]$ is mean zero and $\sigma^2$-subgaussian. Before showing our main result, analogous to Assumption 3.3, for the class $\mathcal{H}_1$ with non-universal features $\phi_1$, we will need an assumption[6] on the residual $a := f^* - f_{\mathcal{H}_1}^* = \mathbb{E}[Y|X_1] - \mathbb{E}^L[Y|\phi_1(X_1)]$:

**Assumption 4.2.** *(Bounded approximation error on pretext phase [32]) There exists a universal constant $b_0$, such that $\|\boldsymbol{\Sigma}_{\phi_1 \phi_1}^{-1/2} \phi_1(X_1) a(X_1)^\top\|_F \leq b_0\sqrt{d_2}$ almost surely.*

**Theorem 4.2.** *For a fixed $\delta \in (0, 1)$, under Assumptions 4.1,3.2 for $\tilde{\psi}$ and $\psi^*$ and 4.2 for non-universal feature maps, if $n_1, n_2 \gg \rho^4(d_2 + \log 1/\delta)$, and we learn the pretext tasks such that: $\mathbb{E}\|\tilde{\psi}(X_1) - \psi^*(X_1)\|_F^2 \leq \epsilon_{pre}^2$. Then the generalization error for downstream task w.p. $1 - \delta$ is:*

$$\mathrm{ER}_{\tilde{\psi}}(\hat{\boldsymbol{W}}) \leq \tilde{\mathcal{O}}\left( \underbrace{\sigma^2 \frac{d_2}{n_2}}_{\text{estimation error}} + \underbrace{\frac{\epsilon_{CI}^2}{\beta^2} + \frac{\epsilon_{pre}^2}{\beta^2}}_{\text{approximation error}} \right) \tag{3}$$

We defer the proof to the appendix. The proof technique is similar to that of Section 3. The difference is that now $\tilde{\psi}(\boldsymbol{X}^{(\text{down})}) \in \mathbb{R}^{n_2 \times d_2}$ will be an approximately low rank matrix, where the low rank part is the high-signal features that implicitly comes from $Y, Z$ that can linearly learn downstream task. The remaining part comes from $\epsilon_{CI}$ and $\epsilon_{pre}$ and causes the approximation error. Again by selecting the top $km$ (dimension of $\phi_{\bar{y}}$) features we could further improve the bound:

---

[6]This rules out the failure if one chooses a very simple function class to learn $\mathbb{E}[X_2|X_1]$. In practice we usually use neural networks (with universal approximation power) and this bound should be very small.

**Remark 4.1.** *By applying PCA on $\tilde{\psi}(\boldsymbol{X}_1^{down})$ and keeping the top $km$ principal components only, we can improve the bound in Theorem 4.2 to* $\mathrm{ER}_{\tilde{\psi}}(\hat{\boldsymbol{W}}) \leq \tilde{\mathcal{O}}\left(\sigma^2 \frac{km}{n_2} + \frac{\epsilon_{CI}^2}{\beta^2} + \frac{\epsilon_{pre}^2}{\beta^2}\right)$.

We take a closer look at the different sources of errors in Lemma 4.1: 1) The first term is estimation error on learning with finite samples $n_2$ with noise level $\sigma^2$ in $Y - f^*(X_1)$; 2) $\epsilon_{\mathrm{CI}}$ measures the approximate CI; and 3) $\epsilon_{\mathrm{pre}}$ is the error from not learning the pretext task exactly. The first term is optimal ignoring log factors as we do linear regression on $mk$-dimensional features. The second and third term together form approximation error. They are non-reducible due to the fact that $f^*$ is not exactly linear in $\psi$ and we use it as a fixed representation. Fine-tuning the representations might be necessary to get rid of these terms when we have sufficient downstream labeled data. We leave this exploring this as future work. Compared to traditional supervised learning, learning $f_{\mathcal{H}}^*$ requires sample complexity scaling with the (Rademacher/Gaussian) complexity of $\mathcal{H}$ (see e.g. [8, 52]), which is very large for complicated models such as deep networks. Thus SSL can significantly reduce the labeled sample complexity down from this complexity measure of $\mathcal{H}$ to $\tilde{\mathcal{O}}(km)$, demonstrating the power of predicting what you already know using unlabeled data. In Section H, we consider a similar result for classification.

## 5  Example: Topic Modeling

In this section, we will demonstrate how our framework can be instantiated for standard data model like topic modeling. Topic modeling for text that has a rich literature [47, 31, 9, 4, 3] and is used for analyzing and designing algorithms for information retrieval, dimensionality reduction and data analysis for large text corpora. We describe the basic setup below, followed by how our results for reconstruction-based SSL can be instantiated to learn such models.

For a set $S$, let $\Delta_S$ denote the set of all distributions on $S$. In the topic modeling framework, generation of a text document with a vocabulary set $[V] = \{1, \ldots, V\}$ is governed by certain latent topics from the set $[k]$, where $k$ is the total number of topics. Each topic $i \in [k]$ is associated with a distribution over the vocabulary $[V]$ that is denoted by vector $A_i \in \Delta_{[V]}$; stack these vectors into the columns of a matrix $A \in \mathbb{R}^{V \times k}$. A document $X = (x_1, \ldots, x_n) \in [V]^N$ of length $N$ is then sampled from a mixture of the $k$ topics $\mu \in \Delta_{[k]}$. The generative process is described below:

1. Sample a topic mixture $\mu \sim \tau$. $\tau$ is some underlying distribution over $\Delta_k$, i.e. $\tau \in \Delta_{\Delta_{[k]}}$
2. For each $i \in [N]$, sample a topic $t_i \sim \mu$ and sample a word $x_i \sim A_{t_i}$ from the topic

For the reconstruction SSL task, we evenly split the document as $X = (\bar{X}_1, \bar{X}_2)$, where $\bar{X}_1$ and $\bar{X}_2$ denote the first and second halves of the document; note that $\bar{X}_1, \bar{X}_2 \in [V]^{N/2}$. We let $X_1$ and $X_2$ be the multiset of words in the two halves by using the normalized bag-of-words representation, i.e. $X_i = \frac{2}{N}\text{bag-of-words}(\bar{X}_i) \in \mathbb{R}^V$, $i \in \{1, 2\}$[7]. The downstream task is chosen to be a linear function of the topic posterior distribution $\mu$ for a given document $X$, i.e. $Y = w^\top \mathbb{E}[\mu|X] + N$, where $N$ is 0 mean and $\sigma^2$-subgaussian. The error of a predictor $f : [V]^N \to \mathbb{R}$ is measured as $\mathbb{E}_{\mu,X}\left[\left(f(X) - \mu^\top w\right)^2\right]$, the optimal predictor being $f^*(X) = \mathbb{E}[Y \mid X]$.

A crucial property of topic model described above is that words in the document are sampled independently given the topic mixture $\mu$, thus giving us the property: $X_1 \perp X_2 \mid \mu$. Although the cardinality of $\mu \in \Delta_{[k]}$ (that implicitly shows up in Theorem 4.2) is infinite, we can still show the benefit of SSL using our theoretical framework. We will show appropriate bounds for $\epsilon_{\mathrm{CI}}$ and $\beta$, that show up in Theorem 4.2, using the topic model generative process. We make the following standard assumptions about the topic modeling distribution, motivated by prior work [4, 3].

**Assumption 5.1.** *Let $A \in \mathbb{R}^{V \times k}$ be the word-topic matrix and $\Gamma = \mathbb{E}_{\mu \sim \tau}\left[\mu\mu^\top\right]$ be the topic covariance matrix, then the following hold*

- *(Anchor word) The word-topic matrix $A$ is $p$-separable for $p > 0$, i.e. for every topic $i \in [k]$ there is a word $j$ such that $A_i(j) \geq p$ and $A_{i'}(j) = 0$ when $i' \neq i$*
- *$\Gamma$ is full rank, so condition number $\kappa = \frac{\lambda_{\max}(\Gamma)}{\lambda_{\min}(\Gamma)} < \infty$*

---

[7]We only need $X_2$ to be the bag-of-word representation, $X_1$ can be an ordered sentence.

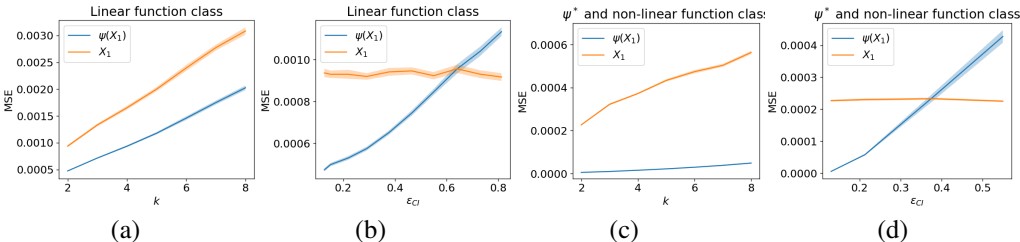

Figure 1: **Left two**: how MSE scales with $k$ (the dimension of $Y$) and $\epsilon_{CI}$ (Definition 4.1) with the linear function class. **Right two**: how MSE scales with $k$ and $\epsilon$ with $\psi^*$ and non-linear function class. Mean of 30 trials are shown in solid line and one standard error is shown by shadow.

**Theorem 5.1.** *Let $\epsilon_{CI}$ be the definition (2) from Definition 4.1 and $\beta$ as defined in Assumption 4.1 and suppose the topic model satisfies Assumption 5.1, then there exists a latent variable $\bar{Y} \in \bar{\mathcal{Y}}$ such that the following hold*

1. *$\bar{Y}$ takes $k$ distinct values, i.e. $|\bar{\mathcal{Y}}| = k$*
2. *$X_1$ and $X_1$ are uncorrelated given $\bar{Y}$, which implies $\boldsymbol{\epsilon_{CI} = 0}$.*
3. *$\mathbb{E}[Y|X_1]$ is a linear function of $\mathbb{E}[\bar{Y}|X_1]$*
4. *$\boldsymbol{\beta^{-1} \leq \kappa \|w\|_2 / \lambda_{\min}(A) \leq \kappa \|w\|_2 / p}$*

The proof for this is presented in Section E.1. Note that the $p$-separability is not necessarily needed, and the bound with $\lambda_{\min}(A)$ can be invoked instead. Thus the upper bound from Theorem 4.2) will look like $\tilde{\mathcal{O}}\left(\sigma^2 \frac{k}{n_2} + \epsilon_{\text{pre}}^2 \frac{\kappa\|w\|_2}{p}\right)$, thus requiring only $\mathcal{O}(k)$ samples for the downstream task.

## 6 Experiments

In this section, we empirically verify our claim that SSL performs well when ACI is satisfied. More details for experiments can be found in Section J, including experiments in the text domain.

**Simulations.** With synthetic data, we verify how excess risk (ER) scales with the cardinality/feature dimension of $\mathcal{Y}$ ($k$), and ACI ($\epsilon_{CI}$ in Definition 4.1). We consider a mixture of Gaussian data and conduct experiments with both linear function space ($\mathcal{H}_1$ with $\phi_1$ as identity map) and universal function space $\mathcal{H}_u$. We sample the label $Y$ uniformly from $\{1, ..., k\}$. For $i$-th class, the centers $\mu_{1i} \in \mathbb{R}^{d_1}$ and $\mu_{2i} \in \mathbb{R}^{d_2}$ are uniformly sampled from $[0, 10]$. Given $Y = i$, $\alpha \in [0, 1]$, let $X_1 \sim \mathcal{N}(\mu_{1i}, \mathbf{I})$, $\hat{X}_2 \sim \mathcal{N}(\mu_{2i}, \mathbf{I})$, and $X_2 = (1 - \alpha)\hat{X}_2 + \alpha X_1$. Therefore $\alpha$ is a correlation coefficient: $\alpha = 0$ ensures $X_2$ being CI with $X_1$ given $Y$ and when $\alpha = 1$, $X_2$ fully depends on $X_1$. (if $d_1 \neq d_2$, we append zeros or truncate to fit accordingly).

We first conduct experiments with linear function class. We learn a linear representation $\psi$ with $n_1$ samples and the linear prediction of $Y$ from $\psi$ with $n_2$ samples. We set $d_1 = 50$, $d_2 = 40$, $n_1 = 4000$, $n_2 = 1000$ and ER is measured with Mean Squared Error (MSE). As shown in Figure 1(a)(b), the MSE of learning with $\psi(X_1)$ scales linearly with $k$ as indicated in Theorem 3.5, and scales linearly with $\epsilon_{CI}$ associated with linear function class as indicated in Theorem 4.2. Next we move on to general function class, i.e., $\psi^* = \mathbb{E}[Y|X_1]$ with a closed form solution (see example 3.1). We use the same parameter settings as above. For baseline method, we use kernel linear regression to predict $Y$ using $X_1$ (we use RBF kernel which also has universal approximation power). As shown in Figure 1(c)(d), the phenomenon is the same as what we observe in the linear function class setting, and hence they respectively verify Theorem 3.2 and Theorem 4.2 with $\mathcal{H}_u$.

**Computer Vision Task.** We verify if learning from $\psi$ is more effective than learning directly from $X_1$, in a realistic setting (without enforcing conditional independence). Specifically, we test on the Yearbook dataset [23], and try to predict the date when the portraits are taken (denoted as $Y_D$), which ranges from 1905 to 2013. We resize all the portraits to be 128 by 128. We crop out the center 64 by 64 pixels (the face), and treat it as $X_2$, and treat the outer rim as $X_1$ as shown in Figure 2. Our task is to predict $Y_D$, which is the year when the portraits are taken, and the year ranges from 1905 to 2013. For $\psi$, we learn $X_2$ from $X_1$ with standard image inpainting techniques [48], and full set of training data (without labels). After that we fix the learned $\psi$ and learn a linear model to predict $Y_D$ from $\psi$ using a smaller set of data (with labels). Besides linear model on $X_1$, another strong baseline that

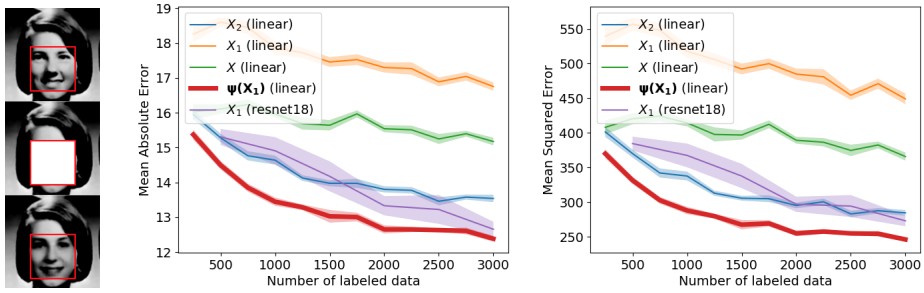

Figure 2: **Left**: Example of the $X_2$ (in the red box of the 1st row), the $X_1$ (out of the red box of the 1st row), the input to the inpainting task (the second row), $\psi(X_1)$ (the 3 row in the red box), and in this example $Y = 1967$. **Middle**: Mean Squared Error comparison of yearbook regression predicting dates. **Right**: Mean Absolute Error comparison of yearbook regression predicting dates. Experiments are repeated 10 times, with mean shown in solid line and one standard deviation in shadow.

we compare with is using ResNet18 [28] to predict $Y_D$ from $X_1$. With the full set of training data, this model is able to achieve a Mean Absolute Difference of 6.89, close to what state-of-the-art can achieve [23]. ResNet18 has similar amount of parameters as our generator, and hence roughly in the same function class. We show the MSE result as in Figure 2. Learning from $\psi$ is more effective than learning from $X_1$ or $X_2$ directly, with linear model as well as with ResNet18. Practitioner usually fine-tune $\psi$ with the downstream task, which leads to more competitive performance [48].

# 7 Conclusion

In this work we theoretically quantify how an approximate conditional independence assumption that connects pretext and downstream task data distributions can give sample complexity benefits of self-supervised learning on downstream tasks. Our theoretical findings are also supported by experiments on simulated data and also on real CV and NLP tasks. We would like to note that approximate CI is only a sufficient condition for a useful pretext task. We leave it for future work to investigate other mechanisms by which pretext tasks help with downstream tasks.

## Acknowledgment

JDL acknowledges support of the ARO under MURI Award W911NF-11-1-0304, the Sloan Research Fellowship, NSF CCF 2002272, NSF IIS 2107304, and an ONR Young Investigator Award. QL was supported by NSF #2030859 and the Computing Research Association for the CIFellows Project. NS is supported by NSF, ONR, Simons Foundation, DARPA and SRC. JZ acknowledges sponsorship of Army Research Office and support of Cooperative Agreement Number W911NF-19-2-0333 and NSF grant 2019844. The views and conclusions contained in this document are those of the authors and should not be interpreted as representing the official policies, either expressed or implied, of the Army Research Office or the U.S. Government. The U.S. Government is authorized to reproduce and distribute reprints for Government purposes notwithstanding any copyright notation herein.

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
