# A  Some Useful Facts

## A.1  Relation of Inverse Covariance Matrix and Partial Correlation

For a covariance matrix of joint distribution for variables $X, Y$, the covariance matrix is

$$\begin{bmatrix} \boldsymbol{\Sigma}_{XX} & \boldsymbol{\Sigma}_{XY} \\ \boldsymbol{\Sigma}_{YX} & \boldsymbol{\Sigma}_{YY} \end{bmatrix} = \begin{bmatrix} \boldsymbol{\Sigma}_{X_1 X_1} & \boldsymbol{\Sigma}_{X_1 X_2} & \boldsymbol{\Sigma}_{X_1 Y} \\ \boldsymbol{\Sigma}_{X_2 X_1} & \boldsymbol{\Sigma}_{X_2 X_2} & \boldsymbol{\Sigma}_{X_2 Y} \\ \boldsymbol{\Sigma}_{YX_1} & \boldsymbol{\Sigma}_{X_2 Y} & \boldsymbol{\Sigma}_{YY} \end{bmatrix}.$$

Its inverse matrix $\boldsymbol{\Sigma}^{-1}$ satisfies

$$\boldsymbol{\Sigma}^{-1} = \begin{bmatrix} \boldsymbol{A} & \rho \\ \rho^\top & \boldsymbol{B} \end{bmatrix}.$$

Here $\boldsymbol{A}^{-1} = \boldsymbol{\Sigma}_{XX} - \boldsymbol{\Sigma}_{XY} \boldsymbol{\Sigma}_{YY}^{-1} \boldsymbol{\Sigma}_{YX} \equiv \mathrm{cov}(X - \mathbb{E}^L[X|Y], X - \mathbb{E}^L[X|Y]) := \boldsymbol{\Sigma}_{XX \cdot Y}$, the partial covariance matrix of $X$ given $Y$.

## A.2  Relation to Conditional Independence

*Proof of Lemma D.4.*

**Fact A.1.** *When $X_1 \perp X_2 | Y$, the partial covariance between $X_1, X_2$ given $Y$ is $0$:*

$$\begin{aligned} \boldsymbol{\Sigma}_{X_1 X_2 \cdot Y} :=& \mathrm{cov}(X_1 - \mathbb{E}^L[X_1|Y], X_2 - \mathbb{E}^L[X_2|Y]) \\ \equiv& \boldsymbol{\Sigma}_{X_1 X_2} - \boldsymbol{\Sigma}_{X_1 Y} \boldsymbol{\Sigma}_{YY}^{-1} \boldsymbol{\Sigma}_{YX_2} = 0. \end{aligned}$$

The derivation comes from the following:

**Lemma A.1** (Conditional independence (Adapted from [34]))**.** *For random variables $X_1, X_2$ and a random variable $Y$ with finite values, conditional independence $X_1 \perp X_2 | Y$ is equivalent to:*

$$\sup_{f \in N_1, g \in N_2} \mathbb{E}[f(X_1) g(X_2) | Y] = 0. \tag{4}$$

*Here $N_i = \{f : \mathbb{R}^{d_i} \to R : E[f(X_i)|Y] = 0\}$, $i = 1, 2$.*

Notice for arbitrary function $f$, $\mathbb{E}[f(X)|Y] = \mathbb{E}^L[f(X)|\phi_y(Y)]$ with one-hot encoding of discrete variable $Y$. Therefore for any feature map we can also get that conditional independence ensures:

$$\begin{aligned} \boldsymbol{\Sigma}_{\phi_1(X_1)\phi_2(X_2)|Y} :=& \mathrm{cov}(\phi_1(X_1) - \mathbb{E}^L[\phi_1(X_1)|\phi_y(Y)], \phi_2(X_2) - \mathbb{E}^L[\phi_2(X_2)|\phi_y(Y)]) \\ =& \mathbb{E}[\bar{\phi}_1(X_1)\bar{\phi}_2(X_2)^\top] = 0. \end{aligned}$$

Here $\bar{\phi}_1(X_1) = \phi_1(X_1) - \mathbb{E}[\phi_1(X_1)|\phi_y(Y)]$ is mean zero given $Y$, and vice versa for $\bar{\phi}_2(X_2)$. This thus finishes the proof for Lemma D.4. $\qquad\square$

## A.3  Technical Facts for Matrix Concentration

We include this covariance concentration result that is adapted from Claim A.2 in [18]:

**Claim A.2** (covariance concentration for gaussian variables)**.** *Let $\boldsymbol{X} = [\boldsymbol{x}_1, \boldsymbol{x}_2, \cdots \boldsymbol{x}_n]^\top \in \mathbb{R}^{n \times d}$ where each $x_i \sim \mathcal{N}(0, \boldsymbol{\Sigma}_X)$. Suppose $n \gg k + \log(1/\delta)$ for $\delta \in (0, 1)$. Then for any given matrix $B \in \mathbb{R}^{d \times m}$ that is of rank $k$ and is independent of $\boldsymbol{X}$, with probability at least $1 - \frac{\delta}{10}$ over $\boldsymbol{X}$ we have*

$$0.9 \boldsymbol{B}^\top \boldsymbol{\Sigma}_X \boldsymbol{B} \preceq \frac{1}{n} \boldsymbol{B}^\top \boldsymbol{X}^\top \boldsymbol{X} \boldsymbol{B} \preceq 1.1 \boldsymbol{B}^\top \boldsymbol{\Sigma}_X \boldsymbol{B}. \tag{5}$$

And we will also use Claim A.2 from [18] for concentrating subgaussian random variable.

**Claim A.3** (covariance concentration for subgaussian variables)**.** *Let $\boldsymbol{X} = [\boldsymbol{x}_1, \boldsymbol{x}_2, \cdots \boldsymbol{x}_n]^\top \in \mathbb{R}^{n \times d}$ where each $\boldsymbol{x}_i$ is $\rho^2$-sub-gaussian. Suppose $n \gg \rho^4(k + \log(1/\delta))$ for $\delta \in (0, 1)$. Then for any given matrix $B \in \mathbb{R}^{d \times m}$ that is of rank $k$ and is independent of $\boldsymbol{X}$, with probability at least $1 - \frac{\delta}{10}$ over $\boldsymbol{X}$ we have*

$$0.9 \boldsymbol{B}^\top \boldsymbol{\Sigma}_X \boldsymbol{B} \preceq \frac{1}{n} \boldsymbol{B}^\top \boldsymbol{X}^\top \boldsymbol{X} \boldsymbol{B} \preceq 1.1 \boldsymbol{B}^\top \boldsymbol{\Sigma}_X \boldsymbol{B}. \tag{6}$$

**Claim A.4.** *Let $\boldsymbol{Z} \in \mathbb{R}^{n \times k}$ be a matrix with row vectors sampled from i.i.d Gaussian distribution $\mathcal{N}(0, \boldsymbol{\Sigma}_Z)$. Let $\mathbf{P} \in \mathbb{R}^{n \times n}$ be a fixed projection onto a space of dimension $d$. Then with a fixed $\delta \in (0, 1)$, we have:*

$$\|\mathbf{P}\boldsymbol{Z}\|_F^2 \lesssim \mathrm{Tr}(\boldsymbol{\Sigma}_Z)(d + \log(k/\delta)),$$

*with probability at least $1 - \delta$.*

*Claim A.4.* Each $t$-th column of $Z$ is an $n$-dim vector that is i.i.d sampled from Gaussian distribution $\mathcal{N}(0, \boldsymbol{\Sigma}_{tt})$.

$$\|\mathbf{P}\boldsymbol{Z}\|_F^2 = \sum_{t=1}^{k} \|\mathbf{P}\boldsymbol{z}_t\|^2$$
$$= \sum_{t=1}^{k} \boldsymbol{z}_t^\top \mathbf{P}\boldsymbol{z}_t.$$

Each term satisfy $\boldsymbol{\Sigma}_{kk}^{-1}\|\mathbf{P}\boldsymbol{z}_t\|^2 \sim \chi^2(d)$, and therefore with probability at least $1 - \delta'$ over $\boldsymbol{z}_t$,

$$\boldsymbol{\Sigma}_{kk}^{-1}\|\mathbf{P}\boldsymbol{z}_t\|^2 \lesssim d + \log(1/\delta').$$

Using union bound, take $\delta' = \delta/k$ and summing over $t \in [k]$ we get:

$$\|\mathbf{P}\boldsymbol{Z}\|_F^2 \lesssim \mathrm{Tr}(\boldsymbol{\Sigma}_Z)(d + \log(k/\delta)).$$

$\square$

**Theorem A.5** (Vector Bernstein Inequality (Theorem 12 in [26]))**.** *Let $X_1, \cdots, X_m$ be independent zero-mean vector-valued random variables. Let*

$$N = \|\sum_{i=1}^{m} X_i\|_2.$$

*Then*

$$\mathbb{P}[N \geq \sqrt{V} + t] \leq \exp\left(\frac{-t^2}{4V}\right),$$

*where $V = \sum_i \mathbb{E}\|X_i\|_2^2$ and $t \leq V/(\max \|X_i\|_2)$.*

**Lemma A.6.** *Let $\boldsymbol{Z} \in \mathbb{R}^{n \times d}$ be a matrix whose row vectors are $n$ independent mean-zero (conditional on $\mathbf{P}$ being a rank-$d$ projection matrix) $\sigma$-sub-Gaussian random vectors. With probability $1 - \delta$:*

$$\|\mathbf{P}\boldsymbol{Z}\|_F^2 \lesssim \sigma^2(d + \log(d/\delta)).$$

*Proof of Lemma A.6.* Write $\mathbf{P} = \boldsymbol{U}\boldsymbol{U}^\top = [\boldsymbol{u}_1, \cdots, \boldsymbol{u}_d]$ where $\boldsymbol{U}$ is orthogonal matrix in $\mathbb{R}^{n \times d}$ where $\boldsymbol{U}^\top \boldsymbol{U} = I$. Notice $\|\boldsymbol{U}\boldsymbol{U}^\top \boldsymbol{Z}\|_F^2 = \mathrm{Tr}(\boldsymbol{Z}^\top \boldsymbol{U}\boldsymbol{U}^\top \boldsymbol{U}\boldsymbol{U}^\top \boldsymbol{Z}) = \mathrm{Tr}(\boldsymbol{Z}^\top \boldsymbol{U}\boldsymbol{U}^\top \boldsymbol{Z})$. Therefore:

$$\|\mathbf{P}\boldsymbol{Z}\|_F^2 = \|\boldsymbol{U}^\top \boldsymbol{Z}\|_F^2$$
$$= \sum_{j=1}^{d} \|\boldsymbol{u}_j^\top \boldsymbol{Z}\|^2$$
$$= \sum_{j=1}^{d} \|\sum_{i=1}^{n} \boldsymbol{u}_{ji}\boldsymbol{z}_i\|^2,$$

where each $\boldsymbol{z}_i \in \mathbb{R}^k$ being the $i$-th row of $\boldsymbol{Z}$ is a centered independent $\sigma$ sub-Gaussian random vectors. To use vector Bernstein inequality, we let $X := \sum_{i=1}^{n} X_i$ with $X_i$ taking the value of $\boldsymbol{u}_{ji}\boldsymbol{z}_i$.

We have $X_i$ is zero mean: $\mathbb{E}[X_i] = \mathbb{E}[\boldsymbol{u}_{ji}\mathbb{E}[\boldsymbol{z}_i|\boldsymbol{u}_{ji}]] = \mathbb{E}[\boldsymbol{u}_{ji} \cdot 0] = 0$.

$$
\begin{aligned}
V &:= \sum_i \mathbb{E}\|X_i\|_2^2 \\
&= \sum_i \mathbb{E}[\boldsymbol{u}_{ji}^2 \boldsymbol{z}_i^\top \boldsymbol{z}_i] \\
&= \sum_i \mathbb{E}_{\boldsymbol{u}_{ji}}[\boldsymbol{u}_{ji}^2 \mathbb{E}[\|\boldsymbol{z}_i\|_2^2|\boldsymbol{u}_{ji}]] \\
&\leq \sigma^2 \sum_i \mathbb{E}_{\boldsymbol{u}_{ji}}[\boldsymbol{u}_{ji}^2] \\
&= \sigma^2.
\end{aligned}
$$

Therefore by vector Bernstein Inequality, with probability at least $1 - \delta/d$, $\|X\| \leq \sigma(1 + \sqrt{\log(d/\delta)})$. Then by taking union bound, we get that $\|\boldsymbol{P}\boldsymbol{Z}\|^2 = \sum_{j=1}^d \|\boldsymbol{u}_j^\top \boldsymbol{Z}\|^2 \lesssim \sigma^2 d(1 + \log(d/\delta))$ with probability $1 - \delta$.

$\square$

# B  Warm-up: jointly Gaussian variables

We assume $X_1, X_2, Y$ are jointly Gaussian, and so the optimal regression functions are all linear, i.e., $\mathbb{E}[Y|X_1] = \mathbb{E}^L[Y|X_1]$. We also assume data is centered: $\mathbb{E}[X_i] = 0$ and $\mathbb{E}[Y] = 0$. Non-centered data can easily be handled by learning an intercept. All relationships between random variables can then be captured by the (partial) covariance matrix. Therefore it is easy to quantify the CI property and establish the necessary and sufficient conditions that make $X_2$ a reasonable pretext task.

**Assumption B.1** (Jointly Gaussian). *$X_1, X_2, Y$ are jointly Gaussian.*

**Assumption B.2** (Conditional independence). *$X_1 \perp X_2 | Y$.*

**Claim B.1** (Closed-form solution). *Under Assumption B.1, the representation function and optimal prediction that minimize the population risk can be expressed as follows:*

$$\psi^*(\boldsymbol{x}_1) := \mathbb{E}^L[X_2|X_1 = \boldsymbol{x}_1] = \boldsymbol{\Sigma}_{X_2 X_1} \boldsymbol{\Sigma}_{X_1 X_1}^{-1} \boldsymbol{x}_1 \tag{7}$$

$$\textit{Our target } f^*(\boldsymbol{x}_1) := \mathbb{E}^L[Y|X_1 = \boldsymbol{x}_1] = \boldsymbol{\Sigma}_{Y X_1} \boldsymbol{\Sigma}_{X_1 X_1}^{-1} \boldsymbol{x}_1. \tag{8}$$

Our prediction for downstream task with representation $\psi^*$ will be: $g(\cdot) := \mathbb{E}^L[Y|\psi^*(X_1)]$. Recall from Equation (1) that the partial covariance matrix between $X_1$ and $X_2$ given $Y$ is $\boldsymbol{\Sigma}_{X_1 X_2|Y} \equiv \boldsymbol{\Sigma}_{X_1 X_2} - \boldsymbol{\Sigma}_{X_1 Y}\boldsymbol{\Sigma}_{YY}^{-1}\boldsymbol{\Sigma}_{Y X_2}$. This partial covariance matrix captures the correlation between $X_1$ and $X_2$ given $Y$. For jointly Gaussian random variables, CI is equivalent to $\boldsymbol{\Sigma}_{X_1 X_2|Y} = 0$. We first analyze the approximation error based on the property of this partial covariance matrix.

**Lemma B.2** (Approximation error). *Under Assumption B.1, B.2, if $\boldsymbol{\Sigma}_{X_2 Y}$ has rank $k$, we have $f^*(\boldsymbol{x}_1) \equiv \boldsymbol{W}^* \psi^*(\boldsymbol{x}_1)$, i.e., $e_{apx}(\psi^*) = 0$.*

**Remark B.1.** *$\boldsymbol{\Sigma}_{X_2 Y}$ being full column rank implies that $\mathbb{E}[X_2|Y]$ has rank $k$, i.e., $X_2$ depends on all directions of $Y$ and thus captures all directions of information of $Y$. This is a necessary assumption for $X_2$ to be a reasonable pretext task for predicting $Y$. $e_{apx}(\psi^*) = 0$ means $f^*$ is linear in $\psi^*$. Therefore $\psi^*$ selects $d_2$ out of $d_1$ features that are sufficient to predict $Y$.*

Next we consider the estimation error that characterizes the number of samples needed to learn a prediction function $f(\boldsymbol{x}_1) = \hat{\boldsymbol{W}}\psi^*(\boldsymbol{x}_1)$ that generalizes.

**Theorem B.3** (Excess risk). *Fix a failure probability $\delta \in (0, 1)$. Under Assumption B.1,B.2, if $n_2 \gg k + \log(1/\delta)$, excess risk of the learned predictor $\boldsymbol{x}_1 \to \hat{\boldsymbol{W}}\psi^*(\boldsymbol{x}_1)$ on the target task satisfies*

$$\mathrm{ER}_{\psi^*}(\hat{\boldsymbol{W}}) \leq \mathcal{O}\left(\frac{\mathrm{Tr}(\boldsymbol{\Sigma}_{YY|X_1})(k + \log(k/\delta))}{n_2}\right),$$

*with probability at least $1 - \delta$.*

Here $\Sigma_{YY|X_1} \equiv \Sigma_{YY} - \Sigma_{YX_1}\Sigma_{X_1X_1}^{-1}\Sigma_{X_1Y}$ captures the noise level and is the covariance matrix of the residual term $Y - f^*(X_1) = Y - \Sigma_{YX_1}\Sigma_{X_1X_1}^{-1}X_1$. Compared to directly using $X_1$ to predict $Y$, self-supervised learning reduces the sample complexity from $\tilde{\mathcal{O}}(d_1)$ to $\tilde{\mathcal{O}}(k)$. We generalize these results even when only a weaker form of CI holds.

**Assumption B.3** (Conditional independence given latent variables). *There exists some latent variable $Z \in \mathbb{R}^m$ such that $X_1 \perp X_2|\bar{Y}$, and $\Sigma_{X_2\bar{Y}}$ is of rank $k + m$, where $\bar{Y} = [Y, Z]$.*

This assumption lets introduce some reasonable latent variables that capture the information between $X_1$ and $X_2$ apart from $Y$. $\Sigma_{X_2\bar{Y}}$ being full rank says that all directions of $\bar{Y}$ are needed to predict $X_2$, and therefore $Z$ is not redundant. For instance, when $Z = X_1$, the assumption is trivially true but $Z$ is not the minimal latent information we want to add. Note it implicitly requires $d_2 \geq k + m$.

**Corollary B.4.** *Under Assumption B.1, B.3, we have $f^*(x_1) \equiv W^*\psi^*(x_1)$, i.e., the approximation error $e_{apx}(\psi^*)$ is 0. We can also generalize Theorem B.3 by replacing $k$ by $k + m$.*

## C    Omitted Proofs with Conditional Independence

*Proof of Lemma B.2.*

$$\mathrm{cov}(X_1|Y, X_2|Y) = \Sigma_{X_1X_2} - \Sigma_{X_1Y}\Sigma_{YY}^{-1}\Sigma_{YX_2} = 0.$$

By plugging it into the expression of $\mathbb{E}^L[X_2|X_1]$, we get that

$$\begin{aligned}
\psi(x_1) := \mathbb{E}^L[X_2|X_1 = x_1] &= \Sigma_{X_2X_1}\Sigma_{X_1X_1}^{-1}x_1 \\
&= \Sigma_{X_2Y}\Sigma_{YY}^{-1}\Sigma_{YX_1}\Sigma_{X_1X_1}^{-1}x_1 \\
&= \Sigma_{X_2Y}\Sigma_{YY}^{-1}\mathbb{E}^L[Y|X_1].
\end{aligned}$$

Therefore, as long as $\Sigma_{X_2Y}$ is rank $k$, it has left inverse matrix and we get: $\mathbb{E}^L[Y|X_1 = x_1] = \Sigma_{X_2Y}^\dagger\Sigma_{YY}\psi(x_1)$. Therefore there's no approximation error in using $\psi$ to predict $Y$.

$\square$

*Proof of Corollary B.4.* Let selector operator $S_y$ be the mapping such that $S_y\bar{Y} = Y$, we overload it as the matrix that ensure $S_y\Sigma_{\bar{Y}X} = \Sigma_{YX}$ for any random variable $X$ as well.

From Lemma B.2 we get that there exists $W$ such that $\mathbb{E}^L[\bar{Y}|X_1] = W\mathbb{E}^L[X_2|X_1]$, just plugging in $S_y$ we get that $\mathbb{E}^L[Y|X_1] = (S_yW)\mathbb{E}^L[X_2|X_1]$.

$\square$

*Proof of Theorem B.3.* Write $f^*(X_1) = \mathbb{E}[Y|X_1] = (A^*)^\top X_1$. $\mathbb{E}^L[Y|X_1 = x_1] = \Sigma_{X_2Y}^\dagger\Sigma_{YY}\psi(x_1)$. Let $W^* = \Sigma_{YY}\Sigma_{YX_2}^\dagger$. From Lemma B.2 we know $f^* = W^*\psi$. Recall noise $N = Y - f^*(X_1)$ is mean zero conditional on $X_1$. We write $N = Y - f^*(X_1)$.

First we have the basic inequality,

$$\begin{aligned}
\frac{1}{2n_2}\|Y - \psi(X_1)\hat{W}\|_F^2 &\leq \frac{1}{2n_2}\|Y - X_1A^*\|_F^2 \\
&= \frac{1}{2n_2}\|Y - \psi(X_1)W^*\|_F^2 = \frac{1}{2n_2}\|N\|_F^2.
\end{aligned}$$

Therefore by rearranging both sides, we have:

$$\begin{aligned}
\|\psi(X_1)W^* - \psi(X_1)\hat{W}\|^2 &\leq 2\langle N, \psi(X_1)W^* - \psi(X_1)\hat{W}\rangle \\
&= 2\langle P_{\psi(X_1)}N, \psi(X_1)W^* - \psi(X_1)\hat{W}\rangle \\
&\leq 2\|P_{\psi(X_1)}N\|_F\|\psi(X_1)W^* - \psi(X_1)\hat{W}\|_F \\
\Rightarrow \|\psi(X_1)W^* - \psi(X_1)\hat{W}\| &\leq 2\|P_{\psi(X_1)}N\|_F \\
&\lesssim \sqrt{\mathrm{Tr}(\Sigma_{YY|X_1})(k + \log k/\delta)}. \qquad \text{(from Claim A.4)}
\end{aligned}$$

The last inequality is derived from Claim A.4 and the fact that each row of $N$ follows gaussian distribution $\mathcal{N}(0, \mathbf{\Sigma}_{YY|X_1})$. Therefore

$$\frac{1}{n_2}\|\psi(\mathbf{X}_1)W^* - \psi(\mathbf{X}_1)\hat{W}\|_F^2 \lesssim \frac{\text{Tr}(\mathbf{\Sigma}_{YY|X_1})(k + \log k/\delta)}{n_2}.$$

Next we need to concentrate $1/n\mathbf{X}_1^\top \mathbf{X}_1$ to $\mathbf{\Sigma}_X$. Suppose $\mathbb{E}^L[X_2|X_1] = \mathbf{B}^\top X_1$, i.e., $\psi(x_1) = \mathbf{B}^\top x_1$, and $\psi(\mathbf{X}_1) = \mathbf{X}_1\mathbf{B}$. With Claim A.2 we have $1/n\psi(\mathbf{X}_1)^\top \psi(\mathbf{X}_1) = 1/n\mathbf{B}^\top \mathbf{X}_1^\top \mathbf{X}_1 \mathbf{B}$ satisfies:

$$0.9\mathbf{B}^\top \mathbf{\Sigma}_X \mathbf{B} \preceq 1/n_2\psi(\mathbf{X}_1)^\top \psi(\mathbf{X}_1) \preceq 1.1\mathbf{B}^\top \mathbf{\Sigma}_X \mathbf{B}$$

Therefore we also have:

$$\begin{aligned}
&\mathbb{E}[\|(\mathbf{W}^* - \hat{\mathbf{W}})^\top \psi(x_1)\|^2] \\
=&\|\mathbf{\Sigma}_X^{1/2}\mathbf{B}(\mathbf{W}^* - \hat{\mathbf{W}})\|_F^2 \\
\leq&\frac{1}{0.9n_2}\|\psi(\mathbf{X}_1)\mathbf{W}^* - \psi(\mathbf{X}_1)\hat{\mathbf{W}}\|_F^2 \lesssim \frac{\text{Tr}(\mathbf{\Sigma}_{YY|X_1})(k + \log k/\delta)}{n_2}.
\end{aligned}$$

$\square$

## C.1  Omitted Proof for General Random Variables

*Proof of Lemma 3.1.* Let the representation function $\psi$ be defined as:

$$\begin{aligned}
\psi(\cdot) := \mathbb{E}[X_2|X_1] =&\mathbb{E}[\mathbb{E}[X_2|X_1, Y]|X_1] \\
=&\mathbb{E}[\mathbb{E}[X_2|Y]|X_1] \qquad\qquad\text{(uses CI)} \\
=&\sum_y P(Y = y|X_1)\mathbb{E}[X_2|Y = y] \\
=:&f(X_1)^\top \mathbf{A},
\end{aligned}$$

where $f : \mathbb{R}^{d_1} \to \Delta_{\mathcal{Y}}$ satisfies $f(x_1)_y = P(Y = y|X_1 = x_1)$, and $\mathbf{A} \in \mathbb{R}^{\mathcal{Y} \times d_2}$ satisfies $\mathbf{A}_{y,:} = \mathbb{E}[X_2|Y = y]$. Here $\Delta_d$ denotes simplex of dimension $d$, which represents the discrete probability density over support of size $d$.

Let $\mathbf{B} = \mathbf{A}^\dagger \in \mathbb{R}^{\mathcal{Y} \times d_2}$ be the pseudoinverse of matrix $A$, and we get $\mathbf{B}\mathbf{A} = \mathbf{I}$ from our assumption that $\mathbf{A}$ is of rank $|\mathcal{Y}|$. Therefore $f(x_1) = \mathbf{B}\psi(x_1), \forall x_1$. Next we have:

$$\begin{aligned}
\mathbb{E}[Y|X_1 = x_1] =&\sum_y P(Y = y|X_1 = x_1) \times y \\
=&\mathbf{Y}f(x_1) \\
=&(\mathbf{Y}\mathbf{B}) \cdot \psi(X_1).
\end{aligned}$$

Here we denote by $\mathbf{Y} \in \mathbb{R}^{k \times \mathcal{Y}}, \mathbf{Y}_{:,y} = y$ that spans the whole support $\mathcal{Y}$. Therefore let $\mathbf{W}^* = \mathbf{Y}\mathbf{B}$ will finish the proof.

$\square$

*Proof of Theorem 3.2.* With Lemma 3.1 we know $e_{\text{apx}} = 0$, and therefore $\mathbf{W}^*\psi(X_1) \equiv f^*(X_1)$. Next from basic inequality and the same proof as in Theorem B.3 we have:

$$\|\psi(\mathbf{X}_1)\mathbf{W}^* - \psi(\mathbf{X}_1)\hat{\mathbf{W}}\| \leq 2\|\mathbf{P}_{\psi(\mathbf{X}_1)}\mathbf{N}\|_F$$

Notice $\mathcal{N}$ is a random noise matrix whose row vectors are independent samples from some centered distribution. Note we assumed $\mathbb{E}[\|N\|^2|\mathbf{X}_1] \leq \sigma^2$. $P_{\psi(\mathbf{X}_1)}$ is a projection to dimension $k$. From Lemma A.6 we have:

$$\|f^*(\mathbf{X}_1) - \psi(\mathbf{X}_1)\hat{\mathbf{W}}\| \leq \sigma\sqrt{k(1 + \log k/\delta)}.$$

Next, with Claim A.3 we have when $n \gg \rho^4(k + \log(1/\delta))$, since $\boldsymbol{W}^* - \hat{\boldsymbol{W}} \in \mathbb{R}^{d_2 \times k}$,

$$0.9(\boldsymbol{W}^* - \hat{\boldsymbol{W}})^\top \boldsymbol{\Sigma}_\psi (\boldsymbol{W}^* - \hat{\boldsymbol{W}})$$

$$\preceq \frac{1}{n_2}(\boldsymbol{W}^* - \hat{\boldsymbol{W}})^\top \sum_i \psi(x_1^{(i)})\psi(x_1^{(i)})^\top (\boldsymbol{W}^* - \hat{\boldsymbol{W}}) \preceq 1.1(\boldsymbol{W}^* - \hat{\boldsymbol{W}})^\top \boldsymbol{\Sigma}_\psi (\boldsymbol{W}^* - \hat{\boldsymbol{W}})$$

And therefore we could easily conclude that:

$$\mathbb{E}\|\hat{\boldsymbol{W}}^\top \psi(X_1) - f^*(X_1)\|^2 \lesssim \sigma^2 \frac{k(1 + \log(k/\delta))}{n_2}.$$

$\square$

## C.2 Omitted proof of linear model with approximation error

*Proof of Theorem 3.5.* First we note that $Y = f^*(X_1) + N$, where $\mathbb{E}[N|X_1] = 0$ but $Y - (\boldsymbol{A}^*)^\top X_1$ is not necessarily mean zero, and this is where additional difficulty lies. Write approximation error term $a(X_1) := f^*(X_1) - (\boldsymbol{A}^*)^\top X_1$, namely $Y = a(X_1) + (\boldsymbol{A}^*)^\top X_1 + N$. Also, $(\boldsymbol{A}^*)^\top X_1 \equiv (\boldsymbol{W}^*)^\top \psi(X_1)$ with conditional independence.

Second, with KKT condition on the training data, we know that $\mathbb{E}[a(X_1)X_1^\top] = 0$.

Recall $\hat{\boldsymbol{W}} = \arg\min_{\boldsymbol{W}} \|\boldsymbol{Y} - \psi(\boldsymbol{X}_1)\boldsymbol{W}\|_F^2$. We have the basic inequality,

$$\frac{1}{2n_2}\|\boldsymbol{Y} - \psi(\boldsymbol{X}_1)\hat{\boldsymbol{W}}\|_F^2 \leq \frac{1}{2n_2}\|\boldsymbol{Y} - \boldsymbol{X}_1\boldsymbol{A}^*\|_F^2$$

$$= \frac{1}{2n_2}\|\boldsymbol{Y} - \psi(\boldsymbol{X}_1)\boldsymbol{W}^*\|_F^2.$$

i.e., $\frac{1}{2n_2}\|\psi(\boldsymbol{X}_1)\boldsymbol{W}^* + a(\boldsymbol{X}_1) + \boldsymbol{N} - \psi(\boldsymbol{X}_1)\hat{\boldsymbol{W}}\|_F^2 \leq \frac{1}{2n_2}\|a(\boldsymbol{X}_1) + \boldsymbol{N}\|_F^2.$

Therefore

$$\frac{1}{2n_2}\|\psi(\boldsymbol{X}_1)\boldsymbol{W}^* - \psi(\boldsymbol{X}_1)\hat{\boldsymbol{W}}\|^2$$

$$\leq -\frac{1}{n_2}\langle a(\boldsymbol{X}_1) + \boldsymbol{N}, \psi(\boldsymbol{X}_1)\boldsymbol{W}^* - \psi(\boldsymbol{X}_1)\hat{\boldsymbol{W}}\rangle$$

$$= -\frac{1}{n_2}\langle a(\boldsymbol{X}_1), \psi(\boldsymbol{X}_1)\boldsymbol{W}^* - \psi(\boldsymbol{X}_1)\hat{\boldsymbol{W}}\rangle - \langle \boldsymbol{N}, \psi(\boldsymbol{X}_1)\boldsymbol{W}^* - \psi(\boldsymbol{X}_1)\hat{\boldsymbol{W}}\rangle \qquad (9)$$

With Assumption 3.3 and by concentration $0.9\frac{1}{n_2}\boldsymbol{X}_1\boldsymbol{X}_1^\top \preceq \boldsymbol{\Sigma}_{X_1} \preceq 1.1\frac{1}{n_2}\boldsymbol{X}_1\boldsymbol{X}_1^\top$, we have

$$\frac{1}{\sqrt{n_2}}\|a(\boldsymbol{X}_1)\boldsymbol{X}_1^\top \boldsymbol{\Sigma}_{X_1}^{-1/2}\|_F \leq 1.1b_0\sqrt{k} \qquad (10)$$

Denote $\psi(\boldsymbol{X}_1) = \boldsymbol{X}_1\boldsymbol{B}$, where $\boldsymbol{B} = \boldsymbol{\Sigma}_{X_1}^{-1}\boldsymbol{\Sigma}_{X_1 X_2}$ is rank $k$ under exact CI since $\boldsymbol{\Sigma}_{X_1 X_2} = \boldsymbol{\Sigma}_{X_1 Y}\boldsymbol{\Sigma}_Y^{-1}\boldsymbol{\Sigma}_{Y X_2}$. We have

$$\frac{1}{n_2}\langle a(\boldsymbol{X}_1), \psi(\boldsymbol{X}_1)\boldsymbol{W}^* - \psi(\boldsymbol{X}_1)\hat{\boldsymbol{W}}\rangle$$

$$= \frac{1}{n_2}\langle a(\boldsymbol{X}_1), \boldsymbol{X}_1\boldsymbol{B}\boldsymbol{W}^* - \boldsymbol{X}_1\boldsymbol{B}\hat{\boldsymbol{W}}\rangle$$

$$= \frac{1}{n_2}\langle \boldsymbol{\Sigma}_{X_1}^{-1/2}\boldsymbol{X}_1^\top a(\boldsymbol{X}_1), \boldsymbol{\Sigma}_{X_1}^{1/2}(\boldsymbol{B}\boldsymbol{W}^* - \boldsymbol{B}\hat{\boldsymbol{W}})\rangle$$

$$\leq 1.1b_0\sqrt{\frac{k}{n_2}}\|\boldsymbol{\Sigma}_{X_1}^{1/2}(\boldsymbol{B}\boldsymbol{W}^* - \boldsymbol{B}\hat{\boldsymbol{W}})\|_F \qquad \text{(from Ineq. (10))}$$

Back to Eqn. (9), we get

$$\frac{1}{2n_2}\|\psi(\boldsymbol{X}_1)\boldsymbol{W}^* - \psi(\boldsymbol{X}_1)\hat{\boldsymbol{W}}\|_F^2$$

$$\lesssim \sqrt{\frac{k}{n_2}}\|\boldsymbol{\Sigma}_{X_1}^{1/2}(\boldsymbol{BW}^* - \boldsymbol{B\hat{W}})\|_F + \frac{1}{n_2}\|P_{\boldsymbol{X}_1}\boldsymbol{N}\|_F\|\boldsymbol{X}_1(\boldsymbol{BW}^* - \boldsymbol{B\hat{W}})\|_F$$

$$\lesssim \left(\frac{\sqrt{k}}{n_2} + \frac{1}{n_2}\|P_{\boldsymbol{X}_1}\boldsymbol{N}\|_F\right)\|\boldsymbol{X}_1(\boldsymbol{BW}^* - \boldsymbol{B\hat{W}})\|_F$$

$$\implies \frac{1}{\sqrt{n_2}}\|\psi(\boldsymbol{X}_1)\boldsymbol{W}^* - \psi(\boldsymbol{X}_1)\hat{\boldsymbol{W}}\|_F \lesssim \sqrt{\frac{k(1 + \log k/\delta)}{n_2}}. \qquad \text{(from Lemma A.6)}$$

Finally, by concentration we transfer the result from empirical loss to excess risk and get:

$$\mathbb{E}[\|\psi(X_1)\boldsymbol{W}^* - \psi(X_1)\hat{\boldsymbol{W}}\|^2] \lesssim \frac{k(1 + \log(k/\delta))}{n_2}.$$

$\square$

### C.3 Argument on Denoising Auto-encoder or Context Encoder

**Remark C.1.** *We note that since $X_1 \perp X_2|Y$ ensures $X_1 \perp h(X_2)|Y$ for any deterministic function $h$, we could replace $X_2$ by $h(X_2)$ and all results hold. Therefore in practice, we could use $h(\psi(X_1))$ instead of $\psi(X_1)$ for downstream task. Specifically with denoising auto-encoder or context encoder, one could think about $h$ as the inverse of decoder $D$ ($h = D^{-1}$) and use $D^{-1}\psi \equiv E$ the encoder function as the representation for downstream tasks, which is more commonly used in practice.*

This section explains what we claim in Remark C.1. For context encoder, the reconstruction loss targets to find the encoder $E^*$ and decoder $D^*$ that achieve

$$\min_E \min_D \mathbb{E}\|X_2 - D(E(X_1))\|_F^2, \tag{11}$$

where $X_2$ is the masked part we want to recover and $X_1$ is the remainder.

If we naively apply our theorem we should use $D^*(E^*(\cdot))$ as the representation, while in practice we instead use only the encoder part $E^*(\cdot)$ as the learned representation. We argue that our theory also support this practical usage if we view the problem differently. Consider the pretext task to predict $(D^*)^{-1}(X_2)$ instead of $X_2$ directly, namely,

$$\bar{E} \leftarrow \arg\min_E \mathbb{E}\|(D^*)^{-1}(X_2) - E(X_1)\|^2, \tag{12}$$

and then we should indeed use $E(X_1)$ as the representation. On one hand, when $X_1 \perp X_2|Y$, it also satisfies $X_1 \perp (D^*)^{-1}(X_2)|Y$ since $(D^*)^{-1}$ is a deterministic function of $X_2$ and all our theory applies. On the other hand, the optimization on (11) or (12) give us similar result. Let

$$E^* = \arg\min_E \mathbb{E}[\|X_2 - D^*(E(X_1))\|^2],$$

and $\mathbb{E}\|X_2 - D^*(E^*(X_1))\|^2 \le \epsilon$, then with pretext task as in (12) we have that:

$$\mathbb{E}\|(D^*)^{-1}(X_2) - E^*(X_1)\|^2 = \mathbb{E}\|(D^*)^{-1}(X_2) - (D^*)^{-1} \circ D^*(E^*(X_1))\|^2$$

$$\le \|(D^*)^{-1}\|_{\text{Lip}}^2 \mathbb{E}\|X_2 - D^*(E^*(X_1))\|^2$$

$$\le L^2\epsilon,$$

where $L := \|(D^*)^{-1}\|_{\text{Lip}}$ is the Lipschitz constant for function $(D^*)^{-1}$. This is to say, in practice, we optimize over (11), and achieves a good representation $E^*(X_1)$ such that $\epsilon_{\text{pre}} \le L\sqrt{\epsilon}$ and thus performs well for downstream tasks. (Recall $\epsilon_{\text{pre}}$ is defined in Theorem 4.2 that measures how well we have learned the pretext task.)

# D Omitted Proofs Beyond Conditional Independence

## D.1 Warm-up: Jointly Gaussian Variables

As before, for simplicity we assume all data is centered in this case.

**Assumption D.1** (Approximate Conditional Independent Given Latent Variables). *Assume there exists some latent variable $Z \in \mathbb{R}^m$ such that*

$$\|\boldsymbol{\Sigma}_{X_1}^{-1/2}\boldsymbol{\Sigma}_{X_1,X_2|\bar{Y}}\|_F \leq \epsilon_{CI},$$

$\sigma_{k+m}(\boldsymbol{\Sigma}_{Y\bar{Y}}^{\dagger}\boldsymbol{\Sigma}_{\bar{Y}X_2}) = \beta > 0$ [8] *and* $\boldsymbol{\Sigma}_{X_2,\bar{Y}}$ *is of rank* $k+m$, *where* $\bar{Y} = [Y, Z]$.

When $X_1$ is not exactly CI of $X_2$ given $Y$ and $Z$, the approximation error depends on the norm of $\|\boldsymbol{\Sigma}_{X_1}^{-1/2}\boldsymbol{\Sigma}_{X_1,X_2|\bar{Y}}\|_2$. Let $\hat{\boldsymbol{W}}$ be the solution from Equation uses CI.

**Theorem D.1.** *Under Assumption D.1 with constant $\epsilon_{CI}$ and $\beta$, then the excess risk satisfies*

$$\mathrm{ER}_{\psi^*}[\hat{\boldsymbol{W}}] := \mathbb{E}[\|\hat{\boldsymbol{W}}^{\top}\psi^*(X_1) - f^*(X_1)\|_F^2] \lesssim \frac{\epsilon_{CI}^2}{\beta^2} + \mathrm{Tr}(\boldsymbol{\Sigma}_{YY|X_1})\frac{d_2 + \log(d_2/\delta)}{n_2}.$$

*Proof of Theorem D.1.* Let $\boldsymbol{V} := f^*(\boldsymbol{X}_1) \equiv \boldsymbol{X}_1\boldsymbol{\Sigma}_{X_1X_1}^{-1}\boldsymbol{\Sigma}_{1Y}$ be our target direction. Denote the optimal representation matrix by $\Psi := \psi(\boldsymbol{X}_1) \equiv \boldsymbol{X}_1\boldsymbol{A}$ (where $\boldsymbol{A} := \boldsymbol{\Sigma}_{X_1X_1}^{-1}\boldsymbol{\Sigma}_{X_1X_2}$).

Next we will make use of the conditional covariance matrix:

$$\boldsymbol{\Sigma}_{X_1X_2|\bar{Y}} := \boldsymbol{\Sigma}_{X_1X_2} - \boldsymbol{\Sigma}_{X_1\bar{Y}}\boldsymbol{\Sigma}_{\bar{Y}}^{-1}\boldsymbol{\Sigma}_{\bar{Y}X_2},$$

and plug it in into the definition of $\Psi$:

$$\begin{aligned}\Psi =& \boldsymbol{X}_1\boldsymbol{\Sigma}_{X_1X_1}^{-1}\boldsymbol{\Sigma}_{X_1\bar{Y}}\boldsymbol{\Sigma}_{\bar{Y}}^{-1}\boldsymbol{\Sigma}_{\bar{Y}X_2} + \boldsymbol{X}_1\boldsymbol{\Sigma}_{X_1X_1}^{-1}\boldsymbol{\Sigma}_{X_1X_2|\bar{Y}} \\ =& :\boldsymbol{L} + \boldsymbol{E},\end{aligned}$$

where $\boldsymbol{L} := \boldsymbol{X}_1\boldsymbol{\Sigma}_{X_1X_1}^{-1}\boldsymbol{\Sigma}_{X_1\bar{Y}}\boldsymbol{\Sigma}_{\bar{Y}}^{-1}\boldsymbol{\Sigma}_{\bar{Y}X_2}$ and $\boldsymbol{E} := \boldsymbol{X}_1\boldsymbol{\Sigma}_{X_1X_1}^{-1}\boldsymbol{\Sigma}_{X_1X_2|\bar{Y}}$. We analyze these two terms respectively.

For $\boldsymbol{L}$, we note that span$(\boldsymbol{V}) \subseteq$ span$(\boldsymbol{L})$: $\boldsymbol{L}\boldsymbol{\Sigma}_{X_2\bar{Y}}^{\dagger}\boldsymbol{\Sigma}_{\bar{Y}} = \boldsymbol{X}_1\boldsymbol{\Sigma}_{X_1X_1}^{-1}\boldsymbol{\Sigma}_{X_1\bar{Y}}$. By right multiplying the selector matrix $S_Y$ we have: $\boldsymbol{L}\boldsymbol{\Sigma}_{X_2\bar{Y}}^{\dagger}\boldsymbol{\Sigma}_{\bar{Y}Y} = \boldsymbol{X}_1\boldsymbol{\Sigma}_{X_1X_1}^{-1}\boldsymbol{\Sigma}_{X_1Y}$, i.e., $\boldsymbol{L}\bar{\boldsymbol{W}} = \boldsymbol{V}$, where $\bar{\boldsymbol{W}} := \boldsymbol{\Sigma}_{X_2\bar{Y}}^{\dagger}\boldsymbol{\Sigma}_{\bar{Y}Y}$. From our assumption that $\sigma_r(\boldsymbol{\Sigma}_{\bar{Y}Y}^{\dagger}\boldsymbol{\Sigma}_{\bar{Y}X_2}) = \beta$, we have $\|\bar{\boldsymbol{W}}\|_2 \leq \|\boldsymbol{\Sigma}_{X_2\bar{Y}}^{\dagger}\boldsymbol{\Sigma}_{\bar{Y}}\|_2 \leq 1/\beta$. (Or we could directly define $\beta$ as $\sigma_k(\boldsymbol{\Sigma}_{Y\bar{Y}}^{\dagger}\boldsymbol{\Sigma}_{\bar{Y}X_2}) \equiv \|\bar{\boldsymbol{W}}\|_2$. )

By concentration, we have $\boldsymbol{E} = \boldsymbol{X}_1\boldsymbol{\Sigma}_{X_1X_1}^{-1}\boldsymbol{\Sigma}_{X_1X_2|\bar{Y}}$ converges to $\boldsymbol{\Sigma}_{X_1X_1}^{-1/2}\boldsymbol{\Sigma}_{X_1X_2|\bar{Y}}$. Specifically, when $n \gg k + \log 1/\delta$, $\|\boldsymbol{E}\|_F \leq 1.1\|\boldsymbol{\Sigma}_{X_1X_1}^{-1/2}\boldsymbol{\Sigma}_{X_1X_2|\bar{Y}}\|_F \leq 1.1\epsilon_{CI}$ (by using Lemma A.2 ). Together we have $\|\boldsymbol{E}\bar{\boldsymbol{W}}\|_F \lesssim \epsilon_{CI}/\beta$.

Let $\hat{\boldsymbol{W}} = \arg\min_{\boldsymbol{W}} \|\boldsymbol{Y} - \Psi\boldsymbol{W}\|^2$. We note that $\boldsymbol{Y} = \boldsymbol{N} + \boldsymbol{V} = \boldsymbol{N} + \Psi\bar{\boldsymbol{W}} - \boldsymbol{E}\bar{\boldsymbol{W}}$ where $\boldsymbol{V}$ is our target direction and $\boldsymbol{N}$ is random noise (each row of $\boldsymbol{N}$ has covariance matrix $\boldsymbol{\Sigma}_{YY|X_1}$).

From basic inequality, we have:

$$\begin{aligned}\|\Psi\hat{\boldsymbol{W}} - \boldsymbol{Y}\|_F^2 \leq& \|\Psi\bar{\boldsymbol{W}} - \boldsymbol{Y}\|_F^2 = \|\boldsymbol{N} - \boldsymbol{E}\bar{\boldsymbol{W}}\|_F^2. \\ \implies \|\Psi\hat{\boldsymbol{W}} - \boldsymbol{V} - \boldsymbol{E}\bar{\boldsymbol{W}}\|^2 \leq& 2\langle \Psi\hat{\boldsymbol{W}} - \boldsymbol{V} - \boldsymbol{E}\bar{\boldsymbol{W}}, \boldsymbol{N} - \boldsymbol{E}\bar{\boldsymbol{W}}\rangle \\ \implies \|\Psi\hat{\boldsymbol{W}} - \boldsymbol{V} - \boldsymbol{E}\bar{\boldsymbol{W}}\| \leq& \|P_{[\Psi,E,V]}\boldsymbol{N}\| + \|\boldsymbol{E}\bar{\boldsymbol{W}}\| \\ \implies \|\Psi\hat{\boldsymbol{W}} - \boldsymbol{V}\| \lesssim& \|\boldsymbol{E}\|_F\|\bar{\boldsymbol{W}}\| + (\sqrt{d_2} + \sqrt{\log 1/\delta})\sqrt{\mathrm{Tr}(\boldsymbol{\Sigma}_{YY|X_1})}.\end{aligned}$$
(from Lemma A.4)

$$\leq \sqrt{n_2}\frac{\epsilon_{CI}}{\beta} + (\sqrt{d_2} + \sqrt{\log 1/\delta})\sqrt{\mathrm{Tr}(\boldsymbol{\Sigma}_{YY|X_1})}.$$
(from Assumption D.1)

---

[8] $\sigma_k(\boldsymbol{A})$ denotes $k$-th singular value of $\boldsymbol{A}$, and $\boldsymbol{A}^{\dagger}$ is the pseudo-inverse of $\boldsymbol{A}$.

Next, by the same procedure that concentrates $\frac{1}{n_2}\boldsymbol{X}_1^\top \boldsymbol{X}_1$ to $\boldsymbol{\Sigma}_{X_1 X_1}$ with Claim A.2, we could easily get

$$\text{ER}[\hat{\boldsymbol{W}}] := \mathbb{E}[\|\hat{\boldsymbol{W}}^\top \psi(X_1) - f^*(X_1)\|^2] \lesssim \frac{\epsilon_{\text{CI}}^2}{\beta^2} + \text{Tr}(\boldsymbol{\Sigma}_{YY|X_1})\frac{d_2 + \log 1/\delta}{n_2}.$$

$\square$

## D.2 Measuring conditional dependence with cross-covariance operator

$L^2(P_X)$ denotes the Hilbert space of square integrable function with respect to the measure $P_X$, the marginal distribution of $X$. We are interested in some function class $\mathcal{H}_x \subset L^2(P_X)$ that is induced from some feature maps:

**Definition D.2** (General and Universal feature Map). *We denote feature map $\phi : \mathcal{X} \to \mathcal{F}$ that maps from a compact input space $\mathcal{X}$ to the feature space $\mathcal{F}$. $\mathcal{F}$ is a Hilbert space associated with inner product: $\langle \phi(\boldsymbol{x}), \phi(\boldsymbol{x}')\rangle_{\mathcal{F}}$. The associated function class is: $\mathcal{H}_x = \{h : \mathcal{X} \to \mathbb{R} | \exists w \in \mathcal{F}, h(\boldsymbol{x}) = \langle w, \phi(\boldsymbol{x})\rangle_{\mathcal{F}}, \forall \boldsymbol{x} \in \mathcal{X}\}$. We call $\phi$ universal if the induced $\mathcal{H}_x$ is dense in $L^2(P_X)$.*

Linear model is a special case when feature map $\phi = Id$ is identity mapping and the inner product is over Euclidean space. A feature map with higher order polynomials correspondingly incorporate high order moments [20, 24]. For discrete variable $Y$ we overload $\phi$ as the one-hot embedding.

**Remark D.1.** *For continuous data, any universal kernel like Gaussian kernel or RBF kernel induce the universal feature map that we require [41]. Two-layer neural network with infinite width also satisfy it, i.e., $\forall \boldsymbol{x} \in \mathcal{X} \subset \mathbb{R}^d, \phi_{NN}(\boldsymbol{x}) : \mathcal{S}^{d-1} \times \mathbb{R} \to \mathbb{R}, \phi_{NN}(\boldsymbol{x})[\boldsymbol{w}, b] = \sigma(\boldsymbol{w}^\top \boldsymbol{x} + b)$ [7].*

When there's no ambiguity, we overload $\phi_1$ as the random variable $\phi_1(X_1)$ over domain $\mathcal{F}_1$, and $\mathcal{H}_1$ as the function class over $X_1$. Next we characterize CI using the cross-covariance operator.

**Definition D.3** (Cross-covariance operator). *For random variables $X \in \mathcal{X}, Y \in \mathcal{Y}$ with joint distribution $P : \mathcal{X} \times \mathcal{Y} \to \mathbb{R}$, and associated feature maps $\phi_x$ and $\phi_y$, we denote by $\mathcal{C}_{\phi_x \phi_y} = \mathbb{E}[\phi_x(X) \otimes \phi_y(Y)] = \int_{\mathcal{X} \times \mathcal{Y}} \phi_x(x) \otimes \phi_y(y) dP(x, y)$, the (un-centered) cross-covariance operator. Similarly we denote by $\mathcal{C}_{X\phi_y} = \mathbb{E}[X \otimes \phi_y(Y)] : \mathcal{F}_y \to \mathcal{X}$.*

To understand what $\mathcal{C}_{\phi_x \phi_y}$ is, we note it is of the same shape as $\phi_x(x) \otimes \phi_y(y)$ for each individual $x \in \mathcal{X}, y \in \mathcal{Y}$. It can be viewed as an operator: $\mathcal{C}_{\phi_x \phi_y} : \mathcal{F}_y \to \mathcal{F}_x, \mathcal{C}_{\phi_x \phi_y} f = \int_{\mathcal{X} \times \mathcal{Y}} \langle \phi_y(y), f \rangle \phi_x(x) dP(x, y), \forall f \in \mathcal{F}_y$. For any $f \in \mathcal{H}_x$ and $g \in \mathcal{H}_y$, it satisfies: $\langle f, \mathcal{C}_{\phi_x \phi_y} g \rangle_{\mathcal{H}_x} = \mathbb{E}_{XY}[f(X)g(Y)]$[6, 20]. CI ensures $\mathcal{C}_{\phi_1 X_2 | \phi_y} = 0$ for arbitrary $\phi_1, \phi_2$:

**Lemma D.4.** *With one-hot encoding map $\phi_y$ and arbitrary $\phi_1$, $X_1 \perp X_2 | Y$ ensures:*

$$\mathcal{C}_{\phi_1 X_2 | \phi_y} := \mathcal{C}_{\phi_1 X_2} - \mathcal{C}_{\phi_1 \phi_y} \mathcal{C}_{\phi_y \phi_y}^{-1} \mathcal{C}_{\phi_y X_2} = 0. \tag{13}$$

A more complete discussion of cross-covariance operator and CI can be found in [20]. Also, recall that an operator $\mathcal{C} : \mathcal{F}_y \to \mathcal{F}_x$ is Hilbert-Schmidt (HS) [50] if for complete orthonormal systems (CONSs) $\{\zeta_i\}$ of $\mathcal{F}_x$ and $\{\eta_i\}$ of $\mathcal{F}_y$, $\|\mathcal{C}\|_{\text{HS}}^2 := \sum_{i,j} \langle \zeta_j, \mathcal{C}\eta_i \rangle_{\mathcal{F}_x}^2 < \infty$. The Hilbert-Schmidt norm generalizes the Frobenius norm from matrices to operators, and we will later use $\|\mathcal{C}_{\phi_1 X_2 | \phi_y}\|$ to quantify approximate CI.

We note that covariance operators [21, 20, 6] are commonly used to capture conditional dependence of random variables. In this work, we utilize the covariance operator to quantify the performance of the algorithm even when the algorithm is *not a kernel method*.

## D.3 Omitted Proof in General Setting

**Claim D.5.** *For feature maps $\phi_1$ with universal property, we have:*

$$\psi^*(X_1) := \mathbb{E}[X_2|X_1] = \mathbb{E}^L[X_2|\phi_1]$$
$$= \mathcal{C}_{X_2\phi_1}\mathcal{C}_{\phi_1\phi_1}^{-1}\phi_1(X_1).$$
$$\textit{Our target } f^*(X_1) := \mathbb{E}[Y|X_1] = \mathbb{E}^L[Y|\phi_1]$$
$$= \mathcal{C}_{Y\phi_1}\mathcal{C}_{\phi_1\phi_1}^{-1}\phi_1(X_1).$$

*For general feature maps, we instead have:*

$$\psi^*(X_1) := \arg\min_{f \in \mathcal{H}_1^{d_2}} \mathbb{E}_{X_1 X_2} \|X_2 - f(X_1)\|_2^2$$

$$= \mathcal{C}_{X_2 \phi_1} \mathcal{C}_{\phi_1 \phi_1}^{-1} \phi_1(X_1).$$

$$\textit{Our target } f^*(X_1) := \arg\min_{f \in \mathcal{H}_1^k} \mathbb{E}_{X_1 Y} \|Y - f(X_1)\|_2^2$$

$$= \mathcal{C}_{Y \phi_1} \mathcal{C}_{\phi_1 \phi_1}^{-1} \phi_1(X_1).$$

To prove Claim D.5, we show the following lemma:

**Lemma D.6.** *Let $\phi : \mathcal{X} \to \mathcal{F}_x$ be a universal feature map, then for random variable $Y \in \mathcal{Y}$ we have:*

$$\mathbb{E}[Y|X] = \mathbb{E}^L[Y|\phi(X)].$$

*Proof of Lemma D.6.* Denote by $\mathbb{E}[Y|X = x] =: f(x)$. Since $\phi$ is dense in $\mathcal{X}$, there exists a linear operator $a : \mathcal{X} \to \mathbb{R}$ such that $\int_{x \in \mathcal{X}} a(x)\phi(x)[\cdot]dx = f(\cdot)$ a.e. Therefore the result comes directly from the universal property of $\phi$. $\qquad \square$

*Proof of Claim D.5.* We want to show that for random variables $Y, X$, where $X$ is associated with a universal feature map $\phi_x$, we have $\mathbb{E}[Y|X] = \mathcal{C}_{Y \phi_x(X)} \mathcal{C}_{\phi_x(X)\phi_x(X)}^{-1} \phi_x(X)$.

First, from Lemma D.6, we have that $\mathbb{E}[Y|X] = \mathbb{E}^L[Y|\phi_x(X)]$. Next, write $A^* : \mathcal{F}_x \to \mathcal{Y}$ as the linear operator that satisfies

$$\mathbb{E}[Y|X] = A^*\phi_x(X)$$

$$\text{s.t. } A^* = \arg\min_{A} \mathbb{E}[\|Y - A\phi_x(X)\|^2].$$

Therefore from the stationary condition we have $A^* \mathbb{E}_X[\phi_x(X) \otimes \phi_x(X)] = \mathbb{E}_{XY}[Y \otimes \phi_x(X)]$. Or namely we get $A^* = \mathcal{C}_{Y\phi_x} \mathcal{C}_{\phi_x\phi_x}^{-1}$ simply from the definition of the cross-covariance operator $\mathcal{C}$. $\quad \square$

**Claim D.7.** $\|\mathcal{C}_{\phi_1\phi_1}^{-1/2} \mathcal{C}_{\phi_1 X_2 | \phi_{\bar{y}}}\|_{HS}^2 = \mathbb{E}_{X_1}[\|\mathbb{E}[X_2|X_1] - \mathbb{E}_{\bar{Y}}[\mathbb{E}[X_2|\bar{Y}]|X_1]\|^2] = \epsilon_{CI}^2.$

*Proof.*

$$\|\mathcal{C}_{\phi_1\phi_1}^{-1/2} \mathcal{C}_{\phi_1 X_2 | \phi_{\bar{y}}}\|_{HS}^2$$

$$= \int_{X_1} \left\| \int_{X_2} \left( \frac{p_{X_1 X_2}(\boldsymbol{x}_1, \boldsymbol{x}_2)}{p_{X_1}(\boldsymbol{x}_1)} - \frac{p_{X_1 \perp X_2 | Y}(\boldsymbol{x}_1, \boldsymbol{x}_2)}{p_{X_1}(\boldsymbol{x}_1)} \right) X_2 dp_{\boldsymbol{x}_2} \right\|^2 dp_{\boldsymbol{x}_1}$$

$$= \mathbb{E}_{X_1}[\|\mathbb{E}[X_2|X_1] - \mathbb{E}_{\bar{Y}}[\mathbb{E}[X_2|\bar{Y}]|X_1]\|^2].$$

$\qquad\qquad\qquad \square$

## D.4 Omitted Proof for Main Results

We first prove a simpler version without approximation error.

**Theorem D.8.** *For a fixed $\delta \in (0, 1)$, under Assumption 4.1, 3.2, if there is no approximation error, i.e., there exists a linear operator $A$ such that $f^*(X_1) \equiv A\phi_1(X_1)$, if $n_1, n_2 \gg \rho^4(d_2 + \log 1/\delta)$, and we learn the pretext tasks such that:*

$$\mathbb{E}\|\tilde{\psi}(X_1) - \psi^*(X_1)\|_F^2 \le \epsilon_{pre}^2.$$

*Then we are able to achieve generalization for downstream task with probability $1 - \delta$:*

$$\mathbb{E}[\|f_{\mathcal{H}_1}^*(X_1) - \hat{\boldsymbol{W}}^\top \tilde{\psi}(X_1)\|^2] \le \tilde{\mathcal{O}}\{\sigma^2 \frac{d_2}{n_2} + \frac{\epsilon_{CI}^2}{\beta^2} + \frac{\epsilon_{pre}^2}{\beta^2}\}. \tag{14}$$

*Proof of Theorem D.8.* We follow the similar procedure as Theorem D.1. For the setting of no approximation error, we have $f^* = f^*_{\mathcal{H}_1}$, and the residual term $N := Y - f^*(X_1)$ is a mean-zero random variable with $\mathbb{E}[\|N\|^2|X_1] \lesssim \sigma^2$ according to our data assumption in Section 3. $N = Y - f^*(X_1^{\text{down}})$ is the collected $n_2$ samples of noise terms. We write $Y \in \mathbb{R}^{d_3}$. For classification task, we have $Y \in \{e_i, i \in [k]\} \subset \mathbb{R}^k$ (i.e, $d_3 = k$) is one-hot encoded random variable. For regression problem, $Y$ might be otherwise encoded. For instance, in the yearbook dataset, Y ranges from 1905 to 2013 and represents the years that the photos are taken. We want to note that our result is general for both cases: the bound doesn't depend on $d_3$, but only depends on the variance of $N$.

Let $\Psi^*, L, E, V$ be defined as follows:

Let $V = f^*(X_1^{\text{down}}) \equiv f^*_{\mathcal{H}_1}(X_1^{\text{down}}) \equiv \phi(X_1^{\text{down}})\mathcal{C}^{-1}_{\phi_1}\mathcal{C}_{\phi_1 Y}$ be our target direction. Denote the optimal representation matrix by

$$
\begin{aligned}
\Psi^* :=& \psi^*(X_1^{\text{down}}) \\
=& \phi(X_1^{\text{down}})\mathcal{C}^{-1}_{\phi_1\phi_1}\mathcal{C}_{\phi_1 X_2} \\
=& \phi(X_1^{\text{down}})\mathcal{C}^{-1}_{\phi_1\phi_1}\mathcal{C}_{\phi_1\phi_{\bar{y}}}\mathcal{C}^{-1}_{\phi_{\bar{y}}}\Sigma_{\phi_{\bar{y}}X_2} + \phi(X_1^{\text{down}})\mathcal{C}^{-1}_{\phi_1\phi_1}\mathcal{C}_{\phi_1 X_2|\phi_{\bar{y}}} \\
=:& L + E,
\end{aligned}
$$

where $L = \phi(X_1^{\text{down}})\mathcal{C}^{-1}_{\phi_1\phi_1}\mathcal{C}_{\phi_1\phi_{\bar{y}}}\mathcal{C}^{-1}_{\phi_{\bar{y}}}\mathcal{C}_{\phi_{\bar{y}}X_2}$ and $E = \phi(X_1^{\text{down}})\mathcal{C}^{-1}_{\phi_1\phi_1}\mathcal{C}_{\phi_1 X_2|\bar{Y}}$.

In this proof, we denote $S_Y$ as the matrix such that $S_Y \phi_{\bar{y}} = Y$. Specifically, if $Y$ is of dimension $d_3$, $S_Y$ is of size $d_3 \times |\mathcal{Y}||\mathcal{Z}|$. Therefore $S_Y\Sigma_{\phi_y A} = \Sigma_{YA}$ for any random variable $A$.

Therefore, similarly we have:

$$
L\Sigma^{\dagger}_{X_2\phi_{\bar{y}}}\Sigma_{\phi_{\bar{y}}\phi_{\bar{y}}}S_Y^{\top} = L\Sigma^{\dagger}_{X_2\phi_{\bar{y}}}\Sigma_{\phi_{\bar{y}}Y} = L\bar{W} = V
$$

where $\bar{W} := \Sigma^{\dagger}_{X_2\phi_{\bar{y}}}\Sigma_{\phi_{\bar{y}}Y}$ satisfies $\|\bar{W}\|_2 = 1/\beta$. Therefore $\text{span}(V) \subseteq \text{span}(L)$ since we have assumed that $\Sigma^{\dagger}_{X_2\phi_{\bar{y}}}\Sigma_{\phi_{\bar{y}}Y}$ to be full rank.

On the other hand, $E = \phi_1(X_1^{\text{down}})\mathcal{C}^{-1}_{\phi_1\phi_1}\mathcal{C}_{\phi_1 X_2|\bar{Y}}$ concentrates to $\mathcal{C}^{-1/2}_{\phi_1\phi_1}\mathcal{C}_{\phi_1 X_2|\phi_{\bar{y}}}$. Specifically, when $n \gg c + \log 1/\delta$, $\frac{1}{n_2}\|E\|_F^2 \leq 1.1\|\mathcal{C}^{-1/2}_{\phi_1\phi_1}\mathcal{C}_{\phi_1 X_2|\phi_{\bar{y}}}\|_F^2 \leq 1.1\epsilon_{\text{CI}}^2$ (by using Lemma A.3 ). Together we have $\|E\bar{W}\|_F \lesssim \epsilon_{\text{CI}}/\beta$.

We also introduce the error from not learning $\psi^*$ exactly: $E^{\text{pre}} = \Psi - \Psi^* := \tilde{\psi}(X_1^{\text{down}}) - \psi^*(X_1^{\text{down}})$. With proper concentration and our assumption, we have that $\mathbb{E}\|\psi(X_1) - \psi^*(X_1)\|^2 \leq \epsilon_{\text{pre}}$ and $\frac{1}{\sqrt{n_2}}\|\psi(X_1^{\text{down}}) - \psi^*(X_1^{\text{down}})\|^2 \leq 1.1\epsilon_{\text{pre}}$.

Also, the noise term after projection satisfies $\|P_{[\Psi,E,V]}N\| \lesssim \sqrt{d_2(1 + \log d_2/\delta)}\sigma$ as using Corollary A.6. Therefore $\Psi = \Psi^* - E^{\text{pre}} = L + E - E^{\text{pre}}$.

Recall that $\hat{W} = \arg\min_W \|\psi(X_1^{\text{down}})W - Y\|_F^2$. And with exactly the same procedure as Theorem D.1 we also get that:

$$
\begin{aligned}
\|\Psi\hat{W} - V\| \leq& 2\|E\bar{W}\| + 2\|E^{\text{pre}}\bar{W}\| + \|P_{[\Psi,E,V,E^{\text{pre}}]}N\| \\
\lesssim& \sqrt{n_2}\frac{\epsilon_{\text{CI}} + \epsilon_{\text{pre}}}{\beta} + \sigma\sqrt{d_2(1 + \log(d_2/\delta))}.
\end{aligned}
$$

With the proper concentration we also get:

$$
\mathbb{E}[\|\hat{W}^{\top}\psi(X_1) - f^*_{\mathcal{H}_1}(X_1)\|^2] \lesssim \frac{\epsilon_{\text{CI}}^2 + \epsilon_{\text{pre}}^2}{\beta^2} + \sigma^2\frac{d_2(1 + \log(d_2/\delta))}{n_2}.
$$

$\square$

Next we move on to the proof of our main result Theorem 4.2 where approximation error occurs.

*Proof of Theorem 4.2.* The proof is a combination of Theorem 3.5 and Theorem D.8. We follow the same notation as in Theorem D.8. Now the only difference is that an additional term $a(\boldsymbol{X}_1^{\mathrm{down}})$ is included in $\boldsymbol{Y}$:

$$
\begin{aligned}
\boldsymbol{Y} &= \boldsymbol{N} + f^*(\boldsymbol{X}_1^{\mathrm{down}}) \\
&= \boldsymbol{N} + \Psi^* \bar{\boldsymbol{W}} + a(\boldsymbol{X}_1^{\mathrm{down}}) \\
&= \boldsymbol{N} + (\Psi + \boldsymbol{E}^{\mathrm{pre}}) \bar{\boldsymbol{W}} + a(\boldsymbol{X}_1^{\mathrm{down}}) \\
&= \Psi \bar{\boldsymbol{W}} + (\boldsymbol{N} + \boldsymbol{E}^{\mathrm{pre}} \bar{\boldsymbol{W}} + a(\boldsymbol{X}_1^{\mathrm{down}})).
\end{aligned}
$$

From re-arranging $\frac{1}{2n_2} \|\boldsymbol{Y} - \Psi \hat{\boldsymbol{W}}\|_F^2 \le \frac{1}{2n_2} \|\boldsymbol{Y} - \Psi \bar{\boldsymbol{W}}\|_F^2$,

$$
\frac{1}{2n_2} \|\Psi(\bar{\boldsymbol{W}} - \hat{\boldsymbol{W}}) + (\boldsymbol{N} + \boldsymbol{E}^{\mathrm{pre}} + a(\boldsymbol{X}_1^{\mathrm{down}}))\|_F^2 \le \frac{1}{2n_2} \|\boldsymbol{N} + \boldsymbol{E}^{\mathrm{pre}} \bar{\boldsymbol{W}} + a(\boldsymbol{X}_1^{\mathrm{down}})\|_F^2 \quad (15)
$$

$$
\Rightarrow \frac{1}{2n_2} \|\Psi(\bar{\boldsymbol{W}} - \hat{\boldsymbol{W}})\|_F^2 \le \frac{1}{n_2} \langle \Psi(\bar{\boldsymbol{W}} - \hat{\boldsymbol{W}}), \boldsymbol{N} + \boldsymbol{E}^{\mathrm{pre}} \bar{\boldsymbol{W}} + a(\boldsymbol{X}_1^{\mathrm{down}}) \rangle. \quad (16)
$$

Then with similar procedure as in the proof of Theorem 3.5, and write $\Psi$ as $\phi(X_1^{\mathrm{down}})\boldsymbol{B}$, we have:

$$
\begin{aligned}
&\frac{1}{n_2} \langle \Psi(\bar{\boldsymbol{W}} - \hat{\boldsymbol{W}}), a(\boldsymbol{X}_1^{\mathrm{down}}) \rangle \\
&= \frac{1}{n_2} \langle \boldsymbol{B}(\bar{\boldsymbol{W}} - \hat{\boldsymbol{W}}), \phi(\boldsymbol{X}_1^{\mathrm{down}})^\top a(\boldsymbol{X}_1^{\mathrm{down}}) \rangle \\
&= \frac{1}{n_2} \langle \mathcal{C}_{\phi_1}^{1/2} \boldsymbol{B}(\bar{\boldsymbol{W}} - \hat{\boldsymbol{W}}), \mathcal{C}_{\phi_1}^{-1/2} \phi(\boldsymbol{X}_1^{\mathrm{down}})^\top a(\boldsymbol{X}_1^{\mathrm{down}}) \rangle \\
&\le \sqrt{\frac{d_2}{n_2}} \|\mathcal{C}_{\phi_1}^{1/2} \boldsymbol{B}(\bar{\boldsymbol{W}} - \hat{\boldsymbol{W}})\|_F \\
&\le 1.1 \frac{1}{\sqrt{n_2}} \sqrt{\frac{d_2}{n_2}} \|\phi(\boldsymbol{X}_1^{\mathrm{down}}) \boldsymbol{B}(\bar{\boldsymbol{W}} - \hat{\boldsymbol{W}})\|_F \\
&= 1.1 \frac{\sqrt{d_2}}{n_2} \|\Psi(\bar{\boldsymbol{W}} - \hat{\boldsymbol{W}})\|_F.
\end{aligned}
$$

Therefore plugging back to (16) we get:

$$
\frac{1}{2n_2} \|\Psi(\bar{\boldsymbol{W}} - \hat{\boldsymbol{W}})\|_F^2 \le \frac{1}{n_2} \langle \Psi(\bar{\boldsymbol{W}} - \hat{\boldsymbol{W}}), \boldsymbol{N} + \boldsymbol{E}^{\mathrm{pre}} \bar{\boldsymbol{W}} + a(\boldsymbol{X}_1^{\mathrm{down}}) \rangle
$$

$$
\Rightarrow \frac{1}{2n_2} \|\Psi(\bar{\boldsymbol{W}} - \hat{\boldsymbol{W}})\|_F \le \frac{1}{2n_2} \|\boldsymbol{E}^{\mathrm{pre}} \bar{\boldsymbol{W}}\|_F + \frac{1}{2n_2} \|P_\Psi \boldsymbol{N}\|_F + 1.1 \frac{\sqrt{d_2}}{n_2}.
$$

$$
\Rightarrow \frac{1}{2\sqrt{n_2}} \|\Psi \hat{\boldsymbol{W}} - f_{\mathcal{H}_1}^*(\boldsymbol{X}_1^{\mathrm{down}})\|_F - \|\boldsymbol{E} \bar{\boldsymbol{W}}\|_F \le \frac{1}{\sqrt{n_2}} (1.1 \sqrt{d_2} + \|\boldsymbol{E}^{\mathrm{pre}} \bar{\boldsymbol{W}}\| + \sqrt{d_2 + \log(d_2/\delta)})
$$

$$
\Rightarrow \frac{1}{2\sqrt{n_2}} \|\Psi \hat{\boldsymbol{W}} - f_{\mathcal{H}_1}^*(\boldsymbol{X}_1^{\mathrm{down}})\|_F \lesssim \sqrt{\frac{d_2(1 + \log d_2/\delta)}{n_2}} + \frac{\epsilon_{\mathrm{CI}} + \epsilon_{\mathrm{pre}}}{\beta}.
$$

Finally by concentrating $\frac{1}{n_2} \Psi^\top \Psi$ to $\mathbb{E}[\tilde{\psi}(X_1)\tilde{\psi}(X_1)^\top]$ we get:

$$
\mathbb{E}[\|\hat{\boldsymbol{W}}^\top \tilde{\psi}(X_1) - f_{\mathcal{H}_1}^*(X_1)\|_2^2] \lesssim \frac{d_2(1 + \log d_2/\delta)}{n_2}) + \frac{\epsilon_{\mathrm{CI}}^2 + \epsilon_{\mathrm{pre}}^2}{\beta^2},
$$

with probability $1 - \delta$. $\qquad\square$

### D.5 Principal Component Regression

**Claim D.9** (Approximation Error of Principle Component Analysis). *Let matrix $\boldsymbol{A} = \boldsymbol{L} + \boldsymbol{E} \in \mathbb{R}^{n \times d}$ where $\boldsymbol{L}$ has rank $r <$ size of $\boldsymbol{A}$. Let $\boldsymbol{A}_r$ be the rank-$r$ PCA of A. Then we have: $\|\boldsymbol{A}_r - \boldsymbol{L}\|_F \le 2\|\boldsymbol{E}\|_F$, and $\|\boldsymbol{A}_r - \boldsymbol{L}\|_2 \le 2\|\boldsymbol{E}\|_2$.*

*Proof.* Due to the property of PCA, $\|\boldsymbol{A}_r - \boldsymbol{A}\|_F \leq \|\boldsymbol{E}\|_F$ and $\|\boldsymbol{A}_r - \boldsymbol{A}\|_2 \leq \|\boldsymbol{E}\|_2$.

$$\begin{aligned}
\|\boldsymbol{A}_r - \boldsymbol{L}\|_2 =& \|\boldsymbol{A}_r - \boldsymbol{A} + \boldsymbol{A} - \boldsymbol{L}\|_2 \\
\leq& \|\boldsymbol{A}_r - \boldsymbol{A}\|_F + \|\boldsymbol{E}\|_F \\
\leq& 2\|\boldsymbol{E}\|_2.
\end{aligned}$$

Similarly we have $\|\boldsymbol{A}_r - \boldsymbol{L}\|_F \leq 2\|\boldsymbol{E}\|_F$. $\qquad\square$

This technical fact could be used to complete the proof for Remark 4.1.

*Proof of Remark 4.1.* We replace the key steps of D.8.

Recall $\Psi^*, \boldsymbol{L}, \boldsymbol{E}, \boldsymbol{V}$ are defined as follows:

$\Psi^* := \psi^*(\boldsymbol{X}_1^{\text{down}})$ is the optimal representation matrix. $\Psi_r$ is the features obtained from $r$-PCA of $\Psi^*$. $\Psi^* = \boldsymbol{L} + \boldsymbol{E}$ which is low rank plus small norm. ($\boldsymbol{L} = \phi(\boldsymbol{X}_1^{\text{down}})\mathcal{C}_{\phi_1\phi_1}^{-1}\mathcal{C}_{\phi_1\phi_{\bar{y}}}\mathcal{C}_{\phi_{\bar{y}}}^{-1}\mathcal{C}_{\phi_{\bar{y}}X_2}$ and $\boldsymbol{E} = \phi(\boldsymbol{X}_1^{\text{down}})\mathcal{C}_{\phi_1\phi_1}^{-1}\mathcal{C}_{\phi_1 X_2|\bar{Y}}$. Suppose $r = |\mathcal{Y}||\mathcal{Z}|$.) Let $\boldsymbol{V} = f^*(\boldsymbol{X}_1^{\text{down}}) \equiv f^*_{\mathcal{H}_1}(\boldsymbol{X}_1^{\text{down}}) \equiv \phi(\boldsymbol{X}_1^{\text{down}})\mathcal{C}_{\phi_1}^{-1}\mathcal{C}_{\phi_1 Y} = \boldsymbol{L}\bar{\boldsymbol{W}}$ be our target direction, where $\bar{\boldsymbol{W}} := \boldsymbol{\Sigma}_{X_2\phi_{\bar{y}}}^{\dagger}\boldsymbol{\Sigma}_{\phi_{\bar{y}}Y}$.

Due to representation learning error (finite sample in the first stage) and approximate conditional independence, the target direction $\boldsymbol{V}$ is not perfectly linear in $\Psi^*$ or its $r$-PCA features $\Psi$.

Now with PCR we learn the linear model with $\hat{\boldsymbol{W}} \leftarrow \arg\min_{\boldsymbol{W}} \|\Psi_r \boldsymbol{W} - \boldsymbol{Y}\|_F^2$. Together with D.9 and the same procedure as Theorem D.8 we also get that:

Let $\bar{\boldsymbol{E}} = \boldsymbol{L} - \Psi_r$ is of rank at most $2r$.

$$\begin{aligned}
\|\Psi_r \hat{\boldsymbol{W}} - \boldsymbol{Y}\|_F^2 \leq& \|\Psi_r \bar{\boldsymbol{W}} - \boldsymbol{Y}\|_F^2 = \|\boldsymbol{N} - \bar{\boldsymbol{E}}\bar{\boldsymbol{W}}\|_F^2. \\
\implies \|\Psi_r \hat{\boldsymbol{W}} - \boldsymbol{V} - \bar{\boldsymbol{E}}\bar{\boldsymbol{W}}\|^2 \leq& 2\langle \Psi_r \hat{\boldsymbol{W}} - \boldsymbol{V} - \bar{\boldsymbol{E}}\bar{\boldsymbol{W}}, \boldsymbol{N} - \bar{\boldsymbol{E}}\bar{\boldsymbol{W}}\rangle \\
\implies \|\Psi_r \hat{\boldsymbol{W}} - \boldsymbol{V} - \bar{\boldsymbol{E}}\bar{\boldsymbol{W}}\| \leq& \|P_{[\Psi_r, \boldsymbol{L}]}\boldsymbol{N}\| + \|\bar{\boldsymbol{E}}\bar{\boldsymbol{W}}\| \\
\implies \|\Psi_r \hat{\boldsymbol{W}} - \boldsymbol{V}\| \leq& 2\|\bar{\boldsymbol{E}}\|_F\|\bar{\boldsymbol{W}}\| + \|P_{2r}\boldsymbol{N}\| \\
\lesssim& \|\boldsymbol{E}\|_F\|\bar{\boldsymbol{W}}\| + \sigma\sqrt{r}(1 + \sqrt{\log(r/\delta)}).
\end{aligned}$$

With concentration on the downstream labeled samples we also get the result in Remark 4.1:

$$\mathbb{E}[\|\hat{\boldsymbol{W}}^{\top}\psi_r(X_1) - f^*_{\mathcal{H}_1}(X_1)\|^2] \lesssim \frac{\epsilon_{\text{CI}}^2 + \epsilon_{\text{pre}}^2}{\beta^2} + \sigma^2\frac{r(1 + \log(r/\delta))}{n_2}.$$

Here $r = |\mathcal{Y}||\mathcal{Z}|$. $\qquad\square$

# E   Omitted Proofs Beyond Conditional Independence

## E.1   Proof for topic modeling example

*Proof for Theorem 5.1.* We will construct a latent variable $\bar{Y}$ such that $\epsilon_{\text{CI}} = 0$. We pick the domain of $\bar{Y}$ to be $[k]$ and the distribution $P(\bar{Y}|X_1)$ to be the distribution $\mathbb{E}[\mu|X_1] \in \Delta_{[k]}$, and define $P(X_2|\bar{Y} = i) = P(X_2|\mu = e_i)$. More specifically we have

$$\begin{aligned}
P(\bar{Y} = i|X_1) = \mathbb{E}[\mu|X_1](i) = \mathbb{E}[\mu(i)|X_1] \text{ and thus } \mathbb{E}[\bar{Y}|X_1] = \mathbb{E}[\mu|X_1] \\
P(X_2|\bar{Y} = i) = P(X_2|\mu = e_i) \text{ and thus } \mathbb{E}[X_2|\bar{Y} = i] = \mathbb{E}[X_2|\mu = e_i]
\end{aligned}$$

To show $\epsilon_{\text{CI}} = 0$, from Definition 4.1 we need to show $\mathbb{E}[X_2|X_1] = \mathbb{E}[\mathbb{E}[X_2|\bar{Y}]|X_1]$. Since $X_2$ is the bag of words representation, we know that $X_2 = \frac{2}{N}\sum_{i=N/2+1}^{N} e_{w_i}$. So for any $\mu \in \Delta_{[k]}$ we get

$$\mathbb{E}[X_2|\mu] =^{(a)} \frac{2}{N}\sum_{i=N/2+1}^{N} \mathbb{E}[e_{w_i}|\mu] =^{(b)} \frac{2}{N}\sum_{i=N/2+1}^{N} A\mu = A\mu$$

where $(a)$ follows from linearity of expectation and $(b)$ follows from the linearity of the probability distribution of each word given $\mu$ for topic models. Thus from the definition of $\bar{Y}$, $\mathbb{E}\left[X_2|\bar{Y}=i\right] = \mathbb{E}\left[X_2|\mu=e_i\right] = Ae_i$. To check if $\epsilon_{\text{CI}} = 0$, we compute the following

$$
\begin{aligned}
\mathbb{E}\left[\mathbb{E}\left[X_2|\bar{Y}\right]|X_1\right] &= \sum_{i=1}^{k} \mathbb{E}\left[X_2|\bar{Y}=i\right] P(\bar{Y}=i|X_1) \\
&= \sum_{i=1}^{k} Ae_i\, \mathbb{E}\left[\mu(i)|X_1\right] = A \sum_{i=1}^{k} \mathbb{E}\left[\mu(i)e_i|X_1\right] \\
&= \mathbb{E}\left[A\mu|X_1\right] = \mathbb{E}\left[\mathbb{E}\left[X_2|\mu\right]|X_1\right]
\end{aligned}
$$

Due to the topic modeling assumption and the independent sampling of words given $\mu$, we know that $X_1 \perp X_2|\mu$ and thus $\mathbb{E}\left[X_2|X_1\right] = \mathbb{E}\left[\mathbb{E}\left[X_2|\mu\right]|X_1\right]$. Combining with the above calculation, we get that $\mathbb{E}\left[\mathbb{E}\left[X_2|\bar{Y}\right]|X_1\right] = \mathbb{E}\left[X_2|X_1\right]$, thus giving $\epsilon_{\text{CI}} = 0$. This proves points 1. and 2.

For point 3., note that $\mathbb{E}[Y|X_1] = \mathbb{E}[w^\top\mu|X_1] = w^\top\mathbb{E}[\mu|X_1] = w^\top\mathbb{E}[\bar{Y}|X_1]$.

Finally for point 4., we use the definition $1/\beta = \|\mathbf{\Sigma}_{Y\phi_{\bar{y}}}\mathbf{\Sigma}_{X_2\phi_{\bar{y}}}^\dagger\|_2$. For the first term, we note that $\mathbb{E}\left[\phi_{\bar{Y}}|\mu\right] = \mathbb{E}\left[\mathbb{E}\left[\phi_{\bar{Y}}|X_1\right]|\mu\right] = \mathbb{E}\left[\mathbb{E}\left[\bar{\mu}|X_1\right]|\mu\right] = \mu$

$$
\begin{aligned}
\mathbf{\Sigma}_{Y\phi_{\bar{y}}} &= \mathbb{E}_{\mu\sim\tau}\left[Y\phi_{\bar{Y}}^\top\right] = \mathbb{E}_{\mu\sim\tau}\left[w^\top\mu\phi_{\bar{Y}}^\top\right] \\
&= \mathbb{E}_{\mu\sim\tau}\left[w^\top\mu\mathbb{E}\left[\phi_{\bar{Y}}^\top|\mu\right]\right] = \mathbb{E}_{\mu\sim\tau}\left[w^\top\mu\mu^\top\right] \\
&= w^\top\Gamma
\end{aligned}
$$

where $\Gamma$ was defined as the topic covariance $\Gamma = \mathbb{E}_{\mu\sim\tau}\left[\mu\mu^\top\right]$. The second term is

$$
\mathbf{\Sigma}_{X_2\phi_{\bar{y}}} = \mathbb{E}_{\mu\sim\tau}\left[\mathbb{E}\left[X_2|\mu\right]\mathbb{E}\left[\phi_{\bar{Y}}^\top|\mu\right]\right] = \mathbb{E}_{\mu\sim\tau}\left[A\mu\mu^\top\right] = A\Gamma
$$

The upper bound for $1/\beta$ can be computed as follows

$$
\begin{aligned}
1/\beta &= \left\|\mathbf{\Sigma}_{Y\phi_{\bar{y}}}\mathbf{\Sigma}_{X_2\phi_{\bar{y}}}^\dagger\right\|_2 = \left\|w^\top\Gamma\left(A\Gamma\right)^\dagger\right\|_2 \\
&\leq \|w\|_2\,\lambda_{\max}(\Gamma)\,\lambda_{\max}\left(\left(A\Gamma\right)^\dagger\right) = \|w\|_2\,\lambda_{\max}(\Gamma)\,\lambda_{\min}\left(A\Gamma\right)^{-1} \\
&\leq \|w\|_2\,\lambda_{\max}(\Gamma)\,\lambda_{\min}\left(A\right)^{-1}\,\lambda_{\min}\left(\Gamma\right)^{-1} \\
&= \|w\|_2\,\frac{\lambda_{\max}(\Gamma)}{\lambda_{\min}\left(\Gamma\right)^{-1}}\,\lambda_{\min}\left(A\right)^{-1} = \frac{\kappa\|w\|_2}{\lambda_{\min}\left(A\right)}
\end{aligned}
$$

For $\lambda_{\min}\left(A\right)$, we need to lower bound $\min_{\|v\|_2=1}\|Av\|_2$ for which we will use the anchor word assumption. Let $\pi(i)$ be the anchor word for topic $i \in [k]$; so $A_i(\pi(i)) \geq p$ and $A_{i'}(\pi(i)) = 0$ for $i' \neq i$. Using the assumption, for any $v \in \mathbb{R}^k$ with $\|v\|_2 = 1$, we get that $|(Av)(\pi(i))| = |v_i|\,A_i(\pi(i)) \geq p|v_i|$ and thus $\|Av\|_2^2 \geq \sum_i (Av)(\pi(i))^2 \geq p^2\sum_i v_i^2 = p^2$. This shows that $\lambda_{\min}\left(A\right) \geq p$, thus combining with the above calculation to prove point 4. and completing the proof. $\qquad\square$

# F General Results and Comparison to [57]

We now show a more general form of our results and also connect the multi-view redundancy assumption from [57] to ours.

## F.1 General Results

We first note that all our results hold for a generalized version of Assumption 4.1 and Definition 4.1 that we state below.

**Assumption F.1.** *Suppose $\bar{Y}$ with $|\bar{Y}| \leq m$ is a discrete latent variable that satisfies*

    *1. $\bar{Y}$ makes $X_1$ and $X_2$ approximately CI as in Definition 4.1, i.e.*

$$
\epsilon_{CI}^2 := \mathbb{E}_{X_1}\left[\|\mathbb{E}[X_2|X_1] - \mathbb{E}_{\bar{Y}}[\mathbb{E}[X_2|\bar{Y}]|X_1]\|^2\right]
$$

2. $\bar{Y}$ also makes $X_1$ and $Y$ approximately CI with

$$\epsilon_{\bar{Y}}^2 := \mathbb{E}_{X_1}\left[\|\mathbb{E}[Y|X_1] - \mathbb{E}_{\bar{Y}}[\mathbb{E}[Y|\bar{Y}]|X_1]\|^2\right]$$

3. $\mathbf{\Sigma}_{\phi_{\bar{y}}X_2}$ is full column rank and $\|\mathbf{\Sigma}_{Y\phi_{\bar{y}}}\mathbf{\Sigma}_{X_2\phi_{\bar{y}}}^{\dagger}\|_2 = 1/\beta$, where $A^{\dagger}$ is pseudo-inverse, and $\phi_{\bar{y}}$ is the one-hot embedding for $\bar{Y}$.

Note that our assumptions from the main paper are a special case of Assumption F.1, with $\epsilon_{\bar{Y}} = 0$ being satisfied automatically as $\bar{Y} = [Y, Z]$ is explicitly defined to contain $Y$ in it. Unlike Assumption 4.1, we do not need $Y$ to be a discrete variable, but just need $\bar{Y}$ to be discrete. We state the generalization of Theorem 4.2 below

**Theorem F.1.** *For a fixed $\delta \in (0,1)$, under Assumptions F.1, 4.2 for $\tilde{\psi}$ and $\psi^*$ and 3.2 for non-universal feature maps, if $n_1, n_2 \gg \rho^4(d_2 + \log 1/\delta)$, and we learn the pretext tasks such that: $\mathbb{E}\|\tilde{\psi}(X_1) - \psi^*(X_1)\|_F^2 \le \epsilon_{pre}^2$. Then the generalization error for downstream task w.p. $1 - \delta$ is:*

$$\mathbb{E}_{X_1}\left[\|\mathbb{E}[Y|X_1] - \hat{\mathbf{W}}^{\top}\tilde{\psi}(X_1)\|_2^2\right] \le \tilde{\mathcal{O}}\left(\sigma^2\frac{d_2}{n_2} + \frac{\epsilon_{CI}^2}{\beta^2} + \frac{\epsilon_{pre}^2}{\beta^2} + \epsilon_{\bar{Y}}^2\right) \tag{17}$$

The result is pretty much the same as Theorem 4.2, except for an additional term of $\epsilon_{\bar{Y}}^2$. The proof is also very similar, the difference being that $\mathbb{E}[\mathbb{E}[Y|\bar{Y}]|X_1]$ can now be expressed as a linear function of $\psi^*$ instead of $\mathbb{E}[Y|X_1]$, and the additional error incurred during to the mismatch between $\mathbb{E}[Y|X_1]$ and $\mathbb{E}[\mathbb{E}[Y|\bar{Y}]|X_1]$ that is $\epsilon_{\bar{Y}}^2$ will be incurred.

## F.2 Comparison to [57]

We show guarantees for our algorithm under the assumption from [57] in the following special case that satisfies: (1) $X_1$ and $X_2$ are *exactly* CI given $\bar{Y}$ (thus $\epsilon_{CI} = 0$), (2) the variation in the target $Y$ is small given $X_1$ and $X_2$. The assumption from [57], in our setting, is equivalent to saying that $\epsilon_{X_1}$ and $\epsilon_{X_2}$ are small, where

$$\epsilon_{X_i}^2 = \mathbb{E}\left[\|\mathbb{E}[Y|X_i] - \mathbb{E}[Y|X_1, X_2]\|^2\right], \quad i \in \{1, 2\}$$

A similar assumption of multi-view redundancy also appears in [58]; however they state it in terms of information-theoretic quantities instead. We will show that these assumptions are also almost sufficient to show results in our setting. In particular we show that if $Y|X_1, X_2$ is almost deterministic (which makes sense for a many regression tasks) and if $\epsilon_{X_2}^2$ is small, then $\epsilon_{\bar{Y}}$ defined in the previous subsection will be small and thus we have meaningful guarantees.

**Lemma F.2.** *Let $\sigma_Y^2 = Var[Y|X_1, X_2]$ be the variance of $Y$. $\bar{Y}$ is as defined in Assumption F.1 with the extra condition that $X_1$ and $X_2$ are exactly CI given $\bar{Y}$. Then we have*

$$\epsilon_{\bar{Y}} \le \sqrt{2}(\sigma_Y + \epsilon_{X_2})$$

Plugging this into Theorem F.1 will give us the desired result. Note however that we did not even use the fact that $\epsilon_{X_1}$ is small. Using this part of the assumption, we can get an even stronger result that shows that even though our learned representation will only $X_1$, if will still predict $Y|X_1, X_2$ well.

**Corollary F.3.** *For a fixed $\delta \in (0,1)$, under Assumptions F.1, 4.2 for $\tilde{\psi}$ and $\psi^*$ and 3.2 for non-universal feature maps, if $n_1, n_2 \gg \rho^4(d_2 + \log 1/\delta)$, and we learn the pretext tasks such that: $\mathbb{E}\|\tilde{\psi}(X_1) - \psi^*(X_1)\|_F^2 \le \epsilon_{pre}^2$. Then the generalization error for downstream task w.p. $1 - \delta$ is:*

$$\mathbb{E}_{X_1, X_2}\left[\|\mathbb{E}[Y|X_1, X_2] - \hat{\mathbf{W}}^{\top}\tilde{\psi}(X_1)\|_2^2\right] \le \tilde{\mathcal{O}}\left(\sigma^2\frac{d_2}{n_2} + \frac{\epsilon_{pre}^2}{\beta^2} + \epsilon_{X_1}^2 + \epsilon_{X_2}^2 + \sigma_Y^2\right)$$

Thus we see that the assumption from [57] is strong enough for us to be able to show stronger results than just our assumption. We complete this section by proving Lemma F.2

*Lemma F.2.* We will also make use of the following lemma that is easily proved using Cauchy-Schwarz inequality

**Lemma F.4.** *For random variables $Z_1, \ldots, Z_n$ for which $\mathbb{E}[\|Z_i\|^2] < \infty$ for every $i \in [n]$, we have*

$$\mathbb{E}[\|Z_1 + \cdots + Z_n\|^2] \leq \left( \sqrt{\mathbb{E}[\|Z_1\|^2]} + \cdots + \sqrt{\mathbb{E}[\|Z_n\|^2]} \right)^2$$

The proof follows from the following sequence of inequalities that uses Jensen's inequality, conditional independence of $X_1$ and $X_2$ and the above lemma. For simplicity we assume that $Y$ is a scalar random variable, the proof is the same for vector values $Y$, except squared values will replaced by norm squared values.

$$
\begin{aligned}
\epsilon_{\bar{Y}}^2 &= \mathbb{E}_{X_1} \left[ (\mathbb{E}[Y|X_1] - \mathbb{E}_{\bar{Y}}[\mathbb{E}[Y|\bar{Y}]|X_1])^2 \right] = \mathbb{E}_{X_1} \left[ (\mathbb{E}_{\bar{Y}}[\mathbb{E}[Y|\bar{Y}, X_1]|X_1] - \mathbb{E}_{\bar{Y}}[\mathbb{E}[Y|\bar{Y}]|X_1])^2 \right] \\
&\leq \mathbb{E}_{X_1, \bar{Y}} \left[ (\mathbb{E}[Y|X_1, \bar{Y}] - \mathbb{E}[Y|\bar{Y}])^2 \right] \\
&= \mathbb{E}_{\bar{Y}} \mathbb{E}_{X_1|\bar{Y}} \mathbb{E}_{X_1'|\bar{Y}} \left[ (\mathbb{E}[Y|X_1, \bar{Y}] - \mathbb{E}[Y|X_1', \bar{Y}])^2 \right] \\
&= \frac{1}{2} \mathbb{E}_{\bar{Y}} \mathbb{E}_{X_1|\bar{Y}} \mathbb{E}_{X_1'|\bar{Y}} \left[ (\mathbb{E}_{X_2}[\mathbb{E}[Y|X_1, X_2, \bar{Y}]|\bar{Y}] - \mathbb{E}_{X_2}[\mathbb{E}[Y|X_1', X_2, \bar{Y}]|\bar{Y}])^2 \right] \\
&\leq \frac{1}{2} \mathbb{E}_{\bar{Y}} \mathbb{E}_{X_1|\bar{Y}} \mathbb{E}_{X_1'|\bar{Y}} \mathbb{E}_{X_2|\bar{Y}} \left[ (\mathbb{E}[Y|X_1, X_2, \bar{Y}] - \mathbb{E}[Y|X_1', X_2, \bar{Y}])^2 \right] \\
&= \frac{1}{2} \mathbb{E} \left[ (Z_1 + Z_2 + Z_3 + Z_4)^2 \right]
\end{aligned}
$$

where $Z_1 = \mathbb{E}[Y|X_1, X_2, \bar{Y}] - \mathbb{E}[Y|X_1, X_2]$, $Z_2 = -\mathbb{E}[Y|X_1', X_2, \bar{Y}] + \mathbb{E}[Y|X_1', X_2]$, $Z_3 = \mathbb{E}[Y|X_1, X_2] - \mathbb{E}[Y|X_2]$ and $Z_4 = -\mathbb{E}[Y|X_1', X_2] + \mathbb{E}[Y|X_2]$. The first and third inequality follow from Jensen's inequality, second inequality follows from $\mathbb{E}[(X - \mathbb{E}[X])^2] = \frac{1}{2}\mathbb{E}[(X - X')^2]$, and the third equality follows from the CI assumption.

We will bound $\mathbb{E}[Z_1^2] = \mathbb{E}[Z_2^2] \leq \mathbb{E}[(\mathbb{E}[Y|X_1, X_2, \bar{Y}] - \mathbb{E}[Y|X_1, X_2])^2] \leq \mathbb{E}[(Y - \mathbb{E}[Y|X_1, X_2])^2] = \sigma_Y^2$ again from Jensen's inequality. $Z_3$ and $Z_4$ can be handled by observing that $\mathbb{E}[Z_3^2] = \mathbb{E}[Z_4^2] = \mathbb{E}[(\mathbb{E}[Y|X_1, X_2] - \mathbb{E}[Y|X_2])^2] = \epsilon_{X_2}^2$.

Thus using the above lemma, we get the desired upper bound on $\epsilon_{\bar{Y}}$. $\qquad\square$

# G   Showing $\mathbb{E}[Y|X_1] \approx \mathbb{E}[Y|X_1, X_2]$

Our main result Theorem 4.2 shows that self-supervised learning can help approximate $\mathbb{E}[Y|X_1]$ as a linear function of the learned features $\tilde{\psi}$. In practice, however, it is more common to predict the label $Y$ using the entire input $X = (X_1, X_2)$ rather than just $X_1$. We show here that learning $\mathbb{E}[Y|X_1]$ is sufficient, under mild assumptions on the task being solved: the Bayes error of the classification task $(X_1, Y)$ is low. We first upper bound the discrepancy between $\mathbb{E}[Y|X_1]$ and $\mathbb{E}[Y|X_1, X_2]$ based on the Bayes error rate.

**Lemma G.1.** *Suppose $\|Y\| \leq 1$ and $k = |\mathcal{Y}|$. Denote the Bayes error for distribution $P_{X_1, Y}$ to be Bayes-error$(P_{X_1, Y}) = \mathbb{E}_{X_1} [1 - \max_y P(y|X_1)]$[9]. Then we have*

$$\mathbb{E}_{X_1, X_2} \left[ \|\mathbb{E}[Y|X_1] - \mathbb{E}[Y|X_1, X_2]\|^2 \right] \leq 2k \, \text{Bayes-error}(P_{X_1, Y})$$

We will show below (for $\mathcal{H} = \mathcal{H}_u$) that if $P_{X_1, Y}$ has low Bayes error, then predicting $\mathbb{E}[Y|X_1]$ is as good as predicting $\mathbb{E}[Y|X_1, X_2]$ up to this small additive error.

**Theorem G.2.** *Suppose $\epsilon_{Bayes} = \text{Bayes-error}(P_{X_1, Y})$ and that $\tilde{\psi}$ is $\epsilon_{pre}^2$-optimal on the SSL task (as in Theorem 4.2). Under the same conditions as Theorem 4.2, with probability $1 - \delta$ we have*

$$\mathbb{E}_{X_1, X_2} \left[ \|\mathbb{E}[Y|X_1, X_2] - \hat{W}^\top \tilde{\psi}(X_1)\|_2^2 \right] \leq \tilde{\mathcal{O}} \left( \sigma^2 \frac{d_2}{n_2} + \frac{\epsilon_{CI}^2}{\beta^2} + \frac{\epsilon_{pre}^2}{\beta^2} \right) + 2\epsilon_{Bayes}$$

*Proof.* The law of total expectation gives $\mathbb{E}_{X_2}[\mathbb{E}[Y|X_1, X_2]|X_1] = \mathbb{E}[Y|X_1]$, thus it is easy to obtain the following decomposition

$$
\begin{aligned}
\mathbb{E}_{X_1, X_2} \left[ \|\mathbb{E}[Y|X_1, X_2] - \hat{W}^\top \tilde{\psi}(X_1)\|_2^2 \right] =& \mathbb{E}_{X_1} \left[ \|\mathbb{E}[Y|X_1] - \hat{W}^\top \tilde{\psi}(X_1)\|_2^2 \right] \\
&+ \mathbb{E}_{X_1, X_2} \left[ \|\mathbb{E}[Y|X_1] - \mathbb{E}[Y|X_1, X_2]\|_2^2 \right]
\end{aligned}
$$

---

[9]We abuse notation and use $P(y|X_1)$ instead of $P_{X_1, Y}(y|X_1)$.

The first term can be upper bounded using Theorem 4.2: $\mathbb{E}_{X_1}\left[\|\mathbb{E}[Y|X_1] - \hat{\boldsymbol{W}}^\top \tilde{\psi}(X_1)\|_2^2\right] = $ $\mathrm{ER}_{\tilde{\psi}}(\hat{\boldsymbol{W}}) \leq \tilde{\mathcal{O}}\left(\sigma^2 \frac{d_2}{n_2} + \frac{\epsilon_{\mathrm{Cl}}^2}{\beta^2} + \frac{\epsilon_{\mathrm{pre}}^2}{\beta^2}\right)$. The second term is upper bounded by $2\epsilon_{\mathrm{Bayes}}$ by invoking Lemma G.1, and this completes the proof $\qquad\square$

*Proof of Lemma G.1.* Notice the following inequality

$$\mathbb{E}_{X_1, X_2}\left[\|\mathbb{E}[Y|X_1] - \mathbb{E}[Y|X_1, X_2]\|^2\right] = \mathbb{E}_{X_1, X_2}\left[\left\|\sum_{y \in \mathcal{Y}} y\left(P(y|X_1) - P(y|X_1, X_2)\right)\right\|^2\right]$$

$$\leq |\mathcal{Y}|(\max_y \|y\|^2)\mathbb{E}_{X_1, X_2}\left[\sum_y \left(P(y|X_1) - P(y|X_1, X_2)\right)^2\right]$$

$$\leq k\mathbb{E}_{X_1}\left[\mathbb{E}_{X_2}\left[\sum_y \left(P(y|X_1) - P(y|X_1, X_2)\right)^2 \mid X_1\right]\right]$$

where the first inequality follows from Cauchy-Schwartz and second inequality follows from $\|Y\| \leq 1$. Thus the problem reduces to bounding the inner expectation for every $X_1$. We first note that for every $X_1, y$, we have $P(y|X_1) = \mathbb{E}_{X_2}[P(y|X_1, X_2)|X_1]$ from the law of total expectation. This gives

$$\mathbb{E}_{X_2}\left[\sum_y \left(P(y|X_1) - P(y|X_1, X_2)\right)^2 \mid X_1\right] = \sum_y \mathbb{E}_{X_2}\left[P(y|X_1, X_2)^2|X_1\right] - P(y|X_1)^2$$

$$\leq \sum_y \mathbb{E}_{X_2}\left[P(y|X_1, X_2)|X_1\right] - P(y|X_1)^2 = \mathbb{E}_{X_2}\left[\sum_y P(y|X_1, X_2)|X_1\right] - \sum_y P(y|X_1)^2$$

$$= 1 - \sum_y P(y|X_1)^2 \leq 1 - \max_y P(y|X_1)^2 \leq 2(1 - \max_y P(y|X_1))$$

where the first inequality follows because $P(y|X_1, X_2) \in [0, 1]$ and second follows trivially and third follows from $1 - x^2 \leq 2(1 - x)$ for $x \in [0, 1]$. Combining everything, we get $\mathbb{E}_{X_1, X_2}\left[\|\mathbb{E}[Y|X_1] - \mathbb{E}[Y|X_1, X_2]\|^2\right] \leq 2k\mathbb{E}_{X_1}\left[1 - \max_y P(y|X_1)\right] = 2k \text{ Bayes-error}(P_{X_1, Y})$, thus proving the result. $\qquad\square$

# H  Theoretical analysis for classification tasks

## H.1  Classification tasks

We now consider the benefit of learning $\psi$ from a class $\mathcal{H}_1$ on linear classification task for label set $\mathcal{Y} = [k]$. The performance of a classifier is measured using the standard logistic loss

**Definition H.1.** *For a task with $\mathcal{Y} = [k]$, classification loss for a predictor $f : \mathcal{X}_1 \to \mathbb{R}^k$ is*

$$\ell_{clf}(f) = \mathbb{E}[\ell_{log}(f(X_1), Y)], \text{ where } \ell_{log}(\hat{y}, y) = \left[-\log\left(\frac{e^{\hat{y}_y}}{\sum_{y'} e^{\hat{y}_{y'}}}\right)\right]$$

*The loss for representation $\psi : \mathcal{X}_1 \to \mathbb{R}^{d_1}$ and linear classifier $\boldsymbol{W} \in \mathbb{R}^{k \times d_1}$ is denoted by $\ell_{clf}(\boldsymbol{W}\psi)$.*

We note that the function $\ell_{\log}$ is 1-Lipschitz in the first argument. The result will also hold for the hinge loss $\ell_{\mathrm{hinge}}(\hat{y}, y) = (1 - \hat{y}_y + \max_{y' \neq y} \hat{y}_{y'})_+$ which is also 1-Lipschitz, instead of $\ell_{\log}$.

We assume that the optimal regressor $f_{\mathcal{H}_1}^*$ for one-hot encoding also does well on linear classification.

**Assumption H.1.** *The best regressor for 1-hot encodings in $\mathcal{H}_1$ does well on classification, i.e. $\ell_{clf}(\gamma f_{\mathcal{H}_1}^*) \leq \epsilon_{one\text{-}hot}$ is small for some scalar $\gamma$.*

**Remark H.1.** *Note that if $\mathcal{H}_1$ is universal, then $f_{\mathcal{H}_1}^*(\boldsymbol{x}_1) = \mathbb{E}[Y|X_1 = \boldsymbol{x}_1]$ and we know that $f_{\mathcal{H}_1}^*$ is the Bayes-optimal predictor for binary classification. In general one can potentially predict the label by looking at $\arg\max_{i \in [k]} f_{\mathcal{H}_1}^*(\boldsymbol{x}_1)_i$. The scalar $\gamma$ captures the margin in the predictor $f_{\mathcal{H}_1}^*$.*

We now show that using the classifier $\hat{\boldsymbol{W}}$ obtained from linear regression on one-hot encoding with learned representations $\tilde{\psi}$ will also be good on linear classification. The proof is in Section H

**Theorem H.2.** *For a fixed $\delta \in (0,1)$, under the same setting as Theorem 4.2 and Assumption H.1, we have:*

$$\ell_{\textit{clf}}\left(\gamma \hat{\boldsymbol{W}} \tilde{\psi}\right) \leq \tilde{\mathcal{O}}\left(\gamma \sqrt{\sigma^2 \frac{d_2}{n_2} + \frac{\epsilon^2}{\beta^2} + \frac{\epsilon_{\textit{pre}}^2}{\beta^2}}\right) + \epsilon_{\textit{one-hot}},$$

*with probability $1 - \delta$.*

*Proof of Theorem H.2.* We simply follow the following sequence of steps

$$\begin{aligned}
\ell_{\text{clf}}\left(\gamma \hat{\boldsymbol{W}} \tilde{\psi}\right) &= \mathbb{E}[\ell_{\log}\left(\gamma \hat{\boldsymbol{W}} \tilde{\psi}(X_1), Y\right)] \\
&\leq^{(a)} \mathbb{E}\left[\ell_{\log}\left(\gamma f^*_{\mathcal{H}_1}(X_1), Y\right) + \gamma \|\hat{\boldsymbol{W}} \tilde{\psi}(X_1) - f^*_{\mathcal{H}_1}(X_1)\|\right] \\
&\leq^{(b)} \epsilon_{\text{one-hot}} + \gamma \sqrt{\mathbb{E}\left[\|\hat{\boldsymbol{W}} \tilde{\psi}(X_1) - f^*_{\mathcal{H}_1}(X_1)\|^2\right]} \\
&= \epsilon_{\text{one-hot}} + \gamma \sqrt{\text{ER}_{\tilde{\psi}}[\hat{\boldsymbol{W}}]}
\end{aligned}$$

where $(a)$ follows because $\ell_{\log}$ is 1-Lipschitz and $(b)$ follows from Assumption H.1 and Jensen's inequality. Plugging in Theorem 4.2 completes the proof. $\square$

# I   Four Different Ways to Use CI

In this section we propose four different ways to use conditional independence to prove zero approximation error, i.e.,

**Claim I.1** (informal)**.** *When conditional independence is satisfied: $X_1 \perp X_2 | Y$, and some non-degeneracy is satisfied, there exists some matrix $\boldsymbol{W}$ such that $\mathbb{E}[Y|X_1] = \boldsymbol{W}\mathbb{E}[X_2|X_1]$.*

We note that for simplicity, most of the results are presented for the jointly Gaussian case, where everything could be captured by linear conditional expectation $\mathbb{E}^L[Y|X_1]$ or the covariance matrices. When generalizing the results for other random variables, we note just replace $X_1, X_2, Y$ by $\phi_1(X_1), \phi_2(X_2), \phi_y(Y)$ will suffice the same arguments.

## I.1   Inverse Covariance Matrix

Write $\boldsymbol{\Sigma}$ as the covariance matrix for the joint distribution $P_{X_1 X_2 Y}$.

$$\boldsymbol{\Sigma} = \begin{bmatrix} \boldsymbol{\Sigma}_{XX} & \boldsymbol{\Sigma}_{XY} \\ \boldsymbol{\Sigma}_{YY}^\top & \boldsymbol{\Sigma}_{YY} \end{bmatrix}, \quad \boldsymbol{\Sigma}^{-1} = \begin{bmatrix} \boldsymbol{A} & \rho \\ \rho^\top & \boldsymbol{B} \end{bmatrix}$$

where $\boldsymbol{A} \in \mathbb{R}^{(d_1+d_2)\times(d_1+d_2)}, \rho \in \mathbb{R}^{(d_1+d_2)\times k}, \boldsymbol{B} \in \mathbb{R}^{k \times k}$. Furthermore

$$\rho = \begin{bmatrix} \rho_1 \\ \rho_2 \end{bmatrix}; \quad \boldsymbol{A} = \begin{bmatrix} \boldsymbol{A}_{11} & \boldsymbol{A}_{12} \\ \boldsymbol{A}_{21} & \boldsymbol{A}_{22} \end{bmatrix}$$

for $\rho_i \in \mathbb{R}^{d_i \times k}, i = 1, 2$ and $\boldsymbol{A}_{ij} \in \mathbb{R}^{d_i \times d_j}$ for $i, j \in \{1, 2\}$.

**Claim I.2.** *When conditional independence is satisfied, $\boldsymbol{A}$ is block diagonal matrix, i.e., $\boldsymbol{A}_{12}$ and $\boldsymbol{A}_{21}$ are zero matrices.*

**Lemma I.3.** *We have the following*

$$\mathbb{E}[X_1|X_2] = (\boldsymbol{A}_{11} - \bar{\rho}_1 \bar{\rho}_1^\top)^{-1}(\bar{\rho}_1 \bar{\rho}_2^\top - \boldsymbol{A}_{12})X_2 \tag{18}$$

$$\mathbb{E}[X_2|X_1] = (\boldsymbol{A}_{22} - \bar{\rho}_2 \bar{\rho}_2^\top)^{-1}(\bar{\rho}_2 \bar{\rho}_1^\top - \boldsymbol{A}_{21})X_1 \tag{19}$$

$$\mathbb{E}[Y|X] = -B^{-\frac{1}{2}}(\bar{\rho}_1^\top X_1 + \bar{\rho}_2^\top X_2) \tag{20}$$

*where $\bar{\rho}_i = \rho_i \boldsymbol{B}^{-\frac{1}{2}}$ for $i \in \{1, 2\}$. Also,*

$$(\boldsymbol{A}_{11} - \bar{\rho}_1 \bar{\rho}_1^\top)^{-1} \bar{\rho}_1 \bar{\rho}_2^\top = \frac{1}{1 - \bar{\rho}_1^\top \boldsymbol{A}_{11}^{-1} \bar{\rho}_1} \boldsymbol{A}_{11}^{-1} \bar{\rho}_1 \bar{\rho}_2^\top$$

$$(\boldsymbol{A}_{22} - \bar{\rho}_2 \bar{\rho}_2^\top)^{-1} \bar{\rho}_2 \bar{\rho}_1^\top = \frac{1}{1 - \bar{\rho}_2^\top \boldsymbol{A}_{22}^{-1} \bar{\rho}_2} \boldsymbol{A}_{22}^{-1} \bar{\rho}_2 \bar{\rho}_1^\top$$

*Proof.* We know that $\mathbb{E}[X_1|X_2] = \mathbf{\Sigma}_{12}\mathbf{\Sigma}_{22}^{-1}X_2$ and $\mathbb{E}[X_2|X_1] = \mathbf{\Sigma}_{21}\mathbf{\Sigma}_{11}^{-1}x_1$, where

$$\mathbf{\Sigma}_{XX} = \begin{bmatrix} \mathbf{\Sigma}_{11} & \mathbf{\Sigma}_{12} \\ \mathbf{\Sigma}_{21} & \mathbf{\Sigma}_{22} \end{bmatrix}$$

First using $\mathbf{\Sigma}\mathbf{\Sigma}^{-1} = I$, we get the following identities

$$\mathbf{\Sigma}_{XX}\mathbf{A} + \mathbf{\Sigma}_{XY}\rho^\top = \mathbf{I} \tag{21}$$

$$\mathbf{\Sigma}_{XY}^\top \mathbf{A} + \mathbf{\Sigma}_{YY}\rho^\top = 0 \tag{22}$$

$$\mathbf{\Sigma}_{XX}\rho + \mathbf{\Sigma}_{XY}\mathbf{B} = 0 \tag{23}$$

$$\mathbf{\Sigma}_{XY}^\top \rho + \mathbf{\Sigma}_{YY}\mathbf{B} = \mathbf{I} \tag{24}$$

From Equation (23) we get that $\mathbf{\Sigma}_{XY} = -\mathbf{\Sigma}_{XX}\rho\mathbf{B}^{-1}$ and plugging this into Equation (21) we get

$$\mathbf{\Sigma}_{XX}\mathbf{A} - \mathbf{\Sigma}_{XX}\rho\mathbf{B}^{-1}\rho^\top = \mathbf{I}$$

$$\implies \mathbf{\Sigma}_{XX} = (\mathbf{A} - \rho\mathbf{B}^{-1}\rho^\top)^{-1} = (\mathbf{A} - \bar{\rho}\bar{\rho}^\top)^{-1}$$

$$\implies \begin{bmatrix} \mathbf{\Sigma}_{11} & \mathbf{\Sigma}_{12} \\ \mathbf{\Sigma}_{21} & \mathbf{\Sigma}_{22} \end{bmatrix} = \left( \begin{bmatrix} \mathbf{A}_{11} - \bar{\rho}_1\bar{\rho}_1^\top & \mathbf{A}_{12} - \bar{\rho}_1\bar{\rho}_2^\top \\ \mathbf{A}_{21} - \bar{\rho}_2\bar{\rho}_1^\top & \mathbf{A}_{22} - \bar{\rho}_2\bar{\rho}_2^\top \end{bmatrix} \right)^{-1}$$

We now make use of the following expression for inverse of a matrix that uses Schur complement: $\mathbf{M}/\alpha = \delta - \gamma\alpha^{-1}\beta$ is the Schur complement of $\alpha$ for $\mathbf{M}$ defined below

$$\text{If } \mathbf{M} = \begin{bmatrix} \alpha & \beta \\ \gamma & \delta \end{bmatrix}, \text{ then, } \mathbf{M}^{-1} = \begin{bmatrix} \alpha^{-1} + \alpha^{-1}\beta(\mathbf{M}/\alpha)^{-1}\gamma\alpha^{-1} & -\alpha^{-1}\beta(\mathbf{M}/\alpha)^{-1} \\ -(\mathbf{M}/\alpha)^{-1}\gamma\alpha^{-1} & (\mathbf{M}/\alpha)^{-1} \end{bmatrix}$$

For $\mathbf{M} = (\mathbf{A} - \bar{\rho}\bar{\rho}^\top)$, we have that $\mathbf{\Sigma}_{XX} = \mathbf{M}^{-1}$ and thus

$$\mathbf{\Sigma}_{12}\mathbf{\Sigma}_{22}^{-1} = -\alpha^{-1}\beta(\mathbf{M}/\alpha)^{-1}((\mathbf{M}/\alpha)^{-1})^{-1}$$

$$= -\alpha^{-1}\beta$$

$$= (\mathbf{A}_{11} - \bar{\rho}_1\bar{\rho}_1^\top)^{-1}(\bar{\rho}_1\bar{\rho}_2^\top - \mathbf{A}_{12})$$

This proves Equation (18) and similarly Equation (19) can be proved.

For Equation (20), we know that $\mathbb{E}[Y|X = (X_1, X_2)] = \mathbf{\Sigma}_{YX}\mathbf{\Sigma}_{XX}^{-1}X = \mathbf{\Sigma}_{XY}^\top\mathbf{\Sigma}_{XX}^{-1}X$. By using Equation (23) we get $\mathbf{\Sigma}_{XY} = -\mathbf{\Sigma}_{XX}\rho\mathbf{B}^{-1}$ and thus

$$\mathbb{E}[Y|X = (X_1, X_2)] = -\mathbf{B}^{-1}\rho^\top\mathbf{\Sigma}_{XX}\mathbf{\Sigma}_{XX}^{-1}X$$

$$= -\mathbf{B}^{-1}\rho^\top X = \mathbf{B}^{-1}(\rho_1^\top X_1 + \rho_2^\top X_2)$$

$$= -\mathbf{B}^{-\frac{1}{2}}(\bar{\rho}_1^\top X_1 + \bar{\rho}_2^\top X_2)$$

For the second part, we will use the fact that $(\mathbf{I} - \mathbf{a}\mathbf{b}^\top)^{-1} = \mathbf{I} + \frac{1}{1-\mathbf{a}^\top\mathbf{b}}\mathbf{a}\mathbf{b}^\top$. Thus

$$(\mathbf{A}_{11} - \bar{\rho}_1\bar{\rho}_1^\top)^{-1}\bar{\rho}_1\bar{\rho}_2 = (\mathbf{I} - \mathbf{A}_{11}^{-1}\bar{\rho}_1\bar{\rho}_1^\top)\mathbf{A}_{11}^{-1}\bar{\rho}_1\bar{\rho}_2^\top$$

$$= (\mathbf{I} + \frac{1}{1 - \bar{\rho}_1^\top\mathbf{A}_{11}^{-1}\bar{\rho}_1}\mathbf{A}_{11}^{-1}\bar{\rho}_1\bar{\rho}_1)\mathbf{A}_{11}^{-1}\bar{\rho}_1\bar{\rho}_2^\top$$

$$= \mathbf{A}_{11}^{-1}(\mathbf{I} + \frac{1}{1 - \bar{\rho}_1^\top\mathbf{A}_{11}^{-1}\bar{\rho}_1}\bar{\rho}_1\bar{\rho}_1\mathbf{A}_{11}^{-1})\bar{\rho}_1\bar{\rho}_2^\top$$

$$= \mathbf{A}_{11}^{-1}(\bar{\rho}_1\bar{\rho}_2^\top + \frac{\bar{\rho}_1\mathbf{A}_{11}^{-1}\bar{\rho}_1}{1 - \bar{\rho}_1^\top\mathbf{A}_{11}^{-1}\bar{\rho}_1}\bar{\rho}_1\bar{\rho}_2^\top)$$

$$= \mathbf{A}_{11}^{-1}\bar{\rho}_1\bar{\rho}_2^\top(1 + \frac{\bar{\rho}_1\mathbf{A}_{11}^{-1}\bar{\rho}_1}{1 - \bar{\rho}_1^\top\mathbf{A}_{11}^{-1}\bar{\rho}_1})$$

$$= \frac{1}{1 - \bar{\rho}_1^\top\mathbf{A}_{11}^{-1}\bar{\rho}_1}A_{11}^{-1}\bar{\rho}_1\bar{\rho}_2^\top$$

The other statement can be proved similarly. $\qquad\square$

**Claim I.4.**

$\mathbb{E}[X_2|X_1] = (\mathbf{A}_{22} - \bar{\rho}_2\bar{\rho}_2^\top)^{-1}\bar{\rho}_2\bar{\rho}_1^\top X_1 . \mathbb{E}[Y|X_1] = -\mathbf{B}^{-1/2}\bar{\rho}_1^\top X_1 - \mathbf{B}^{-1/2}\bar{\rho}_2^\top\mathbb{E}[X_2|X_1]$
*Therefore $\mathbb{E}[Y|X_1]$ is in the same direction as $\mathbb{E}[X_2|X_1]$.*

## I.2 Closed form of Linear Conditional Expectation

Refer to Claim B.1 and proof of Lemma B.2. As this is the simplest proof we used in our paper.

## I.3 From Law of Iterated Expectation

$$
\begin{aligned}
\mathbb{E}^L[X_2|X_1] &= \mathbb{E}^L[\mathbb{E}^L[X_2|X_1,Y]|X_1] \\
&= \mathbb{E}\left[ [\boldsymbol{\Sigma}_{X_2X_1}, \boldsymbol{\Sigma}_{X_2Y}] \begin{bmatrix} \boldsymbol{\Sigma}_{X_1X_1} & \boldsymbol{\Sigma}_{X_1Y} \\ \boldsymbol{\Sigma}_{YX_1} & \boldsymbol{\Sigma}_{YY} \end{bmatrix}^{-1} \begin{bmatrix} X_1 \\ Y \end{bmatrix} \mid X_1 \right] \\
&= \boldsymbol{A}X_1 + \boldsymbol{B}\mathbb{E}^L[Y|X_1].
\end{aligned}
$$

Using block matrix inverse,

$$
\begin{aligned}
\boldsymbol{A} &= (\boldsymbol{\Sigma}_{X_2X_1} - \boldsymbol{\Sigma}_{X_2Y}\boldsymbol{\Sigma}_{YY}^{-1}\boldsymbol{\Sigma}_{YX_1})(\boldsymbol{\Sigma}_{X_1X_1} - \boldsymbol{\Sigma}_{X_1Y}\boldsymbol{\Sigma}_{YY}^{-1}\boldsymbol{\Sigma}_{YX_1})^{-1} \in \mathbb{R}^{d_2 \times d_1} \\
&= \boldsymbol{\Sigma}_{X_1X_2|Y}(\boldsymbol{\Sigma}_{X_1X_1|Y})^{-1} \\
\boldsymbol{B} &= \boldsymbol{\Sigma}_{X_2Y|X_1}(\boldsymbol{\Sigma}_{YY|X_1})^{-1} \in \mathbb{R}^{d_2 \times \mathcal{Y}}.
\end{aligned}
$$

Therefore in general (without conditional independence assumption) our learned representation will be $\psi(x_1) = \boldsymbol{A}x_1 + \boldsymbol{B}f^*(x_1)$, where $f^*(\cdot) := \mathbb{E}^L[Y|X_1]$.

It's easy to see that to learn $f^*$ from representation $\psi$, we need $A$ to have some good property, such as light tail in eigenspace, and $B$ needs to be full rank in its column space.

Notice in the case of conditional independence, $\boldsymbol{\Sigma}_{X_1X_2|Y} = 0$, and $A = 0$. Therefore we could easily learn $f^*$ from $\psi$ if $X_2$ has enough information of $Y$ such that $\boldsymbol{\Sigma}_{X_2Y|X_1}$ is of the same rank as dimension of $Y$.

## I.4 From $\mathbb{E}[X_2|X_1,Y] = \mathbb{E}[X_2|Y]$

*Proof.* Let the representation function $\psi$ be defined as follows, and let we use law of iterated expectation:

$$
\begin{aligned}
\psi(\cdot) := \mathbb{E}[X_2|X_1] &= \mathbb{E}[\mathbb{E}[X_2|X_1,Y]|X_1] \\
&= \mathbb{E}[\mathbb{E}[X_2|Y]|X_1] \qquad\qquad \text{(uses CI)} \\
&= \sum_y P(Y=y|X_1)\mathbb{E}[X_2|Y=y] \\
&=: f(X_1)^\top A,
\end{aligned}
$$

where $f : \mathbb{R}^{d_1} \to \Delta_{\mathcal{Y}}$ satisfies $f(x_1)_y = P(Y=y|X_1=x_1)$, and $\boldsymbol{A} \in \mathbb{R}^{\mathcal{Y} \times d_2}$ satisfies $\boldsymbol{A}_{y,:} = \mathbb{E}[X_2|Y=y]$. Here $\Delta_d$ denotes simplex of dimension $d$, which represents the discrete probability density over support of size $d$.

Let $\boldsymbol{B} = \boldsymbol{A}^\dagger \in \mathbb{R}^{\mathcal{Y} \times d_2}$ be the pseudoinverse of matrix $\boldsymbol{A}$, and we get $\boldsymbol{BA} = \boldsymbol{I}$ from our assumption that $A$ is of rank $|\mathcal{Y}|$. Therefore $f(\boldsymbol{x}_1) = \boldsymbol{B}\psi(\boldsymbol{x}_1), \forall x_1$. Next we have:

$$
\begin{aligned}
\mathbb{E}[Y|X_1 = \boldsymbol{x}_1] &= \sum_y P(Y=y|X_1 = \boldsymbol{x}_1) \times y \\
&= \hat{\boldsymbol{Y}}f(\boldsymbol{x}_1) \\
&= (\hat{\boldsymbol{Y}}\boldsymbol{B}) \cdot \psi(X_1).
\end{aligned}
$$

Here we denote by $\hat{\boldsymbol{Y}} \in \mathbb{R}^{k \times \mathcal{Y}}, \hat{\boldsymbol{Y}}_{:,y} = y$ that spans the whole support $\mathcal{Y}$. Therefore let $\boldsymbol{W}^* = \hat{\boldsymbol{Y}}\boldsymbol{B}$ will finish the proof.

$\square$

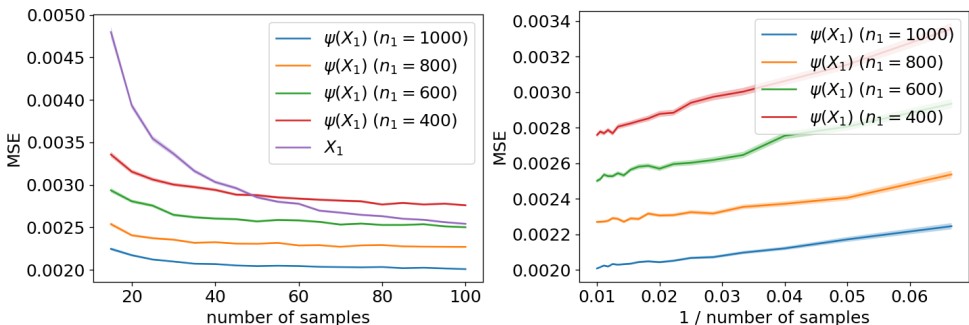

Figure 3: **Left**: MSE of using $\psi$ to predict $Y$ versus using $X_1$ directly to predict $Y$. Using $\psi$ consistently outperforms using $X_1$. **Right**: MSE of $\psi$ learned with different $n_1$. The MSE scale with $1/n_2$ as indicated by our analysis. Simulations are repeated 100 times, with the mean shown in solid line and one standard error shown in shadow.

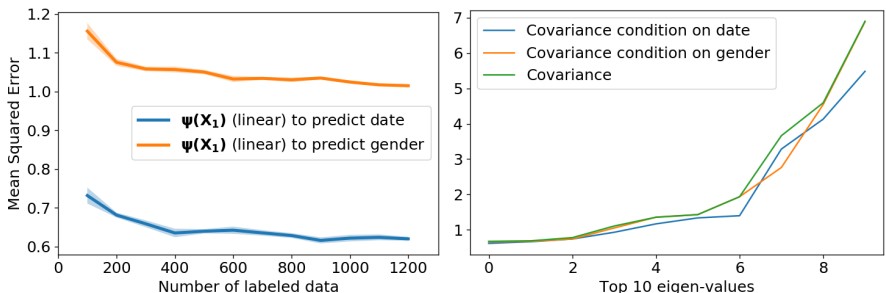

Figure 4: **Left**: Mean Squared Error comparison of predicting gender and predicting date. **Right**: the spectrum comparison of covariance condition on gender and condition on date.

## J    More on the experiments

In this section, we include more experiment setup and results.

**Simulations.**    All the experiments are performed on a desktop computer with Intel i7-8700K, 16GB RAM.

Following Theorem 4.2, we know that the Excessive Risk (ER) is also controlled by (1) the number of samples for the pretext task ($n_1$), and (2) the number of samples for the downstream task ($n_2$), besides $k$ and $\epsilon_{CI}$ as discussed in the main text. In this simulation, we enforce strict conditional independence, and explore how ER varies with $n_1$ and $n_2$. We generate the data the same way as in the main text, and keep $\alpha = 0, k = 2, d_1 = 50$ and $d_2 = 40$ We restrict the function class to linear model. Hence $\psi$ is the linear model to predict $X_2$ from $X_1$ given the pretext dataset. We use Mean Squared Error (MSE) as the metric, since it is the empirical version of the ER. As shown in Figure 3, $\psi$ consistently outperforms $X_1$ in predicting $Y$ using a linear model learnt from the given downstream dataset, and ER does scale linearly with $1/n_2$, as indicated by our analysis.

**Computer Vision Task.**    For the context encoder part, we use all the recommended hyperparameter as in the provided source codes. For the downstream resnet18 regression, we perform grid search over the hyperparameters to achieve best performance. Specifically, we set the batch size to be $24$, and traing the resnet18 for $50$ epoches. One pass of training (loops over all the settings with different number of labeled data) is finished within 6 hours. All the experiments are performed on a desktop computer with Intel i7-8700K, 16GB RAM, and NVIDIA Geforce 1080. Training of the context encoder is finished within 12 hours. The yearbook dataset is distributed under BSD license.

Following the same procedure, we try to predict the gender $Y_G$. We normalize the label $(Y_G, Y_D)$ to unit variance, and confine ourself to linear function class. That is, instead of using a context encoder to impaint $X_2$ from $X_1$, we confine $\psi$ to be a linear function. As shown on the left of Figure 4, the MSE

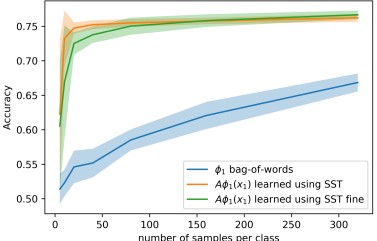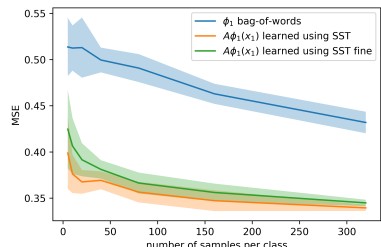

Figure 5: Performance on SST of baseline $\phi_1(\boldsymbol{x}_1)$, i.e. bag-of-words, and learned $\psi(\boldsymbol{x}_1)$ for the two settings. **Left:** Classification accuracy, **Right:** Regression MSE.

of predicting gender is higher than predicting dates. We find that $\|\boldsymbol{\Sigma}_{\boldsymbol{X}_1\boldsymbol{X}_1}^{-1/2}\boldsymbol{\Sigma}_{\boldsymbol{X}_1 X_2|Y_G}\|_F = 9.32$, while $\|\boldsymbol{\Sigma}_{\boldsymbol{X}_1\boldsymbol{X}_1}^{-1/2}\boldsymbol{\Sigma}_{\boldsymbol{X}_1 X_2|Y_D}\|_F = 8.15$. Moreover, as shown on the right of Figure 4, conditioning on $Y_D$ cancels out more spectrum than conditioning on $Y_G$. In this case, we conjecture that, unlike $Y_D$, $Y_G$ does not capture much dependence between $X_1$ and $X_2$. And as a result, $\epsilon_{CI}$ is larger, and the downstream performance is worse, as we expected.

**NLP Task.** We look at the setting where both $\mathcal{X}_1$ and $\mathcal{X}_2$ are the set of sentences and perform experiments by enforcing CI with and without latent variables. The downstream task is sentiment classification with the Stanford Sentiment Treebank (SST) dataset [53], where inputs are movie reviews and the label set $\mathcal{Y}$ is $\{\pm 1\}$. We learn a linear representation $\psi(X_1) = \boldsymbol{B}\phi(X_1)$ in the SSL phase as defined in Section 4. Here we $X_1$, we pick $\phi(X_1)$ to be the bag-of-words representations of the movie review $X_1$, which has a vocabulary size of 13848 For $X_2$ we use a $d_2 = 300$ dimensional embedding of the sentence, that is the mean of word vectors (random Gaussians) for the words in the review $X_2$. For SSL data we consider 2 settings, (a) enforce CI with the labels $\mathcal{Y}$, (b) enforce CI with extra latent variables, for which we use fine-grained version of SST with label set $\bar{\mathcal{Y}} = \{1, 2, 3, 4, 5\}$[10].. In this setting, for every label $y \in \mathcal{Y}$ (or $\bar{y} \in \bar{\mathcal{Y}}$), we independently sample movie reviews $X_1$ and $X_2$ from the class $y$ (or $\bar{y}$), thus simulating the CI (or approximate CI) condition. We test the learned $\psi$ on SST binary task with linear regression and linear classification; results are presented in Figure 5. We observe that in both settings $\psi$ outperforms $\phi_1$, especially in the small-sample-size regime. Exact CI is better than CI with latent variables, as suggested by theory.

The function $\psi$ (or equivalently matrix $\boldsymbol{B} \in \mathbb{R}^{300 \times 13848}$) is learnt by minimizing $\|X_2 - \boldsymbol{B}\phi(X_1)\|^2$ averaged over the SSL train data with an $\|\cdot\|_F^2$ penalty on the matrix $\boldsymbol{B}$. We use the scikit-learn RidgeRegressionCV[11] solver for this with regularizer parameters in the list $[0.001, 0.1, 10, 1000]$. Plotting Figure 5 took less than an hour when using 8 Intel(R) Xeon(R) Silver 4214 CPUs on a cluster.

---

[10]Ratings $\{1, 2\}$ correspond to $y = -1$ and $\{4, 5\}$ correspond to $y = 1$

[11]https://scikit-learn.org/stable/modules/generated/sklearn.linear_model.RidgeCV.html