# OpenReview forum: "Predicting What You Already Know Helps: Provable Self-Supervised Learning"
_NeurIPS.cc/2021/Conference — NeurIPS 2021 Poster_

### Official Review · Reviewer_BbkL · 2021-07-13

**Rating:** 7
**Confidence:** 4

**Summary:**

This paper studies whether reconstruction-based pretext task can provide representations helpful for downstream tasks, which is a question of good interests in the community.

The authors give an affirmative answer that when (approximate) conditional independence holds, reconstruction as a pretext task provides representations such that 1) an additional linear layer is sufficient to ensure small approximation error on classification, and that 2) helps to reduce the sample complexity of the classification task from complexity of the function class (without pretraining) to linear in data dimension.

These theoretical results are empirically verified on synthetic, vision (inpainting), and NLP (sentiment classification) tasks.

**Limitations And Societal Impact:**

Future directions pointed out by the authors include 1) analyze the effect of fine-tuning to bring down the approximation error terms; and 2) study other possible conditions under which pretext tasks are helpful for downstream learning, since CI is sufficient but likely not necessary.

Societal impact is not applicable since this work is theoretical.

**Main Review:**

**Significance and novelty of the contribution**: To the best of my knowledge, this paper is the first to provide a statistical connection on reconstruction-based self-supervised methods and downstream tasks. The setup and (approximate) conditional independence assumption is reasonable; for example, several recent works on multi-view redundancy adopt similar assumptions.
 - It's much appreciated that the theoretical results are verified by multiple experiments. However, it would be better if the authors could comment more on the practical implications of the assumptions required for theory.

**Relation to prior work**: This paper is clearly situated in relation to prior work. The discussions on comparison to prior work (esp multi-view) are sufficiently detailed which is appreciated.

**Clarity & writing**: This paper is well structured and easy to follow. Some minor notes:
- Typo: Thm 3.2: $n$ should be $n_2$.
- Line 235: $d_2 < km$: $m$ hasn't been introduced yet.
- Line 239: on the RHS of $\psi^*$, $x_1$ should be $\phi_1(x_1)$ (otherwise $\psi^*$ is linear in $x_1$).
- Typo: line 321: remove the first word "this".

======
Post-rebuttal update: I thank the authors for their responses and keep my score of recommending an accept.

**Time Spent Reviewing:**

3.5 (have seen the paper before)

---

> ### Author Response · Authors · 2021-08-11
> **Response to Reviewer BbkL**
>
> We thank the reviewer for the valuable feedback and constructive suggestions.
>
> *Q: ”it would be better if the authors could comment more on the practical implications of the assumptions required for theory”*
>
> *A:* Our main practical implication is: the pretext task is suited for learning useful representations when the correlation between pretext task and input is largely captured by the target Y. We have an example in Appendix I that supports the claim with the Yearbook dataset on different tasks. We leave a more thorough empirical investigation to future work.
>
> Thank you for pointing out typos and clarity-related issues. We will fix them in the revision.

---

> > ### Comment · Reviewer_BbkL · 2021-08-22
> > **Thank you for the response**
> >
> > Thank you for your comments on the practical implication, I will stand by my original score.

---

### Official Review · Reviewer_Y1dq · 2021-07-15

**Rating:** 8
**Confidence:** 3

**Summary:**

- The paper theoretically analyzes self-supervision under some simplifying assumptions. They begin with a conditional independent setting, where X1 is conditionally independent of X2 given Y.
- The high-level idea is to first pre-train a model that predicts X2 from X1. This can be done on unlabeled data. Then they can fit a model on top of representations learned by this model, to predict Y.
1. Ground truth self-supervision: They show that if we learn E[X2 | X1], the ground truth self-supervision function, then under mild invertibility conditions, E[Y | X] is a linear function of E[X2 | X1]. Further, E[X2 | X1]'s effective dimension is the same as Y, so this linear function can be fit using very few examples. In contrast, the function from X1 or X2 to Y can be very complicated / non-linear and require many examples to fit.
2. Linear self-supervision: They then examine what happens if we fit a linear function (on feature space) to predict X2 from X1. This can learn a useful lower dimensional embedding, and then we train a linear classifier to predict Y on top of this. They show this is more sample efficient than just training a linear classifier from X1 to Y.
3. Relaxing conditional independence: They quantitatively examine how this degrades when we don't exactly have conditional independence. Their bounds will still be good if things are "almost" conditionally independent.


**Limitations And Societal Impact:**

Looks fine.

**Main Review:**

After rebuttal: added comments below

---------------------------------------------------------------------------------------

Strengths
- I think this is a strong paper. Result (1), that E[Y | X1] is a linear function of E[X2 | X1] under conditional independence, is already conceptually quite intriguing to me. It's easy to prove if someone told you the result was true and asked you to prove it, but it is nice and surprising. In fact, I mentioned the result to a couple people who are really good at ML theory and work in related areas, and they didn't believe the result at first - they figured it could be true for Gaussians under conditional independence, but not in general (for discrete Y).
- I think results (2) and (3) make the paper even stronger, because they handle situations where we don't have exact conditional independence, or if we learn a linear model (even when the ground truth model is much more complicated).

Weaknesses
- Please label all equations for easy reference, both in the main paper and appendix
- The quality of writing in the Appendices should be substantially improved. To be clear, I'm not saying anything is wrong. I sampled some proofs and read them carefully, and they do appear to be correct, but they're very hard to follow.
- Focusing on the proof of Theorem B.3. In line 693, in the first step, I believe it should be <= -2<N, psi(X1) W* - psi(X1) \hat{W}>
- I'm not sure what P_{\psi(X_1)} is, what are we projecting to? And why does that not change N?
- In the last line of the equation array in line 697, you have an extra k in the denominator, it's not clear why that disappears?
- In the first line to the second line of the equation array in line 697, how does a matrix vector product (which is a vector) transform into a scalar (frobenius norm)?
- Unfortunately, it's hard for me to verify the results from Section 3.2 onwards, especially for Section 4. I did some basic sanity checks to see that they seem reasonable, and sampled some parts of the Appendix, and it looks likely correct overall.
- Part of why the proof are hard to verify is because the results are somewhat complicated (which isn't a bad thing!). But the other more important issue is that there aren't substantial proof intuitions given in the main paper, and the proofs in the appendix don't seem to be written very well (for example, a large number of typos just between line 693 and 697). I only sampled a small proportion of proofs, so it's possible that I happened to sample the bad examples.
- I'd personally err on the side of having fewer results, that are very clearly explained and proved, and that the reader can fully understand.

I think it's worth comparing this framework much more carefully with co-training. Both co-training and self-supervision are essentially semi-supervised learning methods, and co-training also assumes conditional independence. How would the bounds compare between the two? The initial co-training paper assumed conditional independence, and later versions assume weaker properties such as expansion [1], inspired from spectral graph theory.

I think it's worth noting that k often increases exponentially. E.g. if Z represents r 'semantic' features (like color, rotation, etc), each of which can take on one of s values, then k = s^r. Could be interesting for future work to avoid this blowup.

Other comments
- Why do we need to apply PCA in remark 4.1? E.g. in theorem 3.2, we didn't need PCA, because the complexity depends on the trace of some matrix so automatically improves if the data lies in a lower dimensional subspace.

Minor comments:
- Remark 3.2 uses m, which is defined later in assumption 4.1
- Claim A.3 has typos, it says it's a concentration result for sub-Gaussians, but assumes the data is Gaussian. Also unclear what Sigma_X is if it's sub-Gaussian?
- I think Theorem 3.2 should use n_2 instead of n in the theorem statement.
- Line 274 appears to change the definition of excess risk - maybe use a different terminology?
- In theorem 4.2, is assumption 3.2 actually not needed for universal feature maps?
- In theorem 4.2, do you really need both a lower bound on n1 *and* an upper bound on the error of the pretext tasks, \epsilon_pre^2?

[1] Co-Training and Expansion: Towards Bridging Theory and Practice. Maria-Florina Balcan. Avrim Blum. Ke Yang.


**Time Spent Reviewing:**

6 hours

---

> ### Author Response · Authors · 2021-08-11
> **Response to Reviewer Y1dq**
>
>
> Thank you for the positive review and detailed feedback.
>
> *1: “Appendix is hard to follow” and “Proofs hard to verify”*
>
> *A:* Thank you for reading our proofs carefully and the feedback. We will clean up our appendix in the revision.
>
> *2: “what is P_{\Psi} and why doesn’t it change N”*
>
> *A:* Notice $\Psi(\mathbf{X})$ is an $n\times d_2$ matrix (where usually the number of samples $n>d_2$, the dimension of the representation) and $P_{\Psi}$ is the projection matrix onto the column span of $\Psi(\mathbf{X})$, which is a $d_2$ dimensional subspace. In general for any matrices $A,B$ of dimension $n\times d_2$, we have that $\langle A, B\rangle = Trace(A^TB)=Trace(A^TP_{A}(B)) = \langle A,P_A(B)\rangle$. Therefore the projection won’t change the value of the inner product.  We will clarify this point in the revision.
>
> *3: “how k in the denominator disappear in line 697” and “the first line is a vector and later on it becomes a scalar”*
>
> *A:* Thank you for catching the typos. We missed the norm signs in the first line, and there is not any $k$ in the denominator.
>
>
> *4: “Compare to cotraining”*
>
> *A:* One major conceptual difference is in co-training some labeled data is used at the start to learn weak predictors, whose performance is then boosted using unlabeled data, which makes it closer to the semi-supervised learning setting. In our setting, we only use unlabeled data in the first phase and labeled data is reserved for the final stage of linear classification. We will add this and some more detailed comparisons to the assumptions made in the co-training literature.
>
>
> *5: “I think it's worth noting that k often increases exponentially. E.g. if Z represents r 'semantic' features (like color, rotation, etc), each of which can take on one of s values, then k = s^r. Could be interesting for future work to avoid this blowup.“*
>
>
> *A:* While there can be exponential (~$s^r$) latent variables not all of them may have a high probability. Many combinations of (color, rotation, label) might be invalid or have tiny probability, and those can be ignored by suffering a small amount in $\epsilon_{CI}$ instead. Furthermore in some specialized settings (like HMMs, topic modeling), while the number of latent variables might seem exponential (all possible mixture of hidden states or mixture of topics), the specific generative model allows us to just use number of latent variables to be proportional to the number of hidden states or topics. Please refer to the response to reviewer hGAR for a discussion on HMMs and topic models.
>
> Mathematically, the SVD of $\Psi = \sum_i \sigma_i u_i v_i^T$ , and $\sigma_i = Pr$(latent var in state $i$ ) . If the singular values of $\Psi$ decay, then the smaller ones can be discarded. In the example you described, it is likely that not all $s^r$ combinations are equally likely, so many can be discarded.
>
> This is not to say that the analysis is ideal, and showing similar guarantees under weaker assumptions is certainly an interesting future direction. We will discuss these points in the revision, thank you for the question.
>
>
> *6: “Why do we need to apply PCA in remark 4.1”*
>
> *A:* Under the assumption of approximate conditional independence, our representation matrix
> $\Psi(\mathbf{X})$ is of rank $d_2$, even though it is close to being rank $k$. Therefore with PCA we will be able to improve the estimation error from $d_2$ to $k$ by explicitly making it low rank. It is also possible to use ridge regression instead of linear regression to get a risk bound with the trace of the data matrix, but this might give a slow rate of ($O(\sqrt{1/n_2})$) instead of the fast rate in our result.
>
>
>
> *7: “do you really need both a lower bound on $n_1$ and an upper bound on the error of the pretext tasks, $\epsilon_{pre}^2$”*
>
> *A:* Yes, the lower bound on $n_1$ is quite weak and unrelated to the pretext task error. It is just to ensure that the empirical covariance of representation $\psi$ is well-conditioned. The main requirement for $n_1$ will come from having to make $\epsilon_{pre}$ small.
>
> *8: “In theorem 4.2, is assumption 3.2 actually not needed for universal feature maps?”*
>
> *A:* Thank you for catching that. We assume Assumption 4.1 and 3.2 for $\tilde \psi$ and $\psi^*$ and Assumption 4.2 for non-universal feature maps. We accidentally had Assumptions 3.2 and 4.2 swapped in the theorem statement.

---

> > ### Comment · Reviewer_Y1dq · 2021-08-17
> > **Thanks for the response**
> >
> > I don't agree with the response to point 5. "Many combinations of (color, rotation, label) might be invalid or have tiny probability" - sounds unlikely, seems to me that it'd be exponential. Thanks for agreeing to discuss this in the paper.
> >
> > I think it's a good paper overall. The only reservation I have, for the AC to decide, is as I said the Appendix seems poorly written and hard to verify precisely, and many proof intuitions aren't given in the main body. But maybe just the parts I sampled. The results and bounds do look plausible and intuitively correct. Thanks to the authors for agreeing to fix these!

---

### Official Review · Reviewer_hGAR · 2021-07-17

**Rating:** 4
**Confidence:** 4

**Summary:**

This paper suggests a theoretical model to capture
why reconstruction-based self-supervised learning leads to good representations, and improved downstream performance.
The suggested model is that the pretext task "factors through" the supervised labels, i.e. that X_1 -- Y -- X_2 is Markov, where X_1-->X_2 is the pretext task, and X_1 --> Y is the supervised task.
In such a setting, it is shown that the optimal pretext function has certain good properties: it linearly separates the supervised task, and
is "low rank", so can be used for efficient downstream learning.

**Limitations And Societal Impact:**

Societal impacts N/A.

**Main Review:**

## Strengths
- The main realization of this paper is interesting: it gives a setting
where learning an "auxiliary" task is actually linearly-related to the task of interest (without assuming any linear structure in the tasks themselves). This is the statement of Lemma 3.1, which I consider to be the main conceptual contribution of this paper.
Lemma 3.1 is technically straightforward, but the statement is interesting.

- The problem studied is important: we do not have much theoretical understanding of why SSL methods work in practice.

## Weaknesses

- The primary weakness of this paper is that the statistical (Markov) assumption is extremely strong (even in its "approximate" formulation),
and does not clearly connect to any situations in practice.
There are no experiments that support the assumptions, and the intuitions presented are not convincing. That is, although the claimed motivation
of this paper is to study "What conceptual connection between pretext and downstream tasks ensures good representations?", the results in the paper do not clearly shed light on this question in realistic settings.
In particular, I do not think Assumption 4.1 will hold in practice, for reasonable values of $m$. More importantly, I do not think it captures the spirit of many pretext tasks: there could be very high mutual information between X_1 and X_2 in practice, even when conditioned on the supervised label (Y). This is especially the case when the supervised task is much "easier" than the pretext task.

- I do not believe the technical contributions are strong enough to stand alone, without appropriate motivation from practice.
Most of the technical results are essentially corollaries of
Lemma 3.1, expanded in straightforward ways.
I do not think these results contribute significantly
to the mathematical theory of learning (and they do not develop any new technical tools, as far as I am aware).

- The empirical section is lacking. It would have been nice to see some attempts to empirically justify the theoretical assumptions made in this work. For example, by trying to approximately estimate the conditional independence in realistic settings.
However, the experiments on real data do not attempt to probe the validity of the assumptions.

**Time Spent Reviewing:**

3

---

> ### Author Response · Authors · 2021-08-11
> **Response to Reviewer hGAR**
>
> We thank the reviewer for the valuable feedback and constructive suggestions. We address your concerns in detail and ask that you raise your score.
>
> *1: “assumption is extremely strong, and does not clearly connect to any situations in practice”.*
>
> A: We would like to emphasize that our assumption is already the most relaxed one compared to prior or concurrent theoretical works that obtain similar types of risk bounds. Our approximate conditional independence (ACI) assumption can always be made to hold for a large enough $\epsilon_{CI}$. The question is whether or not this quantity gives us a vacuous bound, which is a common question for any kind of theoretical bound. Via simulations, we show that the bound is in fact tight in many regimes. For real-life datasets, besides observing good performance of the method, we also establish the correlation between this $\epsilon_{CI}$ quantity and downstream performance on vision and NLP datasets (see the answer to next question).
>
> As with most theoretical results, the assumptions are bound to be stronger than what holds in practice. Our current assumption is the weakest one thus far that can admit such a theoretical analysis, and requires different techniques to study it. We do not believe that this is the end of the story, and analyzing SSL under assumptions that are even closer to practice is certainly a very interesting future direction.
>
> On the theoretical side, the strength of our model can be demonstrated by instantiating it for well-studied theoretical generative models like Hidden Markov Models (HMMs) (or topic modeling [1] and mixed-membership models). For the reconstruction task of predicting the second half of the sequence (document) from the first half, epsilon_{CI}$ can be shown to be 0 with the number of latent variables of the order of the number of hidden states (or topics) as opposed to the number of observations (or vocabulary size). The learned representation can learn the underlying latent state distribution (or latent topic mixture) with very few samples. We will add these examples in the revision to highlight the utility of our framework even further.
>
>
>
> *2: “Most of the technical results are essentially corollaries of Lemma 3.1, expanded in straightforward ways.”*
>
> A: We respectfully disagree with this claim. As also pointed out by reviewer 3 (Y1dq), the proofs for our main results are actually quite complicated and are certainly not direct consequences of Lemma 3.1. In Appendix H, we also demonstrated four different ways of proving Lemma 3.1. Although we provided the most intuitive and easy-to-understand way in the main paper, that approach cannot be easily extended to our other main results. Our final results require non-trivial analyses that use cross-covariance operators in the Hilbert space of random variables.
>
> We have also included several extensions on analyzing encoder-decoder-based representation learning, the connection between predicting $Y$ with both $X_1$ and $X_2$ and only using $X_1$, and results with classification and regression. None of these results are straightforward extensions of Lemma 3.1 in our opinion.
>
> *3: “The empirical section is lacking.”*
>
> *A:* We would like to emphasize that the main contribution of the paper is theoretical, which we have argued above is non-trivial. We did conduct empirical verifications with simulations, computer vision and NLP tasks, some of which are in the appendix since they are not the main focus of this paper.
>
> Some findings that directly or indirectly validate the theory are:
>
> - Simulation experiments verify the tightness of the bounds, among other things.
> - Using the output of the SSL task $\psi(X_1)$ directly for linear probe (as opposed to intermediate layers used in practice), as suggested by theory, is quite effective compared to various baselines in the CV task (Figure 2).
> - The conditional independence parameter $\epsilon_{CI}$ and its correlation to downstream performance is demonstrated for the CV task in Section I.
> - Effective number of latent variables that make the components conditionally independent is also studied for an NLP task in Section I.
>
> [1] Blei, David M., Andrew Y. Ng, and Michael I. Jordan. "Latent dirichlet allocation." the Journal of machine Learning research 3 (2003): 993-1022.

---

> > ### Comment · Reviewer_Y1dq · 2021-08-17
> > **Complicated proofs**
> >
> > Just to clarify, it's possible the result is complicated and the proofs are straightforward. I didn't look into the proof techniques for the later sections, so I don't think what I said applies here (the authors quoted my review in this response).

---

> ### Comment · Reviewer_Y1dq · 2021-08-17
> **Good points**
>
> I've read the review, and reviewer hGAR raises some good points. I agree conditional independence is strong. But I think it's a good first step, and we have to start somewhere. Conditional independence assumptions are pretty common so I think it's a fine starting point, and they do look at deviations from this as well. I see where you're coming from and think it's reasonable, but I'd still maintain my "accept" rating.

---

> > ### Comment · Reviewer_hGAR · 2021-08-18
> > **Response to Y1dq**
> >
> > It is not clear to me that conditional independence is a "good first step".
> > It is some assumption, that lets the authors prove some lemma.
> > Is this lemma at all related to the stated objective, of understanding why SSL methods works in practice? I see no real experiments which support this, nor any justification that I believe.
> >
> > It is not enough to simply do some mathematics, and then say some words to claim it's related to practice.
> >
> > I appreciate that making progress in ML theory is hard, and we need to start small. But we should start in the right direction, and with eyes on the goal.

---

> > > ### Comment · Reviewer_Y1dq · 2021-08-18
> > > **Response to hGAR**
> > >
> > > These are good points. That said, I think it's a fairly common assumption in related settings.
> > >
> > > For example, co-training (Blum and Mitchell, 1998) studied a semi-supervised setting where x1 and x2 are conditionally independent on y. The abstract gives a nice motivation - "For example, the description of a web page can be partitioned into the words occurring on that page, and the words occurring in hyperlinks that point to that page". They don't prove or check conditional independence either, and it's spawned a huge line of work and good algorithms. That paper is stronger (proposes a new algorithm), but it's quite a historic paper. A key advantage of self-supervision is you can do the pre-training in a task agnostic way, and use it for many tasks (as opposed to co-training which needs to be done for each task).
> > >
> > > Another examples is "A Theoretical Analysis of Contrastive Unsupervised Representation Learning" by Arora et al. They're looking at contrastive learning, where e.g. it is common to take crops of an image as x1 and x2. These are not conditionally independent on the label or latent either.
> > >
> > > One could also think of the multi-modal setting, e.g. predicting metadata or text from images.
> > >
> > > I agree it's hard to tell if its realistic, and we should move beyond that, but seems like a fine start to me. And they look at deviations from the conditional independence assumption. I agree that future work could try to provide justification for this, but that's really hard in ML and I see it as being a separate project.

---

### Official Review · Reviewer_f8xA · 2021-07-19

**Rating:** 6
**Confidence:** 2

**Summary:**

This paper aims to theoretically explain and investigate the benefit of reconstruction-based self-supervised learning. This paper studies a specific problem with two input features and one downstream label and shows how the specific connection between them can guarantee successful self-supervised learning.  In particular, this paper quantifies how the approximation independence between two input features can be extracted to solve the downstream tasks. Results show that self-supervised learning can successfully learn the ground-truth with much smaller labeled sample complexity.


**Limitations And Societal Impact:**

This paper has no potential negative societal impact.

**Main Review:**

I have to say I am not familiar with self-supervised learning and its theory. From a general point of view, the theory established in this paper is rigorous and makes sense to me. The assumptions, theorems, claims are clearly presented.

However, my main concern is regarding the clarity of the paper. For example, from the proof sketch of Lemma 3.1, it is not clear why the matrix A has rank $k=|Y|$? Why sometimes the authors use the $\arg\min$ notation to denote $f^*$ and $\psi^*$, and sometimes they are written using the expectation notations.

There are a bunch of notations used in the paper (e.g., $b_0$, $\beta$, $\alpha$, $k$, $m$, $\rho$, etc). It would be better to summarize them somewhere. Besides, I am wondering whether the authors can briefly elaborate on which of them are typically large (e.g., $\omega(1)$) and which of them are typically small (i.e., $o(1)$).

Lastly, does the conditional independence really hold in practice, or how small is their correlation? The authors should add more content to justify this (experimental justification could be better).

=========================

Thanks for the authors' response. I agree the paper presentation can be improved after fixing the problems I mentioned. I would like to keep my evaluation marginally acceptance.

**Time Spent Reviewing:**

3 hrs

---

> ### Author Response · Authors · 2021-08-11
> **Response to Reviewer f8xA**
>
> We thank the reviewer for the valuable feedback and constructive suggestions. We will clarify these in the revision. Please consider increasing your score, given our changes.
>
> *1: ”why the matrix A has rank k=|Y|”?*
>
> *A:* This is an assumption that we discuss right after the proof sketch of lemma 3.1. $A:=\mathbb{E}[X_2|Y] \in \\mathbb{R}^{k\times d_2}$ being full row-rank means $X_2$ is correlated with every direction of Y. This is a necessary requirement for $X_2$ to be a meaningful pretext task.
>
> *2: “why sometimes use argmin, and sometimes use conditional expectation for $f^\ast$ and $\psi^\ast$”*
>
> *A:* The argmin is used in the definition of $f^*$ and $\psi^*$, while the conditional expectations are the closed form expression for the argmin functions $f^*$ and $\psi^*$. This is mostly a presentation issue, and we just wanted to emphasize in the beginning of the paper that the conditional expectation is the argmin of solving the pretext task. We will clarify this distinction in the revision.
>
> *3: “summarize notation: $b_0, \beta, \alpha, k, m, \rho$”*
>
> *A:* Thank you for the suggestions, we will add a notation glossary and some discussions of these terms in the revision. $b_0$ is a small universal constant that doesn’t depend on any other variables considered in our paper. $\beta$ captures the correlation between $Y$ and $X_2$ and therefore depends on the scaling of $X_2$. $k$ is the cardinality of label $Y$ and is for instance 10 for cifar10 or MNIST datasets. $\sigma$ is the noise level and is usually much smaller than 1. $m$ is the size of the latent variable. In our setting, $m<d_2/k$. $\rho$ is the $\psi_2$-norm of sub-gaussian variables ($\rho$ is 1 for Gaussian random variables.).
>
> *4: “Does CI hold in practice? Empirical verification is appreciated.”*
>
> *A:* Even though CI doesn’t always hold in practice, our results suggest that one can benefit from designing pretext tasks that satisfy our assumptions. Our practical implication suggests that: the pretext task is suited for learning useful representations when the correlation between pretext task and input is largely captured by the target Y. There actually exists some paper (by other groups) that followed our results to design pretext tasks for speech representation learning, although we cannot disclose the paper due to anonymity policy.

---

### Decision · Program_Chairs · 2021-09-27

**Decision:**

Accept (Poster)

**Comment:**

Three of four reviewers generally agree to recommend this paper for acceptance, and I agree with the strengths that they highlight, especially as reviewer Y1dq summarizes them.

Reviewer hGAR, who recommends rejection, makes a valid point as well: the experimental section and the discussion of assumptions leaves room for improvement. I encourage the authors to take the committee's suggestions about these parts into consideration when revising the draft.

For instance, two reviewers would have liked to see an empirical signal for whether the conditional independence assumption holds in a real data setting (not simulation). Even if the authors choose not to do this, it is natural to ask and it is lacking from the experiments, and so it is ultimately a limitation. It's best to be clear about whether the experiments are meant to closely verify theory (including its assumptions), or whether they are meant only to show that the analysis is at least not at odds with empirical observations. The latter seems to better describe this paper's experiments, but it is a weaker point than the former, and discussing limitations ultimately helps with clarity.

I will add that there are other ways to support an assumption than by experiment. One is to identify example mathematical constructions that satisfy them (e.g. topic models, HMMs). Another is to point to similar assumptions made in related prior work. The paper does the latter reasonably well. If it is easy enough to do, perhaps the authors can make use of an example construction or two as well.

Overall I second Y1dq's remarks in discussion that this is a field with little theoretical analysis, and that what constitutes a reasonable abstraction and assumption is not yet clear. Making assumptions for progress is fine, especially if similar ones have been make in related work. The validation and discussion of assumptions in this paper could be improved as hGAR emphasizes, but overall the support from the remaining reviewers leads me to recommend acceptance.